# Sgk1 upregulation in hippocampus-projecting amygdala neurons underlies the delayed onset of PTSD-like avoidance behavior

Jia-Xin Zou [1,2,3,7], Wei-Zhu Liu[3,4,7], Ya-Qing Li [1,2,3], Yuan-Yuan Li[5], Wen-Jie You[3], Han-Qing Pan[3,5], Chun-Yan Wang[3,6], Wen-Hua Zhang [1,2,3,6] ✉ & Bing-Xing Pan [1,2,3] ✉

Excessive avoidance is a core symptom of post-traumatic stress disorder (PTSD), yet its underlying circuit mechanisms remain poorly understood. Here, using a mouse model of PTSD induced by inescapable footshock, we observed a delayed and prolonged increase in avoidance behavior associated with selective activation of basolateral amygdala (BLA) projection neurons (PNs) targeting ventral hippocampus (BLA→vHPC PNs), but not nucleus accumbens, in both sexes. This projection-specific activation results from enhanced neuronal excitability and excitatory transmission driven by gluco-corticoid receptor (GR) signaling. Among the cascade of GR signaling molecules, we identified serum- and glucocorticoid-regulated kinase 1 (Sgk1) as a key downstream mediator linking stress exposure to the hyperactivation of BLA→vHPC PNs and PTSD-like avoidance behavior. Manipulating Sgk1 expression bidirectionally regulates neuronal activity and susceptibility to stress-induced avoidance. These findings underscore the critical role of projection-specific upregulation of Sgk1 in BLA PNs in the pathogenesis of PTSD-like avoidance behavior.

Exposure to extreme life threats, such as witnessing the demise of loved ones, or being a victim of earthquakes, can lead to severe psychological impacts, including intrusive memories, augmented avoidance behavior, heightened arousal[1,2]. While these symptoms typically diminish or disappear for most people over time, a significant proportion of individuals may continue to experience them for several months or even longer, potentially leading to the development of post-traumatic stress disorder (PTSD)[1,3]. Studies have estimated PTSD affects approximately 6–8% of the general population[4,5], making it one of the most common mental disorders. Current clinical treatments of PTSD are limited, primarily consisting of psychotherapeutic interventions and pharmacotherapy, with often unsatisfactory outcomes[6–10]. There is an urgent need to thoroughly understand the pathological mechanisms underlying the onset and progression of PTSD in a hope to develop novel and effective treatment strategies.

[1]School of Life Science, Nanchang University, Nanchang, China. [2]School of Basic Medical Sciences, Jiangxi Medical College, Nanchang University, Nanchang, China. [3]Jiangxi Province Key Laboratory of Brain Science and Brain Health, Institute of Biomedical Innovation, Jiangxi Medical College, Nanchang University, Nanchang, China. [4]Department of Pathology, The First Affiliated Hospital, Jiangxi Medical College, Nanchang University, Nanchang, China. [5]Department of Neurology, The Second Affiliated Hospital, Jiangxi Medical College, Nanchang University, Nanchang, China. [6]Jiangxi Institute of Respiratory Disease, The First Affiliated Hospital, Jiangxi Medical College, Nanchang University, Nanchang, China. [7]These authors contributed equally: Jia-Xin Zou, Wei-Zhu Liu. ✉e-mail: whzhang@ncu.edu.cn; panbingxing@ncu.edu.cn

The amygdala, typically comprising the basolateral (BLA), central (CeA), and medial (MeA) subregions, is a critical brain region for emotional processing and is heavily implicated in PTSD pathology[2,11,12]. PTSD patients frequently exhibit elevated amygdala activity and aberrant responses to trauma-related stimuli[12-17], and therapeutic interventions targeting the amygdala, such as deep brain stimulation, can alleviate intrusive memories[18-20]. While considerable progress has been made in understanding the role of amygdala in fear learning and extinction[21-23], the neural circuits and molecular mechanisms underlying other core PTSD symptoms, such as heightened avoidance behavior, remain poorly defined. Rodent PTSD models and behavioral assays of avoidance[24] have begun to clarify the amygdala's dynamic role after traumatic stress. For example, single prolonged stress in rats induces time-dependent changes in basolateral amygdala (BLA) function and plasticity[25,26]. Distinct patterns of glutamatergic and GABAergic neuronal activation emerge at different time points post-stress, correlating with delayed PTSD-like avoidance and anxiety tested in approach–avoidance conflict tasks[25]. Importantly, these delayed behavioral changes are accompanied by corresponding delayed neural plasticity in BLA[24,26]. However, it remains unclear how specific BLA output pathways contribute to the avoidance behaviors and what molecular mechanisms drive these stress-induced changes.

In this study, we employed a mouse model of PTSD using repetitive electric footshocks (FS) to explore the amygdala's contribution to stress-induced avoidance behavior. We found that FS exposure led to a delayed increase of avoidance behavior, which was paralleled by a delayed activation of BLA projection neurons (PNs) targeting ventral hippocampus (BLA→vHPC PNs) as a consequence of recruitment of glucocorticoid receptor signaling. At the molecular level, FS specifically upregulated the expression of serum- and glucocorticoid-regulated kinase 1 (Sgk1) in BLA→vHPC PNs, increasing their intrinsic excitability and excitatory transmission, thereby driving avoidance behavior in the mice.

## Results

### Footshock stress exposure causes delayed and enduring PTSD-like avoidance behavior in mice

To establish a mouse model that mimics persistent PTSD-like avoidance behavior, we delivered a train of footshock stress (FS) to male mice with varying intensity (0.3 or 0.5 mA) and duration (1 or 2 days)[27,28]. The long-lasting effect of FS on mice's avoidance behavior was assessed 7 or 14 days post-FS using the open field test (OFT) and the elevated plus maze test (EPMT), two widely used approach–avoidance conflict tasks that measure anxiety-like and PTSD-related avoidance behavior[24,29-32] (Supplementary Fig. 1–3). Compared to the unstressed controls, the mice experiencing FS trains (0.5 mA for 2 consecutive days) exhibited pronounced avoidance behavior both 7 and 14 days after FS, as evidenced by far less time they spent in the center area in the OFT and in the open arms of EPMT as well as fewer entries into the open arms (Supplementary Fig. 1). In contrast, those experiencing one single day of FS trains (0.5 mA) yielded prominent avoidance behavior only at 7 days post FS, not at 14 days (Supplementary Fig. 2). FS with relatively weaker intensity (0.3 mA) for one day failed to evoke more noticeable avoidance behavior at either time point (Supplementary Fig. 3). Thus, the FS-evoked PTSD-like avoidance behavior in mice was largely dependent on the intensity and duration of FS. Therefore, a 0.5 mA FS paradigm delivered over 2 consecutive days was adopted for subsequent experiments, as it reliably elicited persistent PTSD-like avoidance behavior.

Given that numerous previous studies have indicated that severe traumatic stress often triggers PTSD-like behaviors in animals with delayed onset[25,26], we next investigated whether the avoidance behavior induced by FS followed a similar pattern. To explore this, we conducted behavioral tests on FS-exposed mice with varying durations

of recovery-short (1 day) or long (7 days) (Fig. 1a, b). In comparison to the unstressed controls (referred to as the control group throughout this study, unless specified otherwise), mice having a 7-day but not 1-day recovery displayed significant avoidance behavior and unchanged locomotor activity following FS exposure (Fig. 1c–g). Similar results were obtained in female mice using the same FS paradigm and recovery intervals (Supplementary Fig. 4a, b), with significant avoidance emerging only after 7 days, while locomotor activity remained unaffected (Supplementary Fig. 4c–g).

To further validate this FS model in recapitulating other PTSD-like phenotypes, particularly deficits in fear memory, we assessed trauma-related contextual fear memory in FS-treated mice (Supplementary Fig. 5a and 6a). Results showed that, compared with unstressed controls, FS-exposed mice displayed significantly increased freezing behavior at both 1 day and 7 days post-stress in both male and female mice (Supplementary Figs. 5b and 6b). Collectively, these findings indicate that a 2-day FS regimen caused delayed yet enduring PTSD-like avoidance behavior in mice.

### FS causes a delayed increase in the activity of BLA projection neurons

Extensive clinical and preclinical evidence highlights the critical role of the amygdala, particularly its BLA subregion, in the onset and progression of PTSD[11-13,16,20,26]. To delve deeper into how FS impacts the activity of BLA neurons, we employed in vivo fiber photometry to monitor their calcium signals using calcium indicator GCaMP6s (Fig. 1h, i). Calcium signals were monitored sequentially during stress-related tasks (i.e., exploring OFT and EPMT) before FS exposure (Pre-FS), and at 1 or 7 days post-FS (Fig. 1j). Behavioral analysis revealed that the unstressed control mice showed no significant changes in exploratory behavior across time points, as reflected by comparable time spent in and entries into the center zone of the OFT and the open arms of the EPMT. In contrast, FS-exposed mice exhibited a significant increase in avoidance behavior selectively at 7 days post-FS, but not at 1 day post-FS (Supplementary Fig. 7). Consistent with these, increased calcium signals in BLA neurons were observed when mice entered the center zone during OFT or open arms during EPMT. Notably, this elevation was markedly greater in FS-exposed mice at 7 days post-FS, an effect absent in unstressed control mice. Within the FS-exposed group, calcium signal changes (AUC of Z-score) were markedly elevated at 7 days post-FS, but not at 1 day post-FS (Fig. 1k–p). Cross-group comparisons revealed that FS-exposed mice exhibited significantly higher calcium transients than control mice exclusively at the 7-day time point, with no differences observed at baseline or at 1 day post-FS (Fig. 1k–p). Consistent with the photometry findings, immunohistochemical analysis revealed a significant increase in the number of c-Fos (a marker protein of neuronal activity) positive cells in the BLA at 7 days, but unchanged at 1 day, after FS exposure (Fig. 1q–s). Together, these findings indicate that FS triggered a delayed enhancement of BLA neuronal activity, paralleling the delayed onset of avoidance behavior observed in FS-exposed mice.

### FS preferentially enhances the activity of BLA → vHPC PNs

Accumulating evidence has unveiled the heterogeneity of individual projection neurons (PNs) within BLA in encoding positive or/and negative valence, with those projecting to ventral hippocampus (BLA→vHPC PNs) and nucleus accumbens (BLA→NAc PNs) being engaged in the aversive versus reward-related behaviors[33-36]. We sought to investigate whether these two largely non-overlapping BLA PN subsets[37,38] exhibit differential responses in FS-induced PTSD-like avoidance behavior. As depicted in Fig. 2a–p, projection-specific fiber photometry revealed that these FS-exposed mice, which exhibited significant avoidance behavior selectively at 7 days post-FS but not at 1 day post-FS (Supplementary Fig. 8), exhibited a delayed increase in

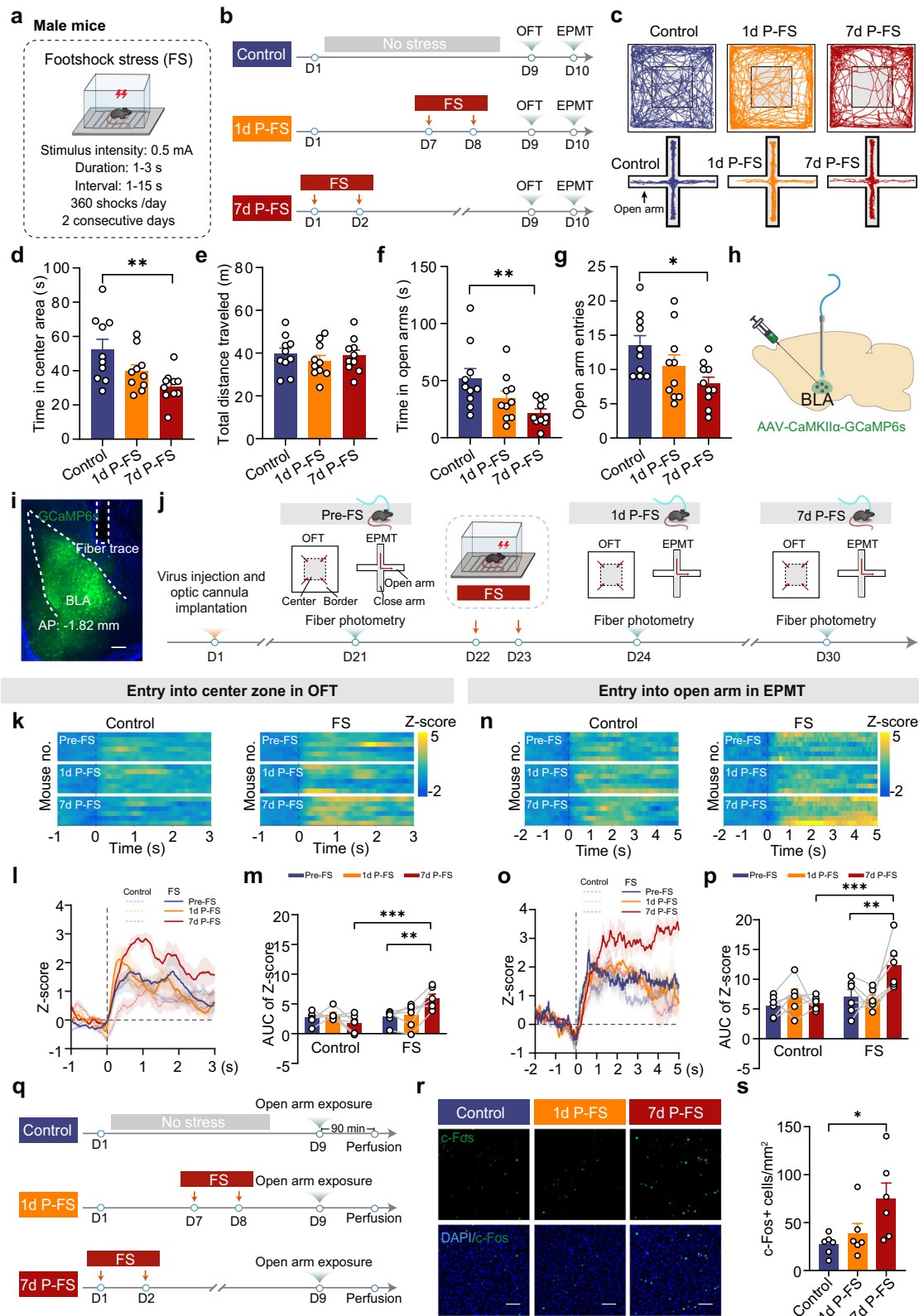

calcium signal changes in BLA→vHPC PNs when entering the center zone or open arms (Fig. 2a–h). Such changes were not observed in BLA→NAc PNs under the same conditions (Fig. 2i–p). Relative to unstressed controls, a selective increase in calcium signal changes was observed in BLA→vHPC PNs of FS-exposed mice exclusively at 7 days post-FS (Fig. 2a–h). Consistent with the photometry data, analysis of these retrogradely labeled neurons revealed that although FS increased the overall number of c-Fos positive cells in the BLA at 7 days post-FS (Fig. 2s, w), it selectively increased the proportion of c-Fos positive cells within the BLA→vHPC PNs (Fig. 2q, r, t), but not BLA→NAc PNs (Fig. 2u, v, x). No such increase was observed in either subset at 1 day post-FS (Fig. 2q–x). These findings suggest that the delayed

**Fig. 1 | Footshock stress causes delayed PTSD-like avoidance behavior and activation of BLA projection neurons in male mice. a** Schematic of FS procedure. **b** Time scheme for avoidance behavior test in the open field (OFT) and the elevated plus maze (EPMT). **c** Representative activity tracking in the OFT and the EPMT. Time in center area (**d**) and total distance traveled (**e**) in OFT ($n = 10$ mice/group). Time in open arms (**f**) and open arm entries (**g**) in EPMT ($n = 10$ mice/group). **h** Schematic of injection of AAV-CaMKIIα-GCaMP6s into the BLA and implantation of optical fiber to record calcium signals. **i** Representative image showing GCaMP6s expression and the cannula implantation onto the BLA, scale bar: 200 μm. ($n = 12$ mice). **j** Time scheme of experimental procedures for in vivo fiber photometry. Unstressed control mice received the same virus injections and fiber implantations but did not undergo the FS procedure. **k** Heatmaps of calcium signal Z-score aligned to OFT center zone entry. Each row displays trial-averaged responses per individual

mouse; the color scale denotes fluorescence intensity. **l** Peri-event average calcium signal traces (Solid line: mean; shaded area: SEM) aligned to OFT center zone entry for visualization. **m** Quantification of AUC of calcium signals from (**l**) ($n = 6$ mice/ group). Heatmaps of calcium signal Z-score (**n**) and peri-event average traces (**o**) (Solid line: mean; shaded area: SEM) aligned to EPMT open arm entry for visualization. **p** Quantification of AUC of calcium signals from (**o**) ($n = 6$ mice/group). **q** Time scheme for c-Fos staining. **r** Representative images showing c-Fos expression in BLA. Scale bar: 100 μm. **s** Summary plots of c-Fos$^+$ cells expression in BLA ($n = 6$ mice/group). Data were analyzed using one-way ANOVA (**d**, **e**, **g**), Kruskal–Wallis test (**f**, **s**) and two-way RM ANOVA (**m**, **p**), followed by Bonferroni-corrected, two-tailed pairwise comparisons for (**d**, **f**, **g**, **m**, **p**, **s**). Data are presented as mean ± SEM. *$p < 0.05$, **$p < 0.01$, ***$p < 0.001$. See Supplementary Data 1 for full statistical information. Source data are provided as a Source data file.

---

enhancement of activity in BLA PNs manifested in a projection-specific manner, with BLA→vHPC PNs being preferentially recruited over their BLA→NAc counterparts by FS.

## FS increases the intrinsic excitability of and excitatory transmission onto BLA → vHPC PNs

To elucidate the mechanisms underlying the preferential recruitment of BLA→vHPC PNs by FS, we investigated how FS would impact the firing of and synaptic transmission onto the two BLA PN subsets. As illustrated in Fig. 3a–c, relative to the unstressed controls, the BLA→vHPC PNs from mice with 7, but not 1 day recovery from FS displayed heightened firing activity in response to depolarization current injection. Analysis of the membrane properties of this PN subset revealed that the input resistance was significantly increased in the 7, but not 1, days post-FS group (Supplementary Table 1). By contrast, no noticeable differences were observed in BLA→NAc PNs in terms of their intrinsic excitability and membrane properties among the three groups (Fig. 3d–f and Supplementary Table 2).

Following these, we explored the FS's effects on synaptic transmission onto the two PN subsets by recording the miniature excitatory/inhibitory postsynaptic currents (mEPSCs/mIPSCs). BLA→vHPC PNs from the 7, but not 1d P-FS mice displayed a marked increase in mEPSCs frequency, while the mEPSCs amplitude remained consistent among groups (Fig. 3g–i). Conversely, there were no alterations in mIPSCs frequency or amplitude in both 1 d and 7d P-FS mice (Fig. 3j–l). For BLA→NAc PNs, no significant between-group differences were observed in both excitatory and inhibitory transmission (Fig. 3m–r).

Taken together, these findings strongly suggest that the delayed activation of BLA→vHPC PNs by FS is related to the increased intrinsic excitability and heightened excitatory synaptic transmission onto this specific PN subset.

## Chemogenetic suppression of BLA → vHPC PNs mitigates the delayed PTSD-like avoidance behavior in 7d P-FS mice

To determine whether the heightened activation of BLA→vHPC PNs contributes to FS-induced avoidance behavior, we applied the inhibitory designer receptors exclusively activated by designer drugs (DREADDs) method, a common chemogenetic tool, to suppress the activity of BLA→vHPC PNs (Fig. 4a, b). The effectiveness of this approach was verified by a dramatic reduction of spike rates in hM4D(Gi)-expressing BLA→vHPC PNs upon clozapine-N-oxide (CNO) perfusion (Fig. 4c). Subsequently, we found that CNO administration in vivo significantly increased the time spent by the 7d P-FS mice in the center area in the OFT (Fig. 4d, e), as well as the time they spent in the open arms and entries to the open arms in the EPMT (Fig. 4f, g). In contrast, CNO administration produced no behavioral changes in 7d P-FS mice expressing the mCherry virus in BLA→vHPC PNs (Fig. 4d–g and Supplementary Fig. 9a–e). These findings indicate that chemogenetic inhibition of BLA→vHPC PNs activity effectively attenuated FS-induced avoidance behavior.

In summary, these results suggest that the heightened activation of BLA→vHPC PNs plays a crucial role in regulating FS-induced increase in avoidance behavior.

## CORT-GR signaling is required for FS-induced increase in avoidance behavior and activation of BLA → vHPC PNs

In both PTSD patients and animal models, dysregulation of the hypothalamic-pituitary-adrenal (HPA) axis and abnormal corticosterone (CORT) release are linked to PTSD development[39–43]. CORT signaling plays a critical role in mediating stress-induced structural and functional changes in the amygdala[44,45]. To explore the role of altered CORT signaling in FS-induced avoidance behavior and BLA→vHPC PNs hyperactivation, we measured serum CORT levels after two days of FS exposure (Fig. 5a). A significant CORT increase was observed only 1 h post-FS, with no differences after 1 or 7 days of recovery, suggesting a transient CORT surge (Fig. 5b). This surge was prevented by metyrapone, a CORT synthesis inhibitor (Supplementary Fig. 10).

Could this temporary surge in CORT contribute to the delayed onset of avoidance behavior and BLA→vHPC PNs activation? Earlier studies indicating that acute CORT administration elicited delayed anxiety and BLA neuronal hypertrophy may give credence to this hypothesis[46,47]. To test this, we administered metyrapone intraperitoneally to the mice prior to FS exposure and observed a clear reversal of FS-induced avoidance behavior in the OFT (Supplementary Fig. 11a–d) and EPMT (Supplementary Fig. 11e, f), highlighting the importance of transient CORT in delayed FS-induced avoidance behavior.

CORT exerts its effect by binding to the glucocorticoid receptors (GRs) or mineralocorticoid receptors (MRs)[48,49]. To delineate their specific roles in FS-induced avoidance behavior, mice were intraperitoneally pre-treated with either the GR antagonist mifepristone or the MR antagonist spironolactone prior to FS exposure. As shown in Supplementary Fig. 11g–j, mifepristone, but not spironolactone, effectively reversed FS-induced avoidance in the OFT (Supplementary Fig. 11g, h) and EPMT (Supplementary Fig. 11i, j), suggesting GR mediates FS-induced avoidance behavior.

We next asked whether pharmacological interventions targeting CORT-GR signaling could reverse the heightened activity of BLA→vHPC PNs in 7d P-FS male mice. We found that both metyrapone and mifepristone pretreatment reduced the FS-induced increase in AP numbers (Supplementary Fig. 11k–m) and mEPSC frequency (Supplementary Fig. 11n–p).

To further delineate the role of CORT-GR signaling within the BLA in mediating FS-induced avoidance behavior and BLA→vHPC neuronal hyperactivation, we implanted cannulas into the BLA and injected retrograde AAV-hSyn-mCherry virus into the vHPC to label BLA→vHPC PNs. Mifepristone was locally administered into the BLA prior to FS exposure (Fig. 5c, d). As shown in Fig. 5e–h, local GR blockade with mifepristone effectively reversed FS-induced avoidance behavior in both OFT and EPMT. Additionally, mifepristone treatment also

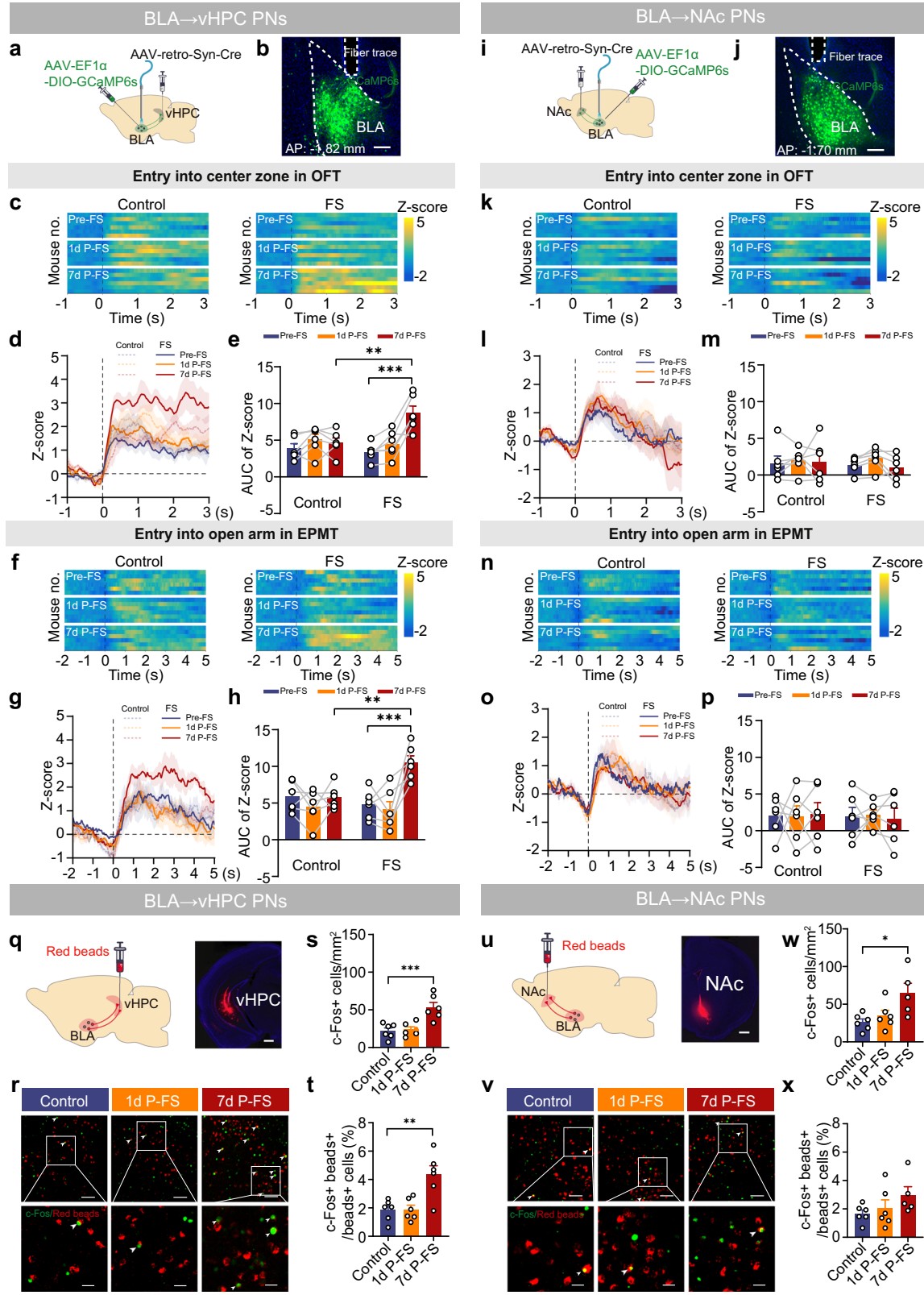

reduced contextual fear memory (Supplementary Fig. 12). Consistent with these behavioral findings, electrophysiological recordings demonstrated that mifepristone prevented the FS-induced increases in AP numbers and mEPSC frequency in BLA→vHPC PNs (Fig. 5i–o).

To determine whether these effects generalize to females, we repeated the same cannulation and pharmacological procedures in female mice. Similar to males, mifepristone pretreatment in

females reversed FS-induced avoidance behavior and contextual fear memory (Supplementary Figs. 13 and 14) and also prevented neuronal hyperactivation in BLA→vHPC PNs (Supplementary Fig. 15). Together, these findings indicate that CORT-GR signaling within the BLA drives FS-induced avoidance behavior through a sexually conserved mechanism involving hyperactivation of BLA→vHPC PNs.

**Fig. 2 | FS preferentially enhances the activity of BLA → vHPC PNs. a** Schematic of calcium signal recording in BLA→vHPC PNs. **b** Representative image showing GCaMP6s expression in BLA→vHPC PNs and the cannula implantation above the BLA, scale bar: 200 μm. (*n* = 12 mice). Heatmaps of calcium signal Z-score (**c**) and peri-event average traces (**d**) (Solid line: mean; shaded area: SEM) aligned to OFT center zone entry. **e** Quantification of AUC of calcium signals from (**d**) (*n* = 6 mice/group). Heatmaps of calcium signal Z-score (**f**) and peri-event average traces (**g**) (Solid line: mean; shaded area: SEM) aligned to EPMT open arm entry. **h** Quantification of AUC of calcium signals from (**g**) (*n* = 6 mice/group). **i–p** Same as in (**a–h**) except that the calcium signals were recorded from BLA→NAc PNs (*n* = 6 mice/group). **q** Schematic and representative images showing injection of red retrobeads into vHPC to label BLA→vHPC PNs. Scale bar: 500 μm. (*n* = 18 mice). **r** Representative images showing c-Fos expression (green) in BLA. The

arrowhead indicating c-Fos staining in BLA→vHPC PNs (red). Scale bar: 100 μm (top row); 30 μm (bottom row). **s, t** Quantification of c-Fos+ cells in the whole BLA (**s**) and in BLA→vHPC PNs (**t**) (*n* = 6 mice/group). **u** Schematic and representative images showing injection of red retrobeads into NAc to label BLA→NAc PNs. Scale bar: 500 μm. (*n* = 17 mice). **v** Representative images showing c-Fos expression (green) in BLA. The arrowhead indicating c-Fos staining in BLA→NAc PNs (red). Scale bar: 100 μm (top row); 30 μm (bottom row). **w, x** Quantification of c-Fos+ cells in the whole BLA (**w**) and in BLA→NAc PNs (**x**) (*n* = 5–6 mice). Data were analyzed using two-way RM ANOVA (**e, h, m, p**) and one-way ANOVA (**s, t, w, x**), followed by Bonferroni-corrected, two-tailed pairwise comparisons for (**e, h, s, t, w**). Data are presented as mean ± SEM. *$p < 0.05$, **$p < 0.01$, ***$p < 0.001$. See Supplementary Data 1 for full statistical information. Source data are provided as a Source data file.

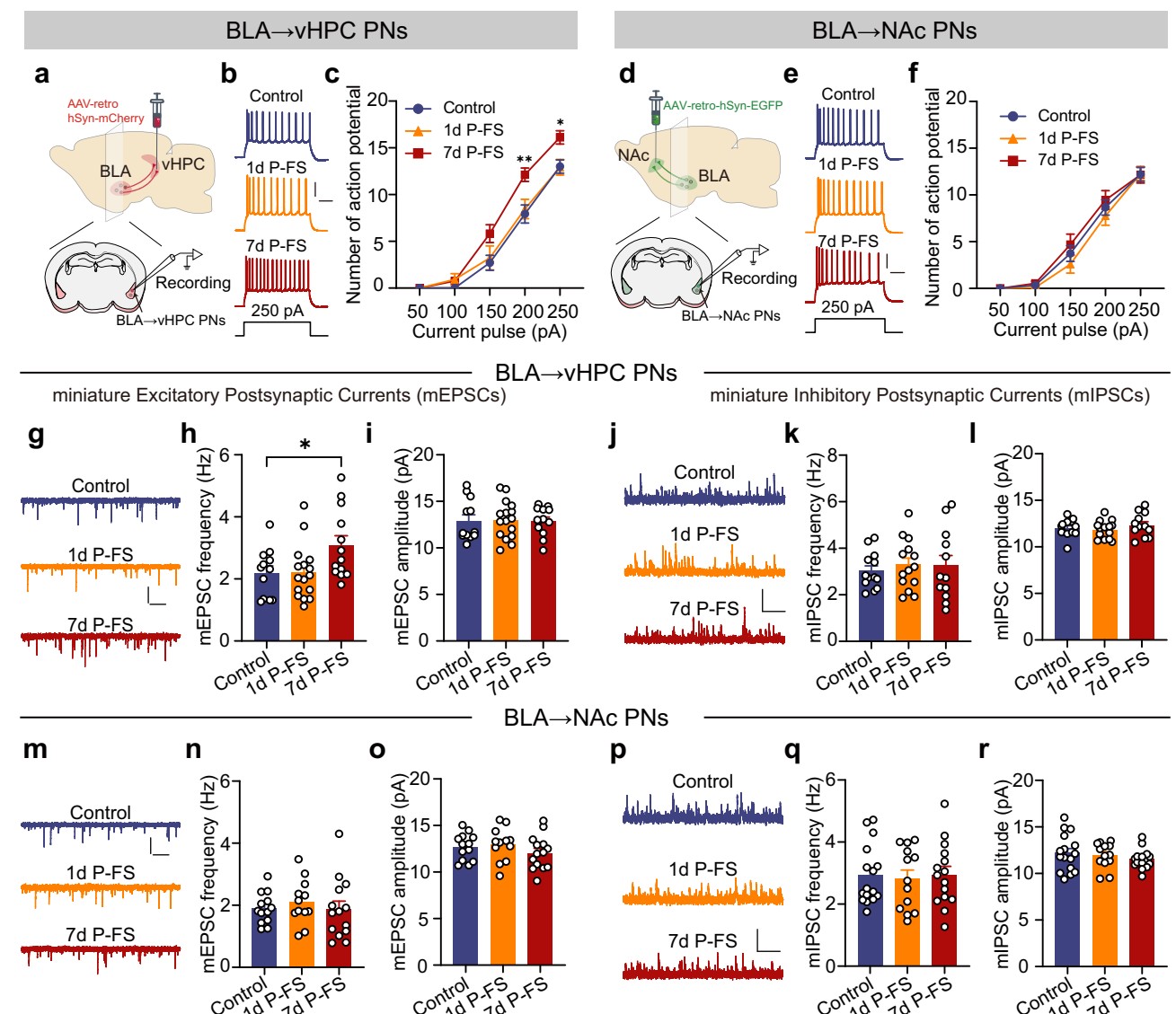

**Fig. 3 | FS preferentially increases the intrinsic excitability of BLA → vHPC PNs and augments the excitatory inputs they receive. a** Schematic showing the recording in BLA→vHPC PNs. **b** Representative firing traces of BLA→vHPC PNs (250 pA current injection). Scale bar: 200 ms, 30 mV. **c** Action potential number in response to varying current injection (*n* = 13–17 neurons/4 mice). **d–f** Same as in (**a–c**) except that the recordings were made on BLA→NAc PNs (*n* = 12–15 neurons/3–4 mice). **g** Representative mEPSC traces of BLA→vHPC PNs. Scale bar: 1 s, 20 pA. **h, i** Averaged mEPSC frequency (**h**) and amplitude (**i**) of BLA→vHPC PNs. (*n* = 12–16 neurons/4 mice) **j** Representative mIPSCs traces of BLA→vHPC PNs.

Scale bar:1 s, 20 pA. **k, l** Averaged mIPSC frequency (**k**) and amplitude (**l**) of BLA→vHPC PNs. (*n* = 12–14 neurons/4 mice) **m–o** Same as in (**g–i**) except that the data were from BLA→NAc PNs (*n* = 12–14 neurons/4 mice). **p–r** Same as in (**j–l**) except that the data were from BLA→NAc PNs (*n* = 13–16 neurons/4 mice). Data were analyzed using two-way RM ANOVA (**c, f**), one-way ANOVA (**h, k, l, n, o, r**) and Kruskal–Wallis test (**i, q**), followed by Bonferroni-corrected, two-tailed pairwise comparisons for (**c, h**). Data are presented as mean ± SEM. *$p < 0.05$, **$p < 0.01$. See Supplementary Data 1 for full statistical information. Source data are provided as a Source data file.

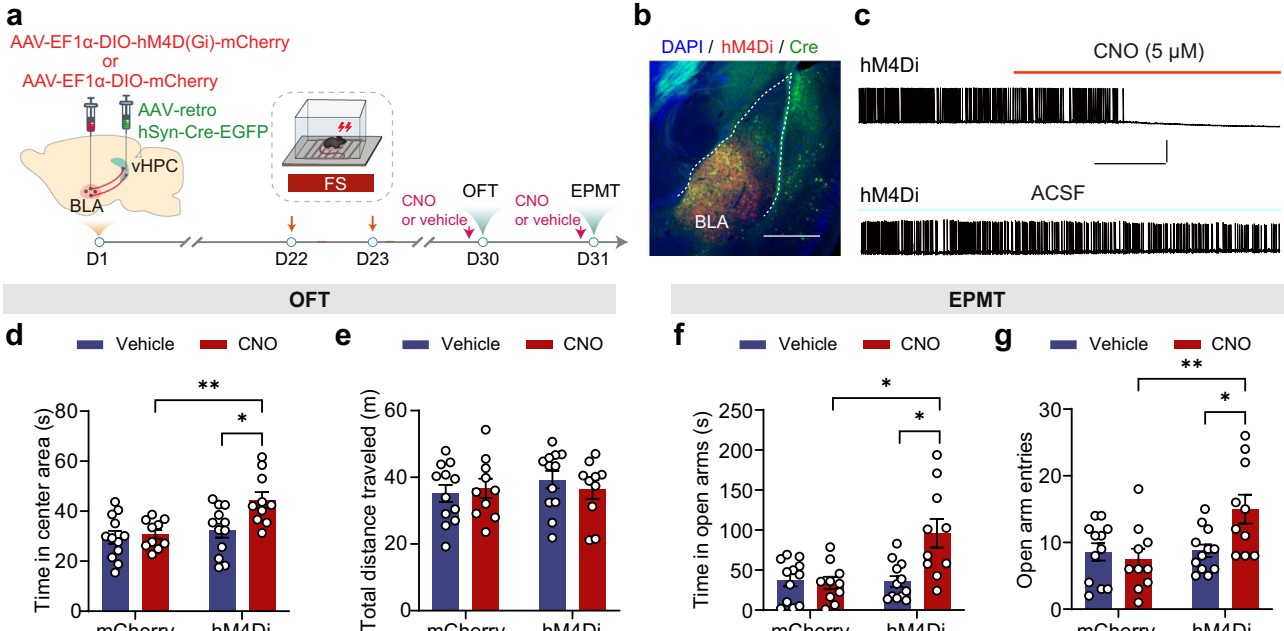

**Fig. 4 | Chemogenetic inhibition of BLA→vHPC PNs alleviates FS-induced PTSD-like avoidance behavior in mice. a** Schematic of the experimental procedures. **b** Representative image showing hM4D (Gi) expression in BLA→vHPC PNs. Scale bar: 500 µm. ($n$ = 22 mice). **c** Representative traces showing CNO-induced inhibition of the firing of BLA→vHPC PNs upon depolarizing current injection. Scale bar: 60 s, 25 mV. Time in center area (**d**) and total distance traveled (**e**) in OFT ($n$ = 10–12 mice). Time in open arms (**f**) and open arm entries (**g**) in EPMT ($n$ = 10–12 mice). Data were analyzed using two-way ANOVA (**d**, **g**) and Kruskal–Wallis test (**e**, **f**), followed by Bonferroni-corrected, two-tailed pairwise comparisons for (**d**, **f**, **g**). Data are presented as mean ± SEM. *$p$ < 0.05, **$p$ < 0.01. See Supplementary Data 1 for full statistical information. Source data are provided as a Source data file.

## FS upregulates *Sgk1* mRNA expression in BLA → vHPC PNs

We next explored how the CORT-GR signaling would account for the delayed PTSD-like avoidance behavior and hyperactivation of BLA→vHPC PNs by FS. As the surge of CORT triggered by FS is temporary, we hypothesized that the delayed effects of FS might involve prolonged changes in GR and/or related downstream proteins. To investigate this, we isolated the fluorescently labeled BLA→vHPC and BLA→NAc PNs from both unstressed and 7d P-FS mice, and quantified mRNA expression of genes encoding these proteins (Fig. 6a–c and Supplementary Fig. 16a–c). The results revealed that FS did not affect the expression of mRNA of GR (*Nr3c1*) and its MR counterpart (*Nr3c2*) in either PN subset (Fig. 6c and Supplementary Fig. 16c). However, FS significantly increased serum- and glucocorticoid-regulated kinase 1 (*Sgk1*) mRNA expression in BLA→vHPC PNs, with no such change in BLA→NAc PNs (Fig. 6c and Supplementary Fig. 16c). Expression of other GR-related genes, including those encoding GR co-chaperones (*Hsp90*, *Fkbp5*) and downstream proteins (*Nfkbia*, *Klf9*, *Smo*, *Per1*), remained unaltered in both groups of neurons (Fig. 6c and Supplementary Fig. 16c).

To further confirm this finding, we employed RNAscope to assess *Sgk1* transcriptional changes by FS. The result revealed a significantly higher *Sgk1* expression in BLA→vHPC PNs from 7d P-FS mice compared to their unstressed counterparts (Fig. 6d, e), with no changes in BLA→NAc PNs (Supplementary Fig. 16d, e). Therefore, within the spectrum of genes associated with GR signaling, FS appeared to preferentially upregulate *Sgk1* expression in BLA→vHPC PNs.

## Loss-of-function of Sgk1 in BLA → vHPC PNs prevents FS-induced avoidance behavior and neuronal activation

To determine whether the heightened Sgk1 expression in BLA→vHPC PNs contributes to the delayed avoidance behavior and neuronal hyperactivation, we employed a dominant negative form of Sgk1 (referred to as Sgk1-DN) to inhibit its function in BLA→vHPC PNs (Fig. 6f, g)[50,51]. The expression of Sgk1-DN in BLA was confirmed by RT-

qPCR analysis and immunofluorescence staining (Fig. 6h–j). To validate the functional efficacy of Sgk1-DN, we measured the phosphorylation of cAMP response element-binding protein at Ser133 (pCREB), a direct Sgk1 target[50], and found it was significantly reduced by Sgk1-DN (Supplementary Fig. 17). The behavioral assays showed that overexpressing Sgk1-DN in BLA→vHPC PNs hindered the delayed avoidance behavior in 7d P-FS mice (Fig. 6k–o).

We then proceeded to investigate whether Sgk1-DN could also mitigate the heightened activation of BLA→vHPC PNs following FS exposure. Fiber photometry recordings revealed that the increase in calcium transients observed in mCherry control mice at 7 days post-FS (Fig. 6p–u), which coincided with a significant increase in avoidance behavior (Supplementary Fig. 18), was effectively blocked by Sgk1-DN overexpression. Electrophysiological recordings further confirmed that Sgk1-DN expression suppressed the FS-induced increases in both AP numbers and mEPSCs frequency in BLA→vHPC PNs (Fig. 6v–z).

Collectively, these results strongly suggest that the increased Sgk1 expression is essential in mediating the delayed increase of avoidance behavior and hyperactivation of BLA→vHPC PNs by FS.

## Sgk1 overexpression in BLA → vHPC PNs confers stress vulnerability

We then examined whether overexpressing Sgk1 in BLA→vHPC PNs (referred to as Sgk1-OE) alone could induce changes in avoidance behavior. After confirming Sgk1 expression efficacy (Fig. 7a–d), we evaluated its impact on avoidance behavior (Supplementary Fig. 19a, b). Somewhat surprisingly, Sgk1-OE mice did not differ from mCherry-expressing controls in both OFT and EPMT (Supplementary Fig. 19c–f), indicating that Sgk1 overexpression in BLA→vHPC PNs alone is insufficient to alter avoidance behavior.

This lack of effect prompted us to hypothesize that Sgk1's role in FS-induced PTSD-like avoidance behavior might be linked to stress susceptibility. To investigate this possibility, we utilized a subthreshold FS (Sub-FS) approach that did not cause avoidance behavior in naive mice

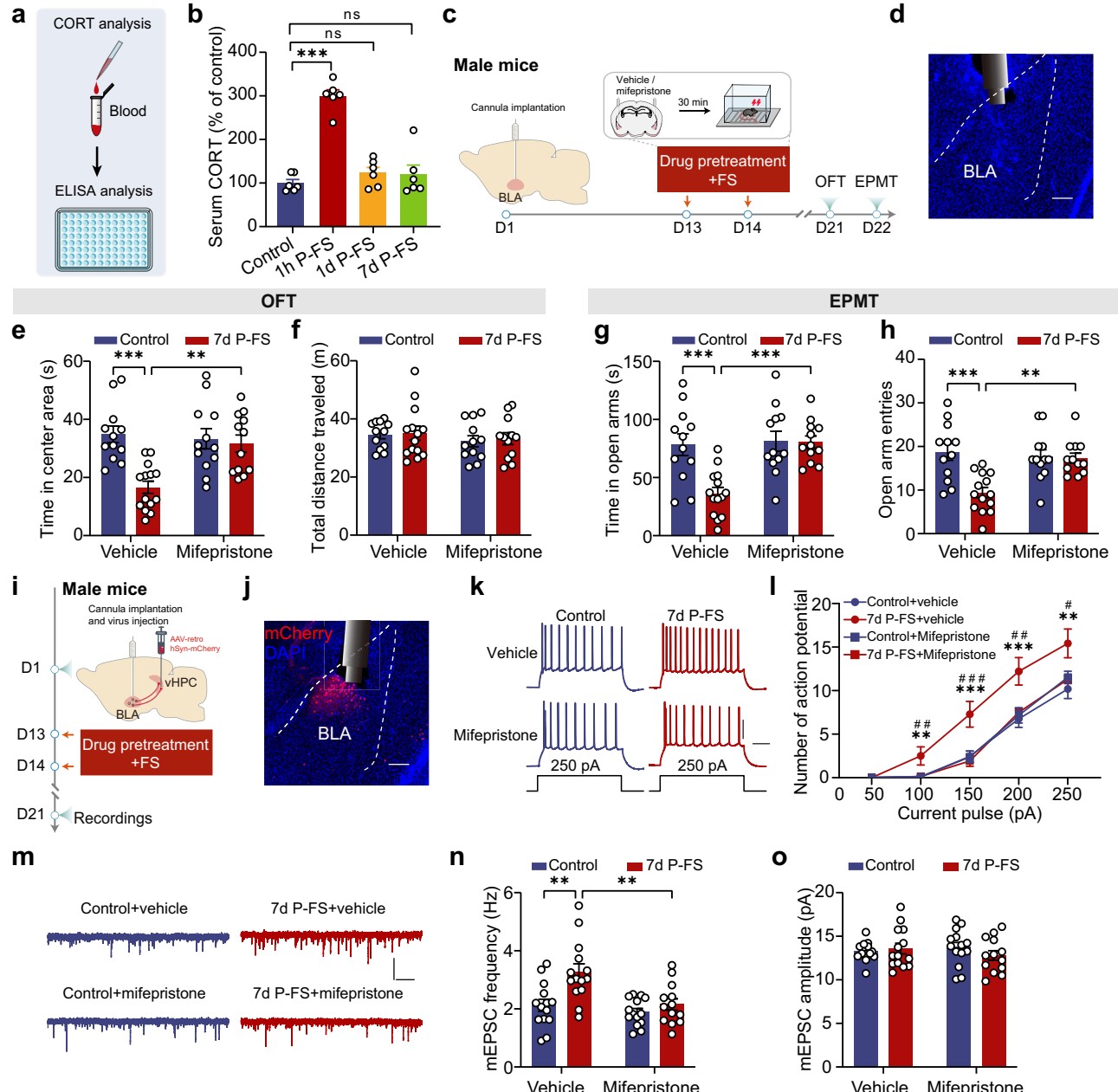

**Fig. 5 | Blocking of CORT-GR signaling within BLA mitigates FS-induced increase in avoidance behavior and BLA→vHPC PNs overactivation in mice.** a Schematic of CORT analysis. b Summary plots of normalized serum CORT levels. Data were normalized to unstressed control group (*n* = 6 mice/group). c Schematic of experiment procedures. d Representative images showing the cannula implantation in BLA. Time in center area (e) and total distance traveled (f) in OFT (*n* = 12–14 mice). Time in open arms (g) and open arm entries (h) in EPMT (*n* = 12–14 mice). i Schematic of experiment procedures. j Representative images showing mCherry expression in BLA→vHPC PNs and the cannula implantation in BLA. (*n* = 16 mice). k Representative firing traces (250 pA current injection). Scale bar: 200 ms, 30 mV.

l Action potential number in response to varying current injection (*n* = 13–16 neurons/4 mice). m Representative mEPSC traces. Scale bar: 1 s, 20 pA. Averaged mEPSC frequency (n) and amplitude (o) of BLA→vHPC PNs (*n* = 13–14 neurons/4 mice). Data were analyzed using one-way ANOVA (b), Kruskal–Wallis test (e, h), two-way ANOVA (f, g, n, o), and three-way RM ANOVA (l), followed by Bonferroni-corrected, two-tailed pairwise comparisons for (b, e, g, h, l, n). Data are presented as mean ± SEM. ns, not significant. \*\**p* < 0.01, \*\*\**p* < 0.001, #*p* < 0.05, ##*p* < 0.01, ###*p* < 0.001. See Supplementary Data 1 for full statistical information. Source data are provided as a Source data file.

(Fig. 7e). Behavioral assessments revealed that Sub-FS did not result in any noticeable alterations in avoidance behavior in mCherry-expressing mice (Fig. 7f–j). Conversely, Sub-FS led to an increase in avoidance behavior in Sgk1-OE mice (Fig. 7g–j). Thus, the increased Sgk1 expression in BLA→vHPC PNs may promote mice to develop avoidance behavior under stress conditions through heightening their susceptibility to stress.

To explore the effect of Sgk1 overexpression in BLA→vHPC PNs on neuronal activity under the Sub-FS condition, we co-expressed Sgk1 and

GCaMP6s in BLA→vHPC PNs and monitored the calcium signaling (Fig. 7k–p). Sub-FS exposure significantly increased calcium signal changes in Sgk1-OE mice but not in mCherry controls (Fig. 7k–p), coinciding with the increased avoidance behavior specifically in the Sgk1-OE group (Supplementary Fig. 20). Consequently, Sgk1-OE mice subjected to Sub-FS exhibited higher calcium signals than their mCherry counterparts under the same conditions (Fig. 7k–p). Likewise, Sub-FS induced notable increases in the intrinsic excitability and excitatory

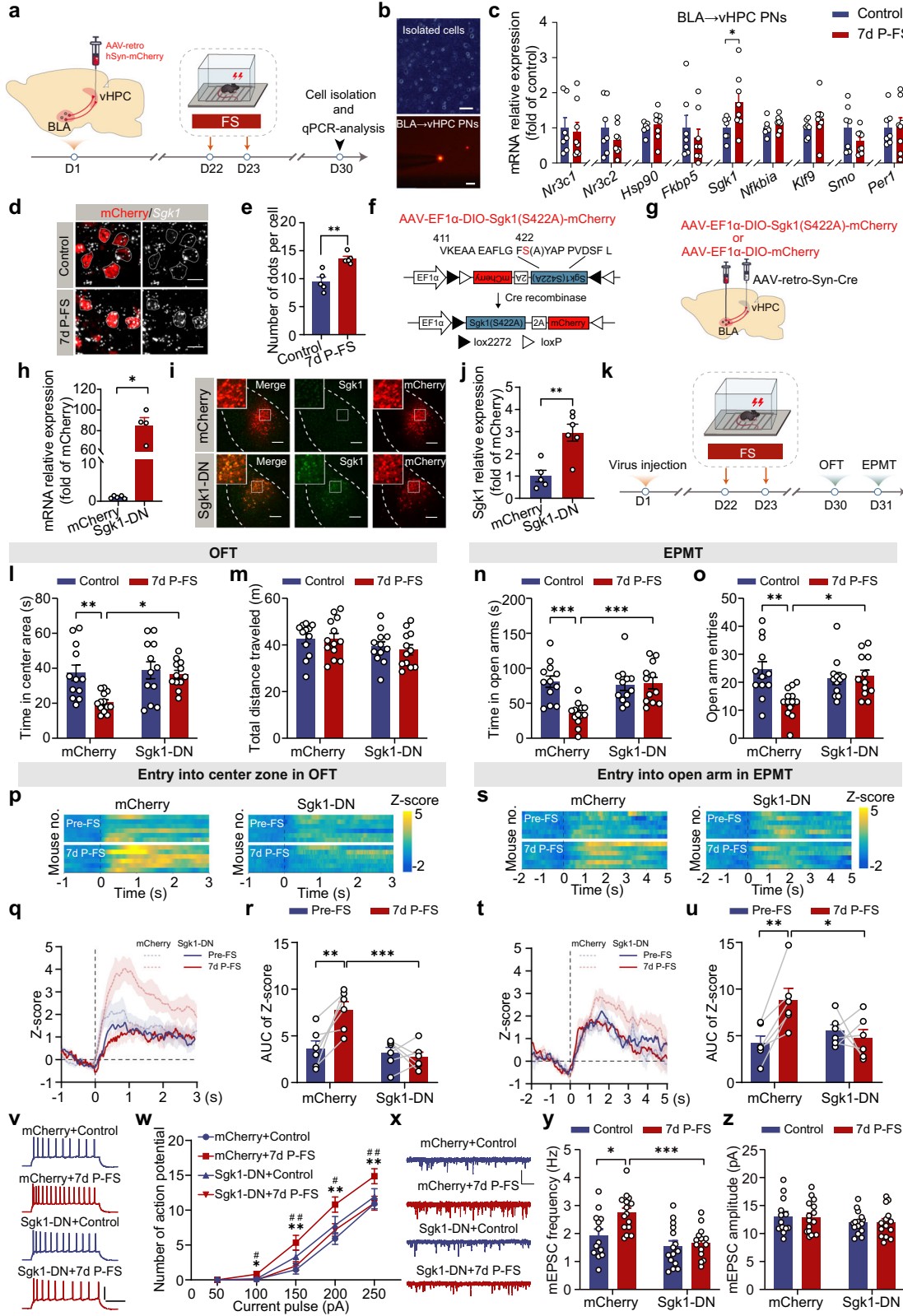

synaptic transmission of BLA→vHPC PNs in Sgk1-OE mice, whereas no such effects were observed in mCherry control mice (Fig. 7q–u).

Cumulatively, these findings highlight that increased Sgk1 expression heightens the susceptibility of BLA→vHPC PNs to stress exposure, thereby increasing the likelihood of individuals developing avoidance behavior.

## Discussion

In this study, using an FS-induced PTSD mouse model, we identified delayed activation of a cluster of BLA neurons projecting to the ventral hippocampus, which correlated with the PTSD-like avoidance behavior observable 7 days post-FS. This delayed activation is driven by increased expression of Sgk1 in these BLA→vHPC PNs (Fig. 8). Loss-of-

**Fig. 6 | *Sgk1* upregulation in BLA→vHPC PNs is critical for FS-induced avoidance behavior in mice and overactivation of these PNs. a** Schematic of experiment procedures. **b** Representative images of single-cell suspension (Upper) and the captured BLA→vHPC PN (Lower). Scale bar: 100 μm (Upper), 50 μm (Lower). (*n* = 15 mice). **c** RT-qPCR analysis of genes involved in CORT signaling in BLA→vHPC PNs (*n* = 7–8 mice). **d** Representative images showing mCherry expression and RNAscope for *Sgk1* transcripts (white dashed lines outline BLA→vHPC PNs). Scale bar: 50 μm. **e** Averaged *Sgk1* transcript dots per cell (*n* = 5 mice/group). AAV vector design (**f**) and schematic of Sgk1(S422A) expression in BLA→vHPC PNs (**g**). **h** RT-qPCR analysis of *Sgk1* expression in BLA (*n* = 4–5 mice). Representative images (**i**) and quantification of Sgk1 expression in mCherry-labeled BLA→vHPC PNs (**j**) (*n* = 5–6 mice). Scale bar: 200 μm. **k** Experiment timeline. Time in center area (**l**) and total distance traveled (**m**) in OFT (*n* = 12–13 mice). Time in open arms (**n**) and open-arm entries (**o**) in EPMT (*n* = 12–13 mice). Heatmaps of calcium signal Z-score and peri-event average traces (Solid line: mean; shaded area: SEM) aligned to OFT center zone entry (**p**, **q**) and EPMT open arm entry (**s**, **t**). Mice expressed GCaMP6s and either mCherry (control) or Sgk1(S422A)-mCherry (Sgk1-DN) in BLA→vHPC PNs; all underwent FS. **r**, **u** Quantification of AUC of calcium signals from (**q**) and (**t**), respectively (*n* = 6 mice/group). **v** Representative firing traces (250 pA current injection). Scale bar: 200 ms, 30 mV. **w** Action potential number in response to varying current injection (*n* = 12–15 neurons/3–4 mice). **x** Representative mEPSC traces. Scale bar: 1 s, 20 pA. **y**, **z** Averaged mEPSC frequency (**y**) and amplitude (**z**) of BLA→vHPC PNs. (*n* = 13–15 neurons/4 mice). Data were analyzed using two-tailed unpaired *t* test (**c**, **e**, **j**), two-tailed Mann-Whitney *U* test (**h**), Kruskal–Wallis test (**l**, **o**), two-way ANOVA (**m**, **n**, **y**, **z**), two-way RM ANOVA (**r**, **u**), and three-way RM ANOVA (**w**), followed by Bonferroni-corrected, two-tailed pairwise comparisons for (**l**, **n**, **o**, **r**, **u**, **w**, **y**). Data are presented as mean ± SEM. \**p* < 0.05, \*\**p* < 0.01, \*\*\**p* < 0.001, #*p* < 0.05, ##*p* < 0.01. See Supplementary Data 1 for full statistical information. Source data are provided as a Source data file.

function of Sgk1 in BLA→vHPC PNs effectively thwarted their overactivation and the development of avoidance behavior, while its overexpression increased stress vulnerability.

Mounting clinical and preclinical studies have consistently demonstrated maladaptive amygdala responses to extreme or prolonged traumatic stimuli, characterized by structural remodeling and excessive neuronal activation[37,52,53]. Consequently, amygdala hyperactivation has been identified as a prominent neurobiological hallmark of PTSD, with its activation positively correlated with PTSD symptom severity and self-reported anxiety[12,54,55]. Some therapies for PTSD appear effective by modulating amygdala activity[20,56], a phenomenon also observed in rodent models[57,58]. Nonetheless, the precise temporal pattern regarding how amygdala neurons react to PTSD-eliciting stimuli remains elusive. Our study found no immediate changes in excitatory inputs or firings of BLA neurons 1 day post-FS, echoing previous findings of delayed anxiety-like behavior, increased dendritic spine density, and enhanced synaptic transmission in the BLA, observed 10 but not 1 day post-stress[26]. Since BLA is highly responsive to adverse stimuli, the negligible changes seen at 1 day post-FS may hint that the BLA neurons in mice were capable of swiftly regulating their activity back to pre-stress level shortly after FS cessation. The swift regulation was also seen in human studies. For example, by using PET scans to compare brain activity just before (anticipation stress) and immediately after (control) public speaking, Tillfors et al. observed increased activation in the amygdala during anticipation but not control phase[59]. These observations support the hypothesis that delayed amygdala activation may underlie the gradual onset of PTSD symptoms.

Abnormal amygdala activation is crucial in forming intrusive traumatic memories in PTSD patients. Functional magnetic resonance imaging (fMRI) studies show that individuals with heightened amygdala activity during trauma-analog events (in contrast to neutral events) experience more memory intrusions afterward[60,61]. Amygdala activation also strongly influences the consolidation of arousal-related memories, with higher activation during emotional events linked to better recall later[60]. Animal studies further highlight the amygdala's role in storing traumatic memories, with fear memory engram cells active during both encoding and retrieval[21–23,62]. While the amygdala's role in intrusive memories is well-documented, its involvement in other core symptoms of PTSD, like avoidance behavior, has received comparatively less exploration. Our study reveals a temporal link between the delayed activation of BLA neurons post-stress and the onset of PTSD-like avoidance behavior in FS mice. Specifically, FS exposure led to long-lasting avoidance behavior closely linked to the BLA neuron activation pattern. Notably, chemogenetic inhibition of BLA activity markedly mitigated FS-induced avoidance behavior. These findings thus expand our understanding of the amygdala's role in PTSD-related avoidance.

The BLA PNs display significant heterogeneity in their connectivity and responses to stress[63]. For instance, recent studies show

that amygdala neurons projecting to the anterodorsal bed nucleus of the stria terminalis (adBNST) underlie pathological anxiety in PTSD, whereas those projecting to the lateral central amygdala (CeL) are linked to fear response in PTSD[64]. On the other hand, fear conditioning paradigms induce divergent synaptic changes in BLA neurons projecting to NAc versus the centromedial amygdala[65]. We found that the delayed augmentation of excitatory inputs to BLA PNs and their delayed activation varied among individual neurons with distinct projection targets. Specifically, while the BLA→vHPC PNs exhibit delayed activation accompanied by an increase of excitatory inputs, their BLA→NAc counterparts remain unaffected. The selective activation of BLA→vHPC PNs plays an important role in the delayed onset of PTSD-like avoidance behavior following FS exposure. Chemogenetic inhibition of the BLA→vHPC PNs significantly mitigates the delayed avoidance behavior induced by FS. Given that BLA→vHPC PNs have been implicated in maladaptive responses and contribute to emotional and behavioral dysregulation under various stress conditions[37,38], it is plausible to consider that modulating this neuronal subpopulation could hold promise for preventing and treating stress-related mental disorders.

Dysregulation of the HPA axis plays a pivotal role in the pathogenesis of PTSD and is closely tied to abnormal amygdala activation[66,67]. Our observations revealed a transient and marked surge in plasma corticosterone (cortisol in humans) levels one hour after FS exposure, swiftly returning to baseline levels. Similarly, a clinical investigation demonstrated that PTSD patients, particularly survivors of the earthquake, displayed elevated cortisol levels one month post-earthquake compared to individuals without earthquake exposure. However, no divergence in cortisol levels was noted between the two groups five months after the traumatic event[68]. The transient surge of CORT appears crucial for PTSD development, as evidenced by blocking CORT synthesis before FS exposure effectively mitigating the delayed increases in avoidance behavior and the activity of BLA→vHPC PNs. Consistent with these results, earlier studies reported that acute administration of high-dose CORT was sufficient to induce delayed anxiety-like behavior in rats, alongside delayed structural remodeling of neurons in the amygdala and prefrontal cortex[46,47]. Collectively, these findings suggest that the promoted elevation in CORT secretion is both sufficient and indispensable for the delayed onset of PTSD-like avoidance behavior and delayed BLA activation.

The critical role of GR signaling in mediating the delayed increase in avoidance behavior and projection-specific adaptation of BLA PNs is unlikely due to the alteration of GR expression itself. Using RNAscope and qPCR analysis, we failed to detect substantial changes in GR mRNA expression in both BLA→vHPC and BLA→NAc PNs following FS exposure. Instead, we found that Sgk1, a downstream target of GR signaling, may account for the impact of GR signaling on the effects of FS on mice's behavior and BLA neuronal activity. Sgk1 has been demonstrated to play a pivotal role in modulating neuronal activity,

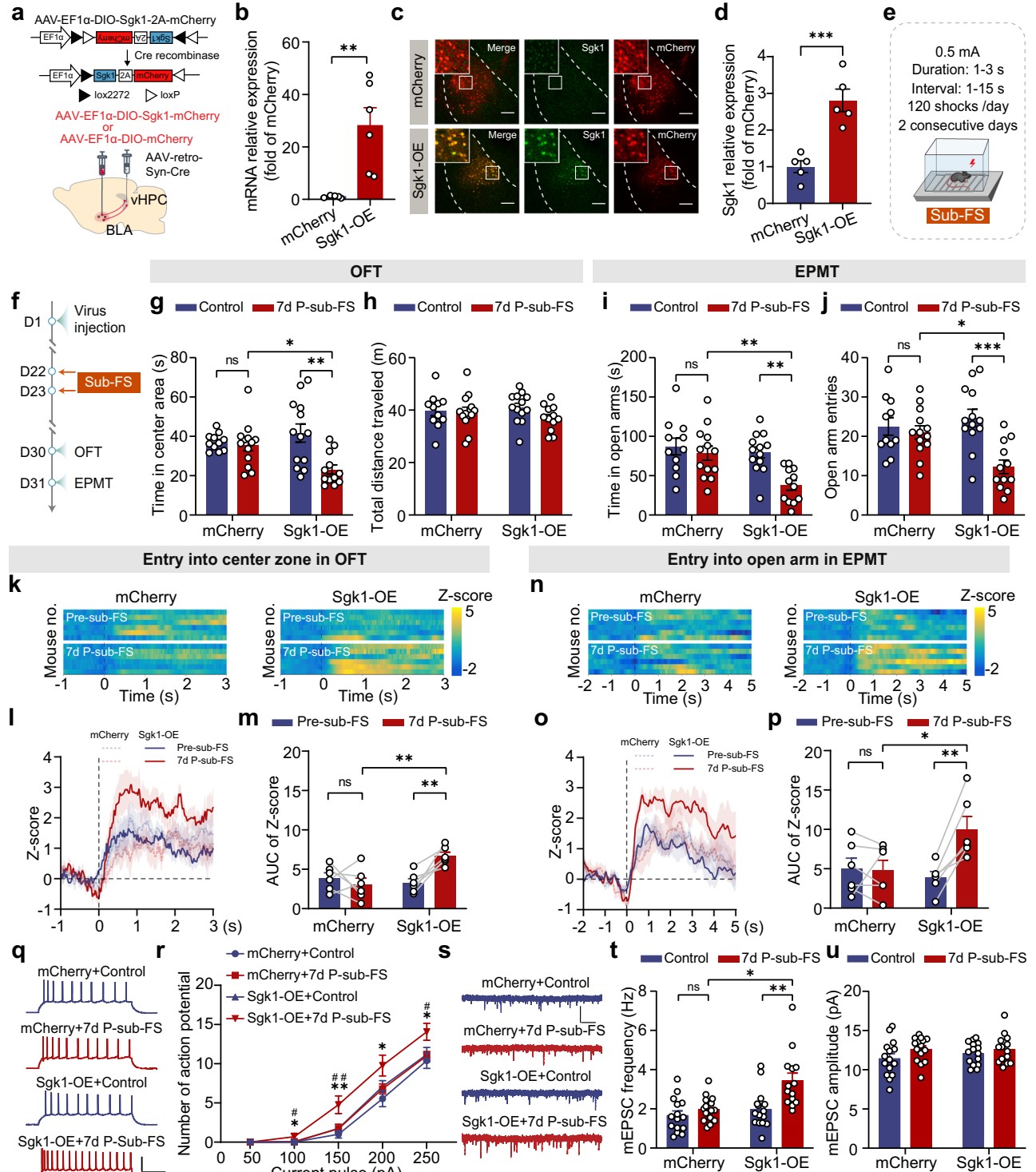

**Fig. 7 | *Sgk1* overexpression in BLA→vHPC PNs increases the avoidance behavior in mice upon Sub-FS exposure and the activity of BLA→vHPC PNs.**
**a** Schematic of using Cre-on viral strategy to express *Sgk1* in BLA→vHPC PNs. **b** RT-qPCR validation of the efficiency of *Sgk1* expression in BLA (*n* = 5–6 mice). Representative images (**c**) and quantification of Sgk1 expression in mCherry-labeled BLA→vHPC PNs (**d**) (*n* = 5 mice/group). **e** Schematic of Sub-FS procedure. **f** Experiment timeline. Time in center area (**g**) and total distance traveled (**h**) in OFT (*n* = 11–13 mice). Time in open arms (**i**) and open arm entries (**j**) in EPMT (*n* = 11–13 mice). Heatmaps of calcium signal Z-score and peri-event average traces (Solid line: mean; shaded area: SEM) aligned to OFT center zone entry (**k**, **l**) and EPMT open arm entry (**n**, **o**). Mice expressed GCaMP6s and either mCherry (control) or Sgk1-mCherry (Sgk1-OE) in BLA→vHPC PNs; all underwent Sub-FS procedure.

**m**, **p** Quantification of AUC of calcium signals from (**l**) to (**o**), respectively (*n* = 6 mice/group). **q** Representative firing traces (250 pA current injection). Scale bar: 200 ms, 30 mV. **r** Action potential number in response to varying current injection (*n* = 12–14 neurons/3–4 mice). **s** Representative mEPSC traces. Scale bar: 1 s, 20 pA. Averaged mEPSC frequency (**t**) and amplitude (**u**) of BLA→vHPC PNs. (*n* = 14–15 neurons/4 mice). Data were analyzed using two-tailed unpaired *t* test (**b**, **d**), two-way ANOVA (**g**–**j**, **u**), Kruskal–Wallis test (**t**), two-way RM ANOVA (**m**, **p**) and three-way RM ANOVA (**r**), followed by Bonferroni-corrected, two-tailed pairwise comparisons for (**g**, **i**, **j**, **m**, **p**, **r**, **t**). Data are presented as mean ± SEM. ns, not significant. *$p < 0.05$, **$p < 0.01$, ***$p < 0.001$, #$p < 0.05$, ##$p < 0.01$. See Supplementary Data 1 for full statistical information. Source data are provided as a Source data file.

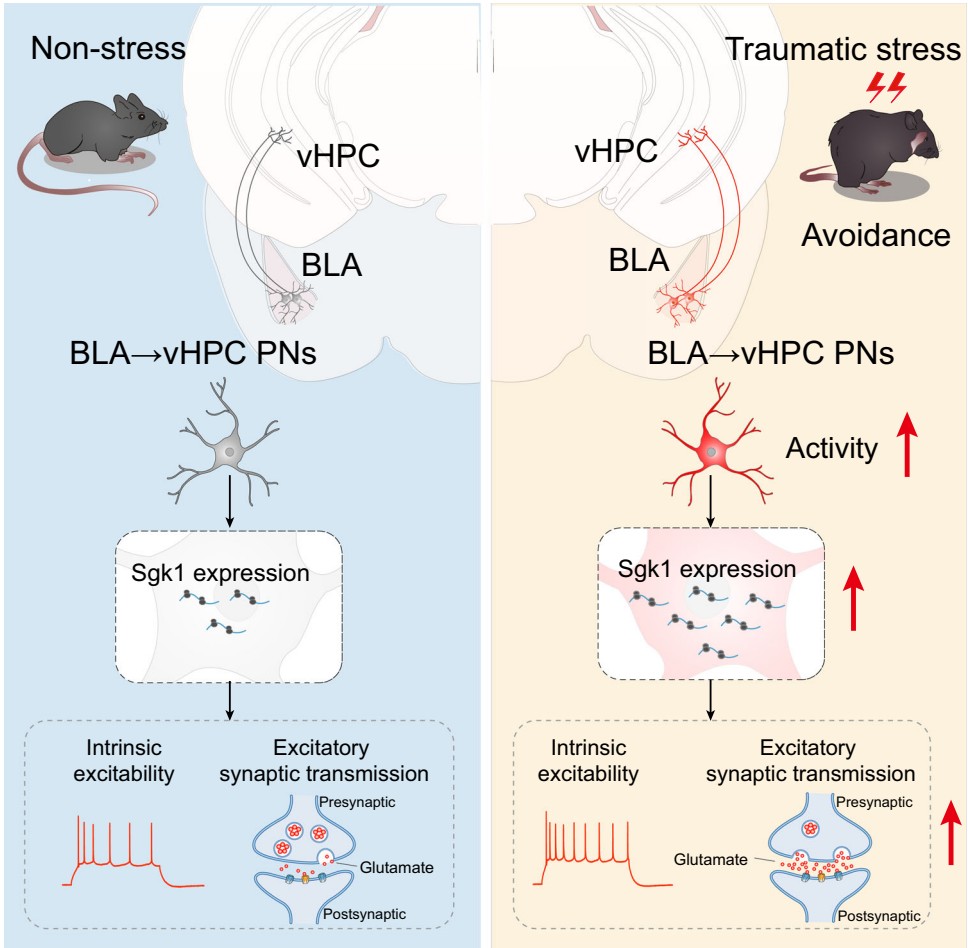

**Fig. 8 | Schematic illustration of the molecular and neural circuitry mechanisms underlying PTSD-like avoidance behavior induced by traumatic stress.** Traumatic stress selectively activates BLA neurons that project to the ventral hippocampus (BLA→vHPC PNs). This effect involves upregulation of *Sgk1* expression, leading to the increased intrinsic excitability and excitatory synaptic transmission, which contributes to the delayed onset of PTSD-like avoidance behavior.

encompassing synaptic plasticity and alterations in neuronal excitability[50,69–71]. Additionally, clinical data suggest that dysregulation of Sgk1 expression or function is closely linked to the development of psychiatric disorders such as depression and PTSD[50,70,72,73]. In our screening of the expression changes of a range of downstream target molecules associated with GR signaling, we observed significant upregulation of Sgk1 mRNA following FS exposure, without notable impacts on the expression of other molecules. Furthermore, this upregulation exhibits clear projection specificity within BLA PNs, predominantly manifesting in BLA→vHPC PNs rather than their BLA→NAc counterparts. Functionally, overexpression of a constitutively inactive form of Sgk1 in BLA→vHPC PNs effectively occluded FS-induced increase of avoidance behavior.

An intriguing question arising from our findings is whether Sgk1, acting as a downstream effector of GR, can in turn modulate GR signaling activity, potentially forming a feed-forward loop that amplifies the initial stress response. Although we did not directly measure GR transcriptional activity following Sgk1 loss or gain-of-function, it is plausible that the elevated Sgk1 expression may function to sustain or even augment the primary GR signal. This notion is supported by prior evidence demonstrating that Sgk1 facilitates GR phosphorylation, nuclear translocation, and stability[72,74]. Therefore, the delayed yet persistent behavioral and neuronal phenotypes we observed may not solely originate from Sgk1 upregulation but could be further reinforced through an Sgk1-dependent positive feedback mechanism that potentiates GR signaling efficacy. This model offers a plausible explanation for why Sgk1 overexpression alone failed to induce alterations in naïve mice but dramatically lowered the threshold for stress-induced pathogenesis: by pre-configuring a potent amplifier of GR signaling, even a subthreshold stressor becomes sufficient to provoke a fulminant pathological response. Conversely, loss-of-function of Sgk1 may disrupt this positive feedback loop, thereby promoting resilience by preventing the GR signaling from reaching the critical threshold necessary for maladaptive plasticity. Future studies employing direct profiling of GR transcriptional activity and its downstream targets upon Sgk1 manipulation will be crucial to substantiate this hypothesis.

We have also established a close correlation between the heightened Sgk1 expression in BLA→vHPC PNs and the augmented intrinsic excitability of this neuronal subset following FS. Notably, the overexpression of Sgk1 in BLA→vHPC neurons in naïve mice does not provoke any discernible alterations in avoidance behavior or neuronal activity. This indicates that the mere upregulation of Sgk1 alone does not suffice to trigger these changes. However, exposure to subthreshold stress significantly amplifies avoidance behavior in Sgk1-overexpressing mice, concomitant with heightened intrinsic excitability and glutamatergic synaptic transmission in BLA→vHPC PNs. These findings imply that Sgk1 may contribute to modulating the behavioral and neuronal activity changes induced by FS by enhancing an individual's susceptibility to stress.

Overall, our findings reveal that delayed activation of BLA→vHPC PNs, driven by Sgk1, contributes to PTSD-like avoidance behavior.

Nevertheless, several pivotal questions remain unanswered. For instance, how does FS trigger the increased Sgk1 expression within specific neuronal subsets in the BLA? Moreover, Sgk1 has been implicated in the regulation of neuronal morphology[50,75,76]. Considering our previous findings of chronic stress-induced circuit-specific structural remodeling of BLA PNs[38], it is also intriguing to explore whether Sgk1 mediates the differential structural remodeling in distinct BLA PN subtypes. Answering these questions could further clarify how stress affects Sgk1 to induce neuronal and behavioral dysfunction in PTSD.

## Methods

### Animals

C57BL/6J mice aged from 4 to 10 weeks were used for all experiments. Unless otherwise specified, all experiments utilized male mice. The mice were initially obtained from the Model Animal Research Center of Nanjing University (Nanjing, China) and bred in the animal facility of Nanchang University. The mice were housed in a room with a standard light/dark cycle (lights on from 7:00 A.M. to 7:00 P.M.), maintained at a controlled temperature of 21–25 °C and humidity of 40–60%. Mice were group-housed (3–5 per cage) with ad libitum access to food and water. All experimental procedures were under the National Institutes of Health guidelines and were approved by the Institutional Animal Care and Use Committee of Nanchang University (NCULAE-20221228059). Virus injection and cannula implantation were performed on mice at 4–5 weeks of age. Beads were injected at 6 weeks of age. Behavioral tests, histological analysis, slice electrophysiological recordings, and fiber photometry were conducted around 10 weeks of age.

### Footshock stress exposure

For footshock stress exposure, mice were subjected to footshock stress for one or two consecutive days, depending on the experimental paradigm. The mice were placed in a Plexiglas box (MED ASSOCIATES INC., Fairfax, VT) equipped with stainless steel conducting strips at the bottom for 1 h. During this time, mice received 360 scrambled footshocks (0.3 mA/0.5 mA) of 1–3 s duration, delivered at random intervals ranging from 1 to 15 s to minimize predictability and avoid habituation to the shocks. Unstressed control mice were transferred to the test room for 1 h, but did not receive any shocks. After stress, all mice were returned to their home cage.

For subthreshold footshock stress exposure, mice underwent a 20-minute stress procedure each day over two consecutive days. This involved 120 randomized footshocks (0.5 mA), delivered with the same duration and intervals as the footshock stress described above.

### Open field test

The open field consisted of a square chamber (50 × 50 cm), with a central area forming a square with a size of 25 × 25 cm. After a 30-min habituation period in the test room with red light, mice were placed in the center area of the chamber and allowed to explore for 10 min. Their locomotion was tracked using a video-tracking system (Med Associates Inc., Fairfax, VT). The time mice spent in the center area and the total distance they traveled were analyzed using ANY-maze software (Stoelting Co., Wood Dale, USA).

### Elevated plus maze test

The elevated plus maze consists of two open arms (30 × 6 cm), two closed arms (30 × 6 cm), and a square central hub area (6 × 6 cm) which is elevated 74 cm from the floor. Following a 30-min habituation period in the test room with red light, mice were placed in the hub area and allowed to explore for 10 min. Their locomotion was monitored by the video-tracking system (Med Associates Inc., Fairfax, VT). The time mice spent in the open arms and their entries into the open arms were analyzed by the ANY-maze software (Stoelting Co., Wood Dale, USA).

### Contextual fear conditioning test

Following a 2-day footshock (FS) conditioning paradigm, fear memory was assessed by re-exposing mice to the same shock chamber for 5 minutes at 1 or 7 days post-FS. Freezing behavior (defined as complete immobility except for respiration) was scored automatically using a video-based system (Video Freeze, Med Associates Inc., Fairfax, VT), with a freezing threshold set at a motion index of <10 in the Video Freeze software. The percentage of time spent freezing was calculated by the software as (cumulative freezing duration/300 s) × 100% and used for statistical analysis. The chambers were cleaned with 70% ethanol between trials.

### Stereotaxic surgery and injections

For viral or red Retrobeads (Lumafluor, Inc., Durham, NC) injection, mice were anesthetized with 2% pentobarbital sodium and placed in the stereotaxic frame (RWD Life Science Co., Ltd, Shenzhen, China). The viruses and beads are injected using a glass micropipette mounted on a 10-μL Hamilton Microlitre syringe (Hamilton Co., Ltd, Reno, USA) and connected to an infusion pump (QSI; Stoelting, Wood Dale, USA). After injection, the glass micropipette was left in place for an additional 10 min[37].

For fiber-photometry recording experiments, mice were unilaterally injected with AAV2/9-CaMKIIα-GCaMP6s (200 nL, Obio Technology, Shanghai, China) into the BLA (anterior/posterior, −1.28 mm; medial/lateral, ±3.08 mm; dorsal/ventral, −5.08 mm), or unilaterally injected with AAV2/9-EF1α-DIO-GCaMP6s (200 nL, Obio Technology, Shanghai, China) into the BLA and the AAV2/retro-Syn-Cre (200 nL, Obio Technology, Shanghai, China) into the ipsilateral NAc (bregma coordinates: anterior/posterior, −1.42 mm; medial/lateral, ±0.85 mm; dorsal/ventral, −4.7 mm) or vHPC (bregma coordinates: anterior/posterior, −3.16 mm; medial/lateral, ±3.4 mm; dorsal/ventral, −3.8 mm), or bilaterally injected with AAV2/9-EF1α-DIO-GCaMP6s mixed with AAV2/9-EF1α-DIO-Sgk1(S422A)-mCherry (Obio Technology, Shanghai, China) or AAV2/9-EF1α-DIO-Sgk1-mCherry (Obio Technology, Shanghai, China) or AAV2/9-EF1α-DIO-mCherry (Obio Technology, Shanghai, China) (200 nL) into the BLA and AAV2/retro-Syn-Cre (200 nL) into the bilateral vHPC. After the viral injection, a 200-μm diameter ceramic fiber-optic cannula (Inper Tech, Zhejiang, China) was implanted into the BLA (dorsal/ventral = −5 mm).

For labeling BLA→vHPC PNs and BLA→NAc PNs in c-Fos staining experiments, red Retrobeads (300 nL) were injected into the bilateral vHPC or bilateral NAc.

For labeling BLA→vHPC PNs and BLA→NAc PNs in electrophysiological recording, real-time quantitative PCR (RT-qPCR) and RNAscope experiments, AAV2/retro-hSyn-mCherry (200 nL, Obio Technology, Shanghai, China) and AAV2/retro-hSyn-EGFP (200 nL, Obio Technology, Shanghai, China) were injected into the bilateral vHPC and bilateral NAc, respectively.

For chemogenetic experiments, the AAV2/9-EF1α-DIO-hM4D(Gi)-mCherry (200 nL, Obio Technology, Shanghai, China) or AAV2/9-EF1α-mCherry (200 nL, Obio Technology, Shanghai, China) was injected into the bilateral BLA, and the AAV2/retro-hSyn-Cre-EGFP (200 nL, Obio Technology, Shanghai, China) was injected into the bilateral vHPC.

For BLA microinjection experiments, retrograde AAV2/retro-hSyn-mCherry (200 nL; Obio Technology, Shanghai, China) was injected into the bilateral vHPC to retrogradely label BLA→vHPC PNs. Following viral injection, mice were implanted with 27-gauge guide cannulae (RWD Life Science Co., Ltd., Shenzhen, China) targeting the BLA. For microinjection, the internal cannula (33-gauge) was inserted into the guide cannulae. For overexpression of dominant negative Sgk1 or Sgk1 in BLA→vHPC PNs, the AAV2/9-EF1α-DIO-Sgk1(S422A)-mCherry (200 nL, Obio Technology, Shanghai, China) or AAV2/9-EF1α-DIO-Sgk1-mCherry (200 nL, Obio Technology, Shanghai, China) or AAV2/9-EF1α-

DIO-mCherry (200 nL, control virus, Obio Technology, Shanghai, China) was injected into the bilateral BLA. The AAV2/retro-Syn-Cre (200 nL, Obio Technology, Shanghai, China) was injected into the bilateral vHPC. All viruses were freshly diluted in fresh and filtered 0.9% NaCl solution to obtain a final GC titer of $2-3 \times 10^{12}$.

## Fiber photometry

Neuronal activity was monitored by recording GCaMP fluorescence signals using an optical fiber recording system (Thinker Tech Nanjing Biotech Limited Co., Ltd.). Three weeks after virus injection and ceramic fiber-optic cannula implantation, mice were subjected to either FS or Sub-FS procedure. Calcium signals of BLA PNs, BLA→vHPC PNs, or BLA→NAc PNs were recorded before FS exposure (Pre-FS), and at 1 or 7 days post-FS. Before recording, mice were habituated in the recording room for 1 h. Then, a 470 nm LED (20 μW at the fiber tip) was delivered through the optical fiber, and the fluorescence signals were acquired at a sampling frequency of 100 Hz. Neuronal calcium signals were recorded during 10-min sessions of the OFT and EPMT, and data were analyzed using MATLAB. During recordings, behavioral tracking was accomplished using a video camera (Softmaze, XR-Video) positioned above the apparatus. Synchronization of behavioral events with calcium fluorescence signals was achieved by applying a screen recorder (EV Capture)[77]. The video footage was subsequently analyzed offline to detect entries into the center zone (OFT) or open arms (EPMT). An entry was defined as the mouse placing all four paws within a zone, and an exit as all four paws leaving it. A valid bout required a continuous stay within the zone lasting at least 0.5 s. Transient entries lasting under 0.5 s were excluded from subsequent neural analyses.

Time 0 s was defined as the moment the mouse entered the center zone during OFT or the open arms during EPMT. The fluorescence responses were calculated using the following formula: z-score = (FSignal − FBasal)/STD (FBasal). FBasal represents the average value during the baseline time (the 1 s pre-entry for OFT, the 2 s pre-entry for EPMT), while STD (FBasal) represents its standard deviation. Z-score values for individual mice were visualized in heatmaps, with group means presented in line plots; shaded areas denoted the SEM. The area under curve (AUC) of the z-score was derived by summing z-score values over a defined post-entry interval: 0–3 s for center zone entries in the OFT and 0–5 s for open arm entries in the EPMT. Subsequently, a session-level mean AUC was computed for each mouse by averaging the AUCs from all events of the same type within that session. All scoring criteria, time windows, and analysis steps were maintained consistently across all animals and testing days (Pre-FS, and 1 or 7 days post-FS).

## Histology and microscopy

For c-Fos staining, mice were first injected with retrograde fluorescent beads to label specific neuronal populations as required. Following a 2-week recovery period, mice were subjected to the FS stress paradigm. At either 1 day (1d P-FS) or 7 days (7d P-FS) post-FS, mice were placed on the open arms of the elevated plus-maze for 10 min to provide exposure to the stress-associated environment[78]. Ninety minutes after this exposure, mice were returned to their home cages and then perfused for subsequent histological analysis. Briefly, mice were anesthetized with 2% pentobarbital sodium and transcardially perfused with 50 mL of 0.1 M phosphate-buffered saline (PBS, pH 7.0–7.4), followed by 50 mL of 4% paraformaldehyde (PFA) in PBS. Brains were quickly removed and fixed in 4% PFA overnight at 4 °C. Coronal brain sections (40 μm thick) were sliced using VT1000S Vibratome (Leica Microsystems, Wetzlar, Germany) and washed three times in PBS (3 × 5 min). Sections containing the BLA were blocked in permeabilization buffer (0.1% triton X-100 in PBS) containing 10% normal donkey serum for 2 h at room temperature, followed by overnight incubation at 4 °C with primary antibody (c-Fos, 1:3000, Synaptic Systems,

226008, Germany; primary antibody against Sgk1, 1:200, Proteintech, 23394-1-AP, Chicago, IL, USA; anti-pCREB Ser133, 1:800, Upstate, 06-519, Germany). After incubation, sections were washed three times in PBST (3 × 10 min), then incubated for 2 h at room temperature with secondary antibodies (secondary antibody for c-Fos and Sgk1 staining: donkey anti-rabbit Alexa Fluor 647, 1:500, Life Technology, A31573, Carlsbad, CA; for pCREB staining: donkey anti-rabbit Alexa Fluor 488, 1:500, Thermo Fisher Scientific, A21206, Waltham, USA). After 3 washes in PBST (3 × 10 min), the sections were finally mounted onto slides using an anti-fading mounting medium with DAPI (Solarbio, S2110, Beijing, China). The images of the BLA were acquired using a confocal laser scanning microscope (Nikon A1, Nikon, Japan).

For c-Fos expression analysis, the number of c-Fos positive cells, red Retrobeads-labeled BLA→vHPC PNs and BLA→NAc PNs, and double-labeled cells were quantified using ImageJ software (NIH, Bethesda, MD, USA). For both stressed and unstressed mice, the number of c-Fos immunoreactive cells was counted by an experimenter blind to the experimental conditions.

For quantitation of Sgk1 and pCREB expression levels, a region of interest (ROI = mCherry positive neurons) was outlined using the ROI manager; the same ROI was then applied to the corresponding Sgk1 or pCREB fluorescence image for quantitation of Sgk1 or pCREB signal intensity. Measured intensity values were corrected for the background nonspecific autofluorescence.

## Electrophysiological slice recording

At 1 or 7 days after footshock, mice were anesthetized with ether and decapitated; the brain was quickly removed and chilled in ice-cold oxygenated (95% $O_2$ and 5% $CO_2$) cutting solution containing (in mM) 80 NaCl, 3.5 KCl, 4.5 $MgSO_4$, 0.5 $CaCl_2$, 1.25 $NaH_2PO_4$, 25 $NaHCO_3$, 90 sucrose, and 10 glucose. Coronal brain slices (320 μm-thick) containing BLA were prepared using VT1000S Vibratome (Leica Microsystems, Wetzlar, Germany). Subsequently, they were transformed into warmed (34 °C) artificial cerebrospinal fluid (ACSF) containing (in mM): 124 NaCl, 2.5 KCl, 2 $MgSO_4$, 2.5 $CaCl_2$, 1.25 $NaH_2PO_4$, 10 glucose, and 22 $NaHCO_3$ for 30 min, and then kept at RT for 1 h before recordings.

After recovery, a single brain slice was transferred to the recording chamber and continuously perfused with oxygenated ACSF at a rate of ~2 mL/min. The temperature of the ACSF was maintained at $29 \pm 1$ °C using an automatic temperature controller (TC-324B, Warner Instrument Co., Hamden, USA). Real-time imaging of the brain slice was conducted under a 40× water immersion objective (40×/N.A. 0.8, LUMPlanFL, Olympus) using an infrared differential interference contrast microscope (BX51WI, Olympus, Japan) equipped with a highly sensitive CCD camera (IR-1000E, DAGE-MTI, USA). Patch pipettes were prepared with a horizontal pipette puller (P-97; Sutter Instrument Co., Novato, CA). Whole-cell patch-clamp recordings were carried out using an Axon 700B amplifier and Digidata 1440A (Molecular Devices, San Jose, CA, USA) digital analog converter. During recording, the resistance of the pipette was maintained at 4–7 MΩ, and series resistance was monitored (within the range of 10–20 MΩ). Data showing a series resistance changes exceeding 20% were excluded. Signals were filtered at 3 kHz and sampled at 10 kHz.

For recording the intrinsic excitability and input resistance, brain slices were perfused with ACSF containing CNQX (20 μM) and picrotoxin (100 μM) to block excitatory and inhibitory synaptic transmission. Patch pipettes were filled with an internal solution containing (in mM): 130 potassium gluconate, 5 NaCl, 1 $MgCl_2$, 10 HEPES, 0.2 EGTA, 2 MgATP, and 0.1 NaGTP, adjusted to pH 7.3–7.4 with KOH. The cell membrane potential was held at -70 mV. Depolarization currents ranging from 50 to 250 pA, and hyperpolarizing currents from −200 to −50 pA (in 50 pA increments), were applied to induce action potentials and assess input resistance, respectively.

For recording miniature excitatory postsynaptic currents (mEPSCs), brain slices were perfused with ACSF containing tetrodotoxin (1 μM), picrotoxin (100 μM), and CGP52432 (5 μM). Patch pipettes were filled with an internal solution containing (in mM): 125 CsMe, 5 CsCl, 5 NaCl, 1 MgCl₂, 10 HEPES, 0.2 EGTA, 2 ATP-Mg, 5 QX314, and 0.1 GTP-Na, adjusted to pH 7.3–7.4 with CsOH. The membrane potential was held at −70 mV.

For recording miniature inhibitory postsynaptic currents (mIPSCs), brain slices were perfused with ACSF containing tetrodotoxin (1 μM), CNQX (20 μM), and D, L-AP5 (50 μM). Patch pipettes were filled with (in mM): 125 CsMe, 5 CsCl, 5 NaCl, 1 MgCl₂, 10 HEPES, 0.2 EGTA, 2 ATP-Mg, 5 QX314, and 0.1 GTP-Na, adjusted to pH 7.3–7.4 with CsOH. The membrane potential was held at 0 mV.

## Chemogenetic behavior

For chemogenetic experiments, virus injection surgery was performed as described above. The Clozapine N-oxide (CNO, MedChemExpress, USA) was given (injected intraperitoneally) at a target dose of 5 mg/kg 30 min before OFT and EPMT.

## Measurement of serum CORT levels

Blood samples were collected from the retroorbital plexus of mice after mild anesthesia, then incubated at room temperature for 60 min, and centrifuged at $2000 \times g$ for 15 min. The supernatant was collected and stored at −80 °C until analysis. The serum CORT levels were measured according to the instructions in the manual of the ELISA kit (No. 501320, Cayman Chemical, Ann Arbor, USA). The optical density was measured with an EnSpire Multimode Plate Reader (PerkinElmer, USA), and the concentration of CORT was determined by referencing a standard curve.

## Drug administration

For systemic drug treatment, stock solutions of metyrapone (MedChemExpress, USA), mifepristone (MedChemExpress, USA), and spironolactone (MedChemExpress, USA) were prepared in EtOH (final concentration < 0.1%) and subsequently dissolved in corn oil. Metyrapone (80 mg/kg/day), mifepristone (20 mg/kg/day), spironolactone (100 mg/kg/day), or vehicle control were administered intraperitoneally 30 min prior to the daily FS stress session.

For BLA local microinjection, mifepristone was first dissolved in DMSO and then diluted 1:10 in saline. A dose of 5 ng mifepristone or vehicle was microinjected into the BLA 30 min before the daily FS stress session.

## Transcription analysis of CORT-GR signaling relative gene in specific BLA PN subset

To label BLA→vHPC PNs and BLA→NAc PNs, AAV2/retro-hSyn-mCherry-WPRE and AAV2/retro-hSyn-EGFP-WPRE were injected into the vHPC and NAc, respectively. Brain sections (450 μm-thick) containing the BLA were prepared as described in the electrophysiological experiments and incubated in oxygenated ACSF containing actinomycin D (15 μM), pronase (1 mg/mL), CNQX (20 μM), D, L-AP5 (50 μM), and tetrodotoxin (1 μM) for 30 min at 11 °C, followed by a 15 min incubation at 37 °C. The BLA tissue was dissected under a stereomicroscope (M50, Leica Microsystems, Wetzlar, Germany) and dissociated using pipettes with decreasing inner diameters of 600 μm, 300 μm, and 150 μm. The cell suspension was then filtered through a 40 μm cell strainer. The mCherry and EGFP fluorescent labeled neurons were collected into a microelectrode by manipulating hydraulic manual microinjection (CellTram 4r Oil, Eppendorf, Hamburg, Germany). Samples containing 30–40 fluorescent labeled neurons were collected in lysis buffer (0.1% Triton X-100, RNase Inhibitor 4 U, Oligo-dT30VN Primer 2.5 μM, and dNTP 5 mM) and lysed using a PCR thermocycler (72 °C for 3 min followed by cooling to 4 °C). For reverse transcription, SuperScript II reverse transcriptase (100 U), RNase

inhibitor (10 U), Superscript II first-strand buffer, DTT (5 mM), Betaine (1 M), MgCl₂ (6 mM), and TSO (1 mM, Takara, Dalian, China) were added to the samples, and the following steps were performed: Step1: 42 °C for 90 min; Step2: 50 °C for 2 min, followed by 42 °C for 2 min, 10 cycles; Step3: 70 °C for 15 min; Step4: 4 °C forever.). The cDNA was obtained using multiplex PCR master mix (KAPA HiFi HotStart ReadyMix, Roche, Basel, Switzerland), IS PCR primers (0.1 mmol/L, Takara, Dalian, China) and Nuclease-free water, with the following amplification procedure: Step1: 98 °C for 3 min; Step2: 98 °C for 20 s followed by 67 °C for 15 s and 72 °C for 5 min, 20 cycles; Step3: 72 °C for 5 min; Step4: hold at 4 °C. The cDNA products were purified using KAPA Pure Beads (Roche, Basel, Switzerland). The RT-qPCR was conducted on StepOnePlus Real-Time PCR System (Thermo Fisher Scientific, Waltham, USA) using SYBR Green PCR Master Mix (Thermo Fisher Scientific, Waltham, USA). Relative mRNA levels of target genes were analyzed using the $2^{-\Delta\Delta CT}$ method, with the housekeeping gene *Gapdh* as the endogenous control for normalization. The sequence of primers used in the experiment is provided in Supplementary Data 2.

## RNAscope analysis

After perfusion fixation, the brain tissue was embedded and sectioned into 15 μm slices using a freezing microtome (CM1950, Leica Microsystems, Wetzlar, Germany) and fixed on an anti-detachment slide. Following the manufacturer's protocol for the RNAscope multiplex fluorescent reagent kit and RNA-protein co-detection ancillary kit (Advanced Cell Diagnostics, California, USA), the tissue sections were fixed and pretreated with 4% PFA and kit reagents to facilitate antigen repair. To assess co-localization of mCherry- and GFP-tagged cells with *Sgk1* mRNA, the slices were incubated overnight at 4 °C with primary antibody for mCherry (1:500, mCherry Monoclonal Antibody (16D7), M11217, Thermo Fisher, USA) and GFP (1:500, anti-green fluorescent protein antibody, GFP697986, Aves Labs, California, USA). Following primary antibody incubation, the sections were hybridized with probes specific for *Sgk1* mRNA (Mm-Sgk1, 434791, Advanced Cell Diagnostics) using the HybEZ hybridization system at 40 °C. Signal amplification was achieved via hybridization with AMP reagents and subsequent incubation with Opal 690 reagent (1:1500, ASOP690, PerkinElmer, USA). After the hybridization reaction was completed, the slices were incubated with the secondary antibody for mCherry (1:100, Alexa Fluor 594 donkey anti-rat, A21209, Thermo Fisher, USA) and GFP (1:100, Alexa Fluor 488 conjugated affinpure donkey anti-chicken, 703-545-155, Jackson ImmunoResearch, USA) for 30 min at RT, then re-stained with DAPI and sealed with a quenching sealing agent. Finally, the slices were imaged using a confocal laser scanning microscope (Nikon A1, Nikon, Japan). The somas of each mCherry-labeled and GFP-labeled neuron were outlined, and the overlaying dots (representing *Sgk1* transcripts) were quantified using ImageJ software. To quantify Sgk1 expression, we analyzed a total of 5 unstressed control and 5 experimental animals. In each animal, Sgk1 expression was assessed only in cells displaying clear and typical morphology; cells with incomplete or distorted morphology were excluded from the analysis. For each animal, at least 15 cell somas were analyzed, sampled from a minimum of 3 distinct fields of view across different sections. Regions of interest (ROIs) were defined as the outline of each cell soma at its maximum diameter, and the number of Sgk1 signal dots within each ROI was counted. To ensure consistent selection and unbiased cell selection between groups, all procedures—including cell identification and subsequent quantification—were performed by an observer blinded to the experimental conditions.

## Statistics

Data analyses were performed with GraphPad (GraphPad Software, La Jolla, USA) and SPSS. The homogeneity of variance and normality of the data were analyzed with Bartlett's test and the Kolmogorov–Smirnov test, respectively. For data that satisfied these

assumptions, comparisons were made using Student's *t*-test, one-way ANOVA, two-way ANOVA or three-way ANOVA with or without repeated measures, followed by post hoc *t*-tests or simple-effect test with Bonferroni's correction. Non-normally distributed data were analyzed using nonparametric tests, such as Mann–Whitney *U* test or Kruskal–Wallis test. Data are presented as mean ± standard error of the mean (SEM), and statistical significance was defined as $*p < 0.05$, $**p < 0.01$, $***p < 0.001$, $\#p < 0.05$, $\#\#p < 0.01$, $\#\#\#p < 0.001$, or $\dagger p < 0.05$, $\dagger\dagger p < 0.01$, $\dagger\dagger\dagger p < 0.001$. See Supplementary Data 1 for full statistical information. Source data are provided as a Source data file.

### Reporting summary

Further information on research design is available in the Nature Portfolio Reporting Summary linked to this article.

### Data availability

All data generated and analyzed during this study are included in this article and its supplementary information files. The datasets have also been deposited in the Figshare repository under the accession link https://doi.org/10.6084/m9.figshare.28417763. Source data are provided with this paper.

### Code availability

The MATLAB-based software package used for analyzing the fiber photometry data in this study was developed by ThinkerTech Nanjing BioScience Inc. and is publicly available via the Zenodo repository at: https://doi.org/10.5281/zenodo.15009455.

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

## Acknowledgements

This work was supported by grants from the National Natural Science Foundation of China (Grant Nos. 82125010 and 82430048 to BXP, U25A2080 and 32222034 to WHZ, 82571729 to WZL), Natural Science Foundation of Jiangxi Province (Grant No. 20242BAB24003 to BXP, 20252BAC230007 to WHZ, 20252BAC220060 to WZL, 20242BAB20383 to HQP), National Key R&D Program of China (2021ZD0202704 to BXP). This work is also supported by the Medical-Engineering Interdisciplinary Talent Development Program and Medical Cross Innovation Fund of Nanchang University (NCUJCCX-2024-05).

## Author contributions

W.H.Z. and B.X.P. conceived the idea for this study, and J.X.Z., W.H.Z., and B.X.P. designed the study. J.X.Z. and W.Z.L. performed the experiments with the help of Y.Q.L., Y.Y.L., W.J.Y., H.Q.P., and C.Y.W. B.X.P., W.H.Z., and J.X.Z. wrote the manuscript. The order of co-first authors was determined based on their involvement and contribution to the project.

## Competing interests

The authors declare no competing interests.
