## [Transparent Peer Review file · Nature Communications]

Sgk1 upregulation in hippocampus-projecting amygdala neurons underlies the delayed onset of PTSD-like avoidance behavior

Corresponding Author: Professor Bing-Xing Pan

Version 0:

Reviewer comments:

Reviewer #1

(Remarks to the Author)

Findings summary:

Zou and Liu et al. present a comprehensive study that shows that the BLA-vHPC circuit is involved in PTSD-like avoidance behavior. They show that this circuit is hyperactive 7 days following a footshock stress paradigm that lasts 2 days using in vivo fiber photometry and ex vivo electrophysiology and immunohistochemistry. Additionally, the authors inhibit the circuit with DREADDs and show a loss of avoidance behavior. The authors also show that CORT synthesis and activation of the glucocorticoid receptor is required for the avoidance behavior. Finally, the authors show that inhibition of SGK1 function blocks avoidance while over expression makes mice more susceptible to stress behaviors.

This manuscript is comprehensive and stands to advance the field. However, this reviewer has several major concerns.

Major concerns

1. Female mice were not used in this study and there is no rationale as to why the authors exclude female mice. Women are at much higher risks for developing PTSD and more likely to seek treatment for PTSD. PTSD in women is also associated with changes to amygdala and hippocampal size and function. It is therefore crucial to examine sex differences between males and females in this study. This reviewer strongly recommends repeating the key findings in female mice. A negative sex difference is stronger support for translational impact of these findings to use SGK or CORT inhibitors as these authors suggest, and important for treating a larger population. <https://www.nlm.nih.gov/health/statistics/post-traumatic-stress-disorder-ptsd> https://www.ptsd.va.gov/professional/treat/specific/ptsd_research_women.asp

2. The analysis of the fiber photometry is very limited.

a. Using normalized peak frequency with the baseline being pre foot shock is particularly concerning as there is not a correct control for the interpretation. This reviewer strongly recommends repeating this experiment with a no shock control for all fiber photometry experiments and not normalizing to pre shock baseline. The interpretation of these data are currently very limited and it cannot be concluded that the increased activity of the BLA-vHipp pathway is from the shock and, as an example, not from increased handling through the experiment without a no shock control. Given that this timepoint was 3-9 days prior to the animal being shocked there is also a possibility that there will be changes in viral expression of gcamp during these times. Changes in viral expression could mean more virus reaching more cells and thus changes in the activity pattern of neurons and it is not recommended to use this as a normalizing baseline.

b. It is recommended to use a non-shocked group control to compare the average zscore at the Baseline recording, 1 day and 7 day. (ie: mean z-score of no shock vs shock baseline, mean z-score no shock vs shock 1 day, etc) Then use the total spikes averaged across animals in the time bin. Additionally, using a MAD of 2 is low although not unacceptable, but limits interpretation when this is the only statistic given. The AUC, min/max, and total count of peaks should be included as well for all fiber photometry analyses.

c. It is not clear where the recording is taking place. Is it a novel cage? If so is the animal exposed to this cage before the baseline test day to eliminate artifacts from a novel cage? Is the animal re exposed to the footshock compartment for recording? Is it homecage and the cagemates are removed? This reviewer recommends stating clearly where the recordings

are taking place because many of these options have caveats to the interpretation of the data. How often is the animal handled before and during the experiment, and where the animal is being recorded from is also a concern that is not addressed in the methods.

d. I recommend recording during OFT and EPM behavior and looking at peri event activity. It would serve the interpretation of the experiment to look at the underlying neural activity while the animal is making the decision to enter open arms or center and look to see if the activity is different in these zones and compare shock vs no shock.

3. For SGK D/N the authors have not clearly established the function of this virus. Although the authors show increased mRNA of the D/N there is not an experiment to show that the Sgk1-DN is functionally valid. I recommend showing an experiment in which the function of Sgk1 is inert, or citations showing that this method has been used previously. The concern is also major because the authors did not cite where this virus was made or how it was made if it was made in house. More details are necessary. The works cited in this paragraph show using AAV-dnSGK1 but not DIO so I recommend a more detailed validation of this tool. A minor concern is that in Figure 6i, the authors show in 1 animal an increase in sgk1 expression but there is no validity that the protein expression is different across multiple mice. I recommend doing a fluorescence analysis in multiple animals to show protein quantification using IHC or a western blot analysis similar to the work that they have cited. Similar major concerns remain for figure 7 using the AAV-DIO-Sgk1 for over expression. Need to show that this effect is true across many mice, especially as the mRNA is variable and there are no statistics in the figure showing that this increase is statistically significant.

Minor Concerns

1. The fiber photometry data are a major conclusion and should be moved to the primary figures from supplement fig. 6 and 8.
2. In figure 2 l-q, adding a schema of the timeline of when the animals are being shocked and what the control is would help readers to understand this experimental design. It is unclear when the animals are being shocked and when the tissue is harvested. It is recommended that the virus be injected at the same time and delay the footshock so that the 1 day collected group and 7 day collected group are occurring on the same day (like in supplement). This delayed pattern is sometimes used throughout the paper and a schema would help decrease these concerns.
3. In the DREADDs experiment it is recommended to have a CNO control with mCherry virus control
4. The sub threshold FS in figure 7 is 0.5mA 120 shocks/day for 2 days with no duration or interval given. Why was this paradigm chosen over the 0.3mA shown in supplement. Need clearer rationale.
5. In the graphic figure 8, the sgk1 expression in the BLA-NAc is a highlight of the figure but this is supplement data. I recommend either making this a main text figure if it is the key conclusion, or removing it from the graphic.
6. In the abstract authors use abbreviation PN without defining

Figure 5 beginning of 6 is a highlight of the paper and shows an interesting mechanism underlying the avoidance behaviors that the authors describe. This reviewer also acknowledges that there is a strength to the breadth of techniques shown in the manuscript.

Reviewer #2

(Remarks to the Author)

Remarks to the Authors

This manuscript combines multiple experimental techniques (behavioral assays, fiber photometry, chemogenetics, and ex vivo electrophysiology) to investigate how BLA circuit alterations underlie elevated anxiety in a mouse model of PTSD. The authors report that following footshock stress, BLA-vHPC neurons exhibit increased spontaneous activity in vivo and heightened excitability ex vivo. They further show that glucocorticoid receptor activation during stress is necessary for both the behavioral phenotype and BLA-vHPC activity changes. Moreover, they identify Sgk1 as being upregulated in BLA-vHPC neurons and bidirectional manipulations of Sgk1 suggest that Sgk1 levels modulate BLA-vHPC neuronal excitability and anxiety in stress-exposed mice. These findings address a significant question in the field—how stress during trauma leads to persistent circuit-level alterations that drive long-term fear and anxiety. The manuscript is generally well-written and organized, and the figures clearly present the results. However, some of the experiments and analyses do not directly address the relevant questions leading to overstated conclusions. In particular, the fiber photometry experiments would benefit from more rigorous design and standard analytical approaches, which currently undermine the overall interpretation.

Major Comments

- 1) The authors do not describe the procedures for their fiber photometry recordings and cFos experiments, only stating that they measured “spontaneous activity” presumably at rest. While differences in resting activity can be informative, they do not directly relate to the state-dependent anxiety behavior in the OFT and EPM. Consequently, it is not justified to conclude that increased BLA-vHPC activity underlies the increased anxiety. To address this, the authors should replicate their photometry and cFos findings during the OFT and EPM. For the photometry experiments, they should correlate neural activity dynamics with real-time behavior (e.g. exploration of anxiogenic vs. non-anxiogenic zones).
- 2) The photometry experiments employ a within-subjects design in which the same mice are recorded at Pre-FS, 1 day Post-FS, and 7 days Post-FS. This approach does not adequately control for time-dependent changes in neural activity that may occur regardless of PTSD induction. To address this, the authors should adopt a between-subjects design that includes an experience-matched control group, ideally mice exposed to a the subthreshold-FS protocol (as in Fig. 7). This group would help determine whether any observed changes in neural activity truly stem from PTSD induction rather than changes in activity over time or due to footshock exposure.

- 3) The photometry data require more rigorous analysis. It appears that the authors used the raw fluorescence rather than computing the %dF/F0 which is standard practice to account for baseline differences across animals or sessions. More critically, they normalized the Post-FS conditions by the Pre-FS condition effectively eliminating variance in the Pre-FS condition and obscuring the true variability in the Post-FS data. Furthermore, the statistical tests applied to this data, one-way ANOVA and t-tests, are inappropriate since the Pre-FS condition was used for baseline normalization which violates model assumptions. The authors should replace these plots with the unnormalized data (Fig 1. L,M; Fig. 2 E-K, Suppl Fig. 6., Supp Fig. 8). Furthermore, the authors should supplement the calcium event analysis by presenting the % dF/F0 (e.g. mean %dF/F0 in each condition, time-locked to behavior in anxiety tests).
- 4) The authors assert that Sgk1-DN prevents, and Sgk1-overexpression increases susceptibility to FS-induced hyperactivity and anxiety. However, the statistical comparisons and some of the results of the Sgk1 experiments do not strongly support these claims. The authors mainly compared mice receiving FS-exposure+Sgk1 manipulation versus no-FS+Sgk1 manipulation, which does not directly test how Sgk1 manipulations affect FS-exposed mice. The critical comparison should be FS-exposure+mCherry versus FS-exposure+Sgk1 manipulation. In several plots, this comparison is not significant (Fig. 6m-n EPM behavior) and entirely not significant for the Sgk1-overexpression experiment in Fig.7, weakening the claim that Sgk1 levels are important for the behavioral and electrophysiological effects of footshock. Furthermore, in Fig.6c the authors report elevated Sgk1 mRNA. However, the statistical testing relied on multiple t-tests without correcting for multiple comparisons which calls into question the robustness of that result.

Minor Comments

- 5) The introduction only describes the clinical literature and is missing the relevant findings in rodents that justify the study's experiments.
- 6) It remains unclear whether the inescapable footshock primarily models general stress-induced anxiety rather than a distinct PTSD-like condition in which fear memory impairments are central. The authors should conduct fear conditioning and extinction at the 7-day time point and demonstrate an extinction retention deficit in this model.
- 7) In the cFos experiments (Fig. 2M-Q), the fewer double-labeled cells in the BLA-NAc condition could be partially explained by fewer cFos+ cells overall. The authors should compare the density of cFos+ cells between tracing conditions to rule this out.
- 8) In Fig. 4, the authors administered either vehicle or CNO prior to OFT and EPM testing and observed reduced avoidance behavior in the CNO group. Previous studies indicate that inhibiting the BLA-vHPC circuit can reduce anxiety-like behavior in healthy mice, so it is unclear if the observed reduction reflects a general anxiolytic effect or a restoration of normal anxiety levels. Also, because CNO (through its metabolite clozapine) is known to have off-target effects, it is essential that the authors administer CNO to the control groups. A more rigorous design would include 2 mCherry-expressing control groups that both receive CNO: one subjected to the footshock protocol and one naive. Additionally, to substantiate the claim that BLA-vHPC hyperactivity underlies the increased anxiety, they should demonstrate specificity by confirming that inhibiting BLA-vHPC neurons but not BLA-NAc neurons in FS mice normalizes anxiety.
- 9) In Fig. 5, mifepristone but not spironolactone administration during FS was found to block subsequent behavioral and electrophysiological changes. This result motivates the subsequent exploration of glucocorticoid-associated mechanisms within BLA-vHPC neurons triggered by FS. However, because these drugs were provided systemically, it is impossible to determine whether their effects are mediated by actions at the amygdala or elsewhere in the brain and periphery. To strengthen the mechanistic link between glucocorticoid receptor activation and the upregulation of Sgk1 in BLA-vHPC neurons, the authors should perform local microinfusions of mifepristone into the BLA during FS and assess its impact on the emergence of behavioral and electrophysiological phenotypes.
- 10) In line 105-106, the statement that BLA-vHPC neurons are preferentially engaged in aversive behaviors is not entirely accurate based on previous studies (Pi et al. 2020, Nat Comm; Beyeler et al. 2016 Neuron).
- 11) What is the difference in experimental conditions between the control data presented in Fig. 7g-j and Suppl Fig. 7?
- 12) Fig.6P and 7L should be analyzed with 3-way ANOVA (stress, protein, current factors).
- 13) The authors have not provided critical details of their fiber photometry experiments. They should describe in the methods their optical setup, sampling rate, light intensity, how raw signals were filtered/transformed, how z-scores were taken and any further normalization.
- what the experimental procedures were for the spontaneous recordings ie. arena, handling, recording protocol, and duration. If the recording duration was 1 minute as presented in the figures, then this is an insufficient amount of sampling time to draw conclusions about resting state activity.
- 14) In line 334, they state 4-10 week old mice were used. Since mice reach adulthood at 8 weeks, did the authors perform behavioral tests on adolescents?
- 15) For the anxiety tests, they should describe the lighting conditions since brightness correlates with avoidance. For the EPM, they should provide the height of the maze.
- 16) In line 467, "concentration" should be "dose".
- 17) In the methods for RNAscope, they should describe how cells were selected, how many cells per mouse, and how they ensured consistent selection between groups.
- 18) The authors have not provided a description of the procedures for the cFos experiments in the methods.

Reviewer #3

(Remarks to the Author)

The authors used a mouse model of PTSD induced by inescapable footshock (FS) stress to identify a delayed yet persistent escalation in avoidance behavior, which was specifically associated with the activation of basolateral amygdala (BLA) neurons projecting to the ventral hippocampus (BLA→vHPC PN), but not those targeting the nucleus accumbens. This

projection-specific activation arises from enhanced excitability and increased excitatory transmission in BLA→vHPC PNs, mediated by glucocorticoid receptor (GR) signaling. They also discovered that upregulation of serum- and glucocorticoid-regulated kinase 1 (Sgk1) acts as a crucial regulator of FS-induced hyperactivation in BLA→vHPC PNs and the consequent development of PTSD-like avoidance behaviors. These results suggest a potential role of projection-specific Sgk1 upregulation in BLA neurons in the pathophysiology of PTSD-like avoidance behavior. While the results are intriguing, I have several concerns:

1. Regarding the animal models of PTSD, avoidance behavior is one of the behavioral characteristics that occur after stress stimuli such as footshock. However, can it fully represent the core symptoms of PTSD? The freezing levels, which are a critical measure of fear response, are not presented anywhere in the manuscript. Do freezing levels increase or decrease after one or two weeks? Does fear generalization develop after one or two weeks? Given that PTSD is fundamentally a memory disorder, the authors did not provide sufficient evidence to demonstrate that their animal model successfully recapitulates PTSD. How does fear memory change with inhibition of BLA→vHPC projections, or disruption of GR signaling in BLA→vHPC pathway, or Sgk1 knockout? It is hard to evaluate the PTSD syndrome only with avoidance behaviors. Some labs have already reported that animals exhibit avoidance behaviors after acute stress, such as footshock. While the authors conducted numerous experiments and concluded that "Footshock stress exposure causes delayed and enduring PTSD-like avoidance behavior in mice," this finding is not novel and has been previously documented. The results are presented in Figure 1 and Supplementary Figures 1-3, but they do not significantly advance the field. Additionally, how does the activity of BLA neurons or BLA-vHPC/NAc projection neurons (PNs) change during behavioral tasks such as the Open Field and Elevated Plus Maze? This information is crucial for understanding the broader functional implications of these neuronal populations in PTSD-like behaviors.
2. Are BLA→vHPC PNs and BLA→NAc PNs are two different neuronal populations in the BLA? What is the overlap rate of BLA→vHPC PNs and BLA→NAc PNs? The double retrograde tracing experiment is suggested.
3. CORT signaling plays a critical role in PTSD is well-known and the results of CORT synthesis inhibitor or GR antagonist on FS-induced avoidance behavior are not that surprise or attractive.

Minor concerners:

In Figure 1i-o, did the author do the photometry recording in the context of FS or in homecage? So, the question is whether the enhanced activities of BLA neurons is induced by FS-related context or automatically appears. The information is lack in the whole manuscript.

How does knockout or expression of Sgk1 affects GR signaling in the BLA neurons? At least, this should be discussed.

The overexpression levels of Sgk1-DN or Sgk1 are excessively high and appear to induce toxic effects. The calculation of mRNA expression is incorrect. They should be compared with the housekeeping gene (e.g., GAPDH or β -actin), rather than mCherry.

Reviewer #4

(Remarks to the Author)

This paper by Jia-Xin Zou, Bing-Xing Pan, and colleagues presents an exciting set of experiments showing that stress-induced increases in Sgk1 within the amygdala – specifically those neurons that project to the hippocampus - support persistent alterations in anxiety-related behaviors. They replicate findings that amygdala projections to the ventral hippocampus are important for anxiety-related behavior (Felix-Ortiz et al, 2013). More importantly, they go on to show that stress-induced glucocorticoid release and persistent increases in Sgk1 expression within this cell population are important for enhancements in neuronal excitability and changes anxiety related-behavior. This work provides a more mechanistic link between stress-induced glucocorticoid release and limbic circuit alterations relevant to anxiety disorders. There a few conceptual and experimental issues with the work but I think all of these can be easily remedied.

Major concerns:

The authors erroneously claim that over-expression of Sgk1 is sufficient to promote anxiety-related behavior in the open field and elevated plus maze tests. The authors suggest that because animals in which they overexpress Sgk1 in the amygdala show a significant increase in anxiety-related behavior in response to a 'subthreshold' stressor, whereas control animals that do not overexpress Sgk1 do not, that Sgk1 promotes this behavior. However, there is an apparent trend in the control animals. Moreover, there is not a statistical interaction between viral group and stress condition – only a main effect of stress. As such, the statistical reasoning they make is false. The authors need to either omit the claim that overexpression promotes anxiety-related behavior or increase sample size until they have sufficient power to detect the interaction. That said, I find the paper just as impactful if Sgk1 is necessary for the observed behavioral change but not sufficient.

The authors repeatedly use the term 'avoidance' and I think that this is problematic for several reasons. First, the behavioral tests they utilize are most often referred to as tests for anxiety-related behavior, including work they base their findings on (Felix-Ortiz, 2013). Moreover, typically people use avoidance in the context of active avoidance procedures (e.g., pressing a lever or moving to platform to avoid an aversive outcome). Consequently, calling this avoidance results in confusion. Second, and perhaps more importantly, avoidance has historically been defined in the learning literature as an instrumental action taken to remove the onset of an aversive outcome (Domjan, Principles of Learning and Behavior textbook). In such cases, there is a clear contingency/relationship between taking an action and the presentation of an aversive outcome. The open-field and elevated plus maze tests are problematic because it is unclear what these contingencies are.

The authors demonstrate that administration of CNO (vs vehicle) is able to diminish anxiety-related behaviors in animals that express HM4Di in BLA→vHC projections. However, they utilize a high dose of CNO that can have off-target effects. A supplementary experiment showing that the dose of CNO used has no effect in control animals would be ideal.

The authors look at spontaneous activity of BLA neurons using fiber photometry and cFos. First, it is unclear where this was done. In the home cage? These methods should be outlined. Secondly, it is somewhat confusing why it was done outside of the open-field or elevated plus maze. Often times there is little to no expression of cFos in the home cage. Could the authors explain. Thirdly, given that heightened activity of BLA to vHC projection neurons are observed outside of the elevated plus maze and open field, what does this say about the expression of the behavioral changes observed. How can activity of a projection be important for a behavior that is not occurring?

MINOR

Please include titers for viruses

It is stated that calcium events were defined as increases of 2 std deviations above baseline, but the authors do not say what that baseline is (the minimum value over the entire session?)

The authors show that the change in anxiety-related behavior observed after an initial stressor increases with time from that stressor. Notably, fear generalization is often thought to increase over time (Wiltgen and Silva, 2007). To what extent are the authors findings dependent upon associative fear of the initial stressor context? Does silencing the amygdala to ventral hippocampus projection also reduce fear of the initial stressor context? I don't feel addressing this question is critical. Mostly my intrigue.

Version 1:

Reviewer comments:

Reviewer #1

(Remarks to the Author)

Zou et al present a manuscript that outlines a specific amygdalar mechanism that underlies the delayed onset of PTSD-like avoidance behavior. They show that 7 days following an inescapable footshock protocol, there is hyperactivation of the BLA-VHPC pathway that is driven by upregulation of Sgk1, which functions downstream of the glucocorticoid receptor.

The authors did an impressive, thorough job addressing the reviewers' comments. The significance and impact of the primary findings are made much clearer in the revision. These findings show a clear mechanistic underpinning of the mechanism of persistent PTSD-like avoidance behavior, and set next steps for intervention to prevent delayed avoidance behavior after inescapable stress.

Two very minor suggestions for the text:

1) In figure 2s and w, the authors should consider making it clearer in the figure legend that these figures are for cfos only in the BLA (if this is the correct interpretation). A simple text edit for clarity would help readers. This wording is particularly confusing for q-t because the figure legend says "same as in q-t except that the data were from BLA-NAc PNs," except the cfos data in w is from BLA, not from the PNs, I believe.

2) The use of just a generic term "control group" became confusing after a few figures because the control can change, particularly when thinking about the control for the DREADD experiment is vehicle not an unstressed control. In figure 5, it was slightly unclear what the control was, given a different control in figure 4. The figure legend only says that "data was normalized to control group." It would help improve readers interpretation to reiterate unstressed control throughout the text. Also in line 91, where this is first defined, it can help the reader to add clarifying detail that this is how "control" is defined throughout.

Overall, the manuscript is well done. Congratulations to the authors on the work.

Reviewer #2

(Remarks to the Author)

I commend the authors for the substantial amount of work they have done for the revision. The photometry experiments add strong support for the main claims, and my questions have been well addressed. I have a few minor comments that primarily concern clarity of the newly added photometry results.

1) Figures 1, 2, 6, and 7: For the AUC analysis, the authors report a 2-way RM ANOVA (treatment x day) and follow up with within-group comparisons. Since the photometry signal can change across days for reasons unrelated to the PTSD manipulation (ie. expression, recording stability, habituation) the key question is whether the FS group differs from the

control group at each time point. Please report the group effects at pre-, 1d, and 7d (corrected for multiple comparisons) and describe these findings in the Results.

2) Figures 1, 2, 6, and 7: Peri-event activity is shown but there is no initial description of what the plots display. Please add a brief description in the Fig 1 legend where these plots first appear. If the peri-event traces are intended for visualization only, please state this explicitly and confirm that AUC analyses were derived from the same data.

3) Because the photometry mice underwent a different behavioral sequence from the main PTSD paradigm (three repeated OFT/EPM tests at pre-, 1d, and 7d), please show their behavioral performance. This will allow readers to relate the day-dependent neural changes and behavioral changes in the same animals. Please also state whether the photometry mice exhibit the same anxiety-like phenotype at the 7-day time point.

4) Methods: The photometry section is missing important details on data handling and behavioral alignment. Was mouse position tracked with software? How was activity aligned to behavior? How were entries/exits/bouts defined (manual vs. automated, head or centre tracking, bout cutoffs, exclusion criteria)? How was AUC computed (e.g. per bout over a fixed window and then averaged across bouts, or some other approach?). Please also indicate whether the same criteria were applied across animals and recording days.

5) Figures 1, 2, 6, and 7: Please specify in the results and/or legend the exact conditions used for the photometry control group.

6) Fig. 1: The legend should state what the photometry heatmaps represent (e.g. data from a single animal vs. group-average, are rows individual bouts/trials?).

7) Fig. 4: Please include the mCherry 7d-P-FS results (vehicle and CNO) in the graphs alongside the hM4D results. Statistical testing should be performed between mCherry and hM4D groups (e.g. mixed ANOVA (group, drug) followed by post-hoc tests within each drug condition).

8) Methods: The contextual fear conditioning section has an unfinished sentence. Please also report if any software was used for freezing detection and the criteria used.

Reviewer #3

(Remarks to the Author)

I commend the authors on the revised version of the manuscript. The authors have addressed all of my concerns. The experiments are extensive and the results are exciting.

Reviewer #4

(Remarks to the Author)

The authors have addressed all of my initial concerns. This is an impressive set of experiments!

Point-to-point response to the reviewers' comments

REVIEWER COMMENTS

Reviewer #1 (Remarks to the Author):

Findings summary:

Zou and Liu et al. present a comprehensive study that shows that the BLA-vHPC circuit is involved in PTSD-like avoidance behavior. They show that this circuit is hyperactive 7 days following a footshock stress paradigm that lasts 2 days using in vivo fiber photometry and ex vivo electrophysiology and immunohistochemistry. Additionally, the authors inhibit the circuit with DREADDs and show a loss of avoidance behavior. The authors also show that CORT synthesis and activation of the glucocorticoid receptor is required for the avoidance behavior. Finally, the authors show that inhibition of SGK1 function blocks avoidance while over expression makes mice more susceptible to stress behaviors.

This manuscript is comprehensive and stands to advance the field. However, this reviewer has several major concerns.

Major concerns

1. Female mice were not used in this study and there is no rationale as to why the authors exclude female mice. Women are at much higher risks for developing PTSD and more likely to seek treatment for PTSD. PTSD in women is also associated with changes to amygdala and hippocampal size and function. It is therefore crucial to examine sex differences between males and females in this study. This reviewer strongly recommends repeating the key findings in female mice. A negative sex difference is stronger support for translational impact of these findings to use SGK or CORT inhibitors as these authors suggest, and important for treating a larger population. <https://www.nimh.nih.gov/health/statistics/post-traumatic-stress-disorder-ptsd> https://www.ptsd.va.gov/professional/treat/specific/ptsd_research_women.asp.

Response: We sincerely thank the reviewer for the insightful and constructive comments. We fully agree that females are at higher risk for developing PTSD, and that assessing sex differences is critical to enhance the translational relevance of our findings. Accordingly, we have conducted additional experiments in female mice to determine whether they exhibit similar behavioral and neural phenotypes to males following exposure to inescapable foot shocks (FS) (Response Fig. 1-5).

We first assessed whether female mice develop delayed and persistent PTSD-like avoidance behavior under the same FS paradigm used in males (Response Fig. 1a, b, now Supplementary Fig. 4a, b in the revised manuscript). Consistent with the findings in male mice, our behavioral analyses revealed that FS did not alter the time spent in the center zone during the open field test (OFT) or the time spent in, or entries into, the open arms during the elevated plus maze test (EPMT) at 1 day post-FS; however, stressed female mice showed significantly reduced time spent in the center zone and the open arms, as well as fewer entries into the open arms, at 7 days post-FS. Importantly, total travel distance was not affected on either day 1 or day 7 after stress, indicating that locomotor activity remained intact (Response Fig. 1c-g, now Supplementary Fig. 4c-g in the revised manuscript, with citation on page 4, lines 92-95). These results suggest that the FS paradigm induces delayed and persistent PTSD-like avoidance behavior in both male and female mice.

 **Response Fig. 1 (related to Supplementary Fig. 4). Footshock stress causes delayed PTSD-like**
 **avoidance behavior in female mice.** **a** Schematic of FS procedure. **b** Time scheme for avoidance
 behavior test in the open field (OFT) and the elevated plus maze (EPMT). **c** Representative activity
 tracking in the OFT and the EPMT. **d, e** Summary plots of time in the center area (**d**) and total distance
 traveled (**e**) in OFT (Control: $n = 11$ mice; 1d P-FS: $n = 12$ mice; 7d P-FS: $n = 11$ mice). **f, g** Summary
 plots of time in the open arms (**f**) and open arm entries (**g**) in EPMT (Control: $n = 11$ mice; 1d P-FS: $n =$
 12 mice; 7d P-FS: $n = 11$ mice). Data were shown as mean \pm SEM. * $p < 0.05$, *** $p < 0.001$. See
 Supplementary Data 1 for full statistical information. Source data are provided as a Source Data file.

 *Given that heightened fear is another core feature of PTSD, we next evaluated contextual fear in*
 *stressed female mice (Response Fig. 2a, now Supplementary Fig. 6a in the revised manuscript). Our*
 *results showed that FS exposure markedly increased freezing behavior on both day 1 and day 7 post-*
 *stress compared with controls, confirming robust fear responses (Response Fig. 2b, now Supplementary*
 *Fig. 6b in the revised manuscript, with citation on page 4, lines 96-101).*

 **Response Fig. 2 (related to Supplementary Fig. 6) FS causes sustained trauma-related fear in mice.**
 Schematic of the contextual fear conditioning test in female mice. **a** Time scheme for contextual fear
 conditioning (CFC) test. **b** Summary plots of freezing level of mice in CFT (Control: $n = 11$ mice; 1d P-
 FS: $n = 12$ mice; 7d P-FS: $n = 11$ mice). Data were shown as mean \pm SEM. ** $p < 0.01$, *** $p < 0.001$.
 See Supplementary Data 1 for full statistical information. Source data are provided as a Source Data file.

Since we previously identified glucocorticoid receptor (GR) signaling as a key mediator of FS-induced
 avoidance behavior in male mice, we further examined whether the same mechanism underlies the
 behavioral phenotype in females. To this end, we bilaterally infused the GR antagonist mifepristone into
 the basolateral amygdala (BLA) prior to FS exposure (Response Fig. 3a, b, now Supplementary Fig. 11a,
 b in the revised manuscript). Behavioral testing revealed that mifepristone pretreatment significantly
 attenuated FS-induced delayed avoidance behavior (Response Fig. 3c-f, now Supplementary Fig. 11c-f
 in the revised manuscript). Similarly, mifepristone administration also reduced FS-induced fear
 responses (Response Fig. 4, now Supplementary Fig. 12 in the revised manuscript, with citation on page
 7, lines 198-201). These findings indicate that GR signaling contributes to the development of PTSD-like
 avoidance behavior in female mice.

 **Response Fig. 3 (related to Supplementary Fig. 11) Blocking of GR signaling within BLA mitigates**
 **FS-induced increase in avoidance behavior and BLA PNs hyperactivation in female mice. a**
 **Schematic of experiment procedures. b Representative images showing the cannula implantation in BLA.**
 **c, d Summary plots of time in center area (c) and total distance traveled (d) in OFT (Vehicle+control: n**
 **= 13 mice; Vehicle+7d P-FS: n = 12 mice, Mifepristone+control: n = 14 mice; Mifepristone +7d P-FS: n**
 **= 14 mice). e, f Summary plots of time in open arms (e) and open arm entries (f) in EPMT, sample size**
 **as in panel (c, d). Data were shown as mean \pm SEM. * p < 0.05, ** p < 0.01. See Supplementary Data 1**
 **for full statistical information. Source data are provided as a Source Data file.**

 **Response Fig. 4 (related to Supplementary Fig. 12) Blocking of GR signaling within BLA mitigates**
 **FS-induced increase in fear memory in female mice. a Schematic of experiment procedures. b**
 **Summary plots of freezing level of mice in CFC test (Vehicle+control: n = 13 mice; Vehicle+7d P-FS: n**
 **= 12 mice, Mifepristone+control: n = 14 mice; Mifepristone +7d P-FS: n = 14 mice). Data were shown**
 **as mean \pm SEM. *** p < 0.001. See Supplementary Data 1 for full statistical information. Source data are**
 **provided as a Source Data file.**

Finally, we investigated whether BLA neurons projecting to the ventral hippocampus (BLA→vHPC
 PNs) are similarly involved in the neural circuit underlying avoidance behavior in females, as they are
 in males. Whole-cell patch-clamp recordings revealed that FS significantly increased the intrinsic
 excitability and the frequency (but not amplitude) of miniature excitatory postsynaptic currents (mEPSCs)
 in BLA→vHPC PNs (Response Fig. 5a-g, now Supplementary Fig. 13a-g in the revised manuscript).
 Furthermore, bilateral mifepristone infusion prior to FS exposure reversed the FS-induced enhancement
 of both intrinsic excitability and excitatory synaptic transmission in these neurons (Response Fig. 5a-g,
 now Supplementary Fig. 13a-g in the revised manuscript, with citation on page 7, lines 201-202).

 **Response Fig. 5 (related to Supplementary Fig. 13) Blocking of GR signaling in BLA mitigates FS-**
 **induced BLA→vHPC PNs overactivation in female mice.** **a** Schematic of experiment procedures. **b**
 Representative images showing mCherry expression in BLA→vHPC PNs and the cannula implantation
 in BLA. **c** Representative traces showing the firing in response to 250 pA injected current. Scale bar: 200
 103 ms, 50 mV. **d** Summary plots of action potentials (APs) number against the injected current with different
 strength (Vehicle+control: $n = 13$ neurons/4 mice; Vehicle+7d P-FS: $n = 13$ neurons/4 mice,
 Mifepristone+control: $n = 13$ neurons/4 mice; Mifepristone +7d P-FS: $n = 14$ neurons/4 mice). **e**
 Representative traces showing mEPSCs in BLA → vHPC PNs. Scale bar: 1 s, 10 pA. **f, g** Summary plots
 of averaged mEPSC frequency (**f**) and amplitude (**g**) of BLA → vHPC PNs (control+ Vehicle: $n = 14$
 neurons/3 mice; 7d P-FS+ Vehicle: $n = 14$ neurons/4 mice; control+ Mifepristone: $n = 15$ neurons/4 mice;
 7d P-FS+ Mifepristone: $n = 14$ neurons/4 mice). Data were shown as mean \pm SEM. * $p < 0.05$, ** $p < 0.01$,
 *** $p < 0.001$, ## $p < 0.01$, ### $p < 0.001$. See Supplementary Data 1 for full statistical information. Source
 data are provided as a Source Data file.

 Collectively, these results demonstrate that female mice, like males, exhibit delayed and enduring PTSD-
 like avoidance behavior following FS, mediated by GR-dependent hyperactivation of BLA→vHPC PNs.
 These additional data not only strengthen our conclusions but also broaden the translational relevance
 of our findings across sexes.

 2. The analysis of the fiber photometry is very limited.

a. Using normalized peak frequency with the baseline being pre foot shock is particularly concerning as
 there is not a correct control for the interpretation. This reviewer strongly recommends repeating this
 experiment with a no shock control for all fiber photometry experiments and not normalizing to pre shock

baseline. The interpretation of these data are currently very limited and it cannot be concluded that the
 increased activity of the BLA-vHipp pathway is from the shock and, as an example, not from increased
 handling through the experiment without a no shock control. Given that this timepoint was 3-9 days prior
 to the animal being shocked there is also a possibility that there will be changes in viral expression of
 gcamp during these times. Changes in viral expression could mean more virus reaching more cells and
 thus changes in the activity pattern of neurons and it is not recommended to use this as a normalizing
 baseline.

**Response:** We thank the reviewer for this constructive comment. Following the reviewer's suggestion,
 as well as similar comments from other reviewers, we have conducted new experiments to strengthen the
 interpretation of our fiber photometry data. Specifically, we replaced the recordings of spontaneous
 calcium events with measurements of Ca^{2+} transients in BLA PNs during behavioral tasks, the elevated
 plus maze test (EPMT) and open field test (OFT), in both control and stressed mice.

We fully agree that temporal variations in GCaMP6s expression and the absence of a proper no-shock
 control could potentially confound data interpretation. To address this, we recorded the Ca^{2+} activity of
 BLA PNs in control mice at three separate time points—prior to foot shock exposure (Pre-FS), and at 1
 137 day and 7 days post-FS—to assess possible effects of viral expression or handling. As shown in **Response**
 **Fig. 6** (now **Fig. 1h-p** in the revised manuscript, with citation on page 4, lines 104-110), control mice
 exhibited stable calcium activity in BLA PNs when entering potentially threatening situations (open arms
 or center) both at 1 and 7 days post-FS, showing no significant change relative to their pre-stress
 baseline. In contrast, FS-exposed mice displayed a pronounced increase in Ca^{2+} transients specifically
 at 7 days post-FS, with no notable change at 1 day post-FS.

 **Response Fig. 6 (related to Fig. 1h-p)** a Schematic of injection of AAV-CaMKII α -GCaMP6s into the
 BLA and implantation of optical fiber to record calcium signals. b Representative image showing
 GCaMP6s expression and the cannula implantation onto the BLA, scale bar: 200 μ m. c Time scheme of
 experimental procedures for *in vivo* fiber photometry. d-f Heatmap of fluorescence (d), mean
 fluorescence (e), and comparisons of calcium transients (f) of mice during entry into the center zone in
 OFT. ($n = 6$ mice per group). g-i Heatmap of fluorescence (g), mean fluorescence (h), and comparisons

of calcium transients (i) of mice during entry into the open arms in EPMT. ($n = 6$ mice per group). Data
 were shown as mean \pm SEM. $**p < 0.01$. See Supplementary Data 1 for full statistical information.
 Source data are provided as a Source Data file.

We next examined changes in calcium signaling at the circuit level. As shown in *Response Figure 7*
 (now *Fig. 2a-p* in the revised manuscript, with citation on page 5, lines 115-121), control mice showed
 no significant change in the activity of either BLA \rightarrow vHPC or BLA \rightarrow NAc PNs at 1 or 7 days post-FS
 relative to their own pre-FS baseline. In the FS group, however, a delayed (at 7 days post-FS)
 enhancement of calcium activity was specifically induced in the BLA \rightarrow vHPC pathway, whereas no
 concurrent change was observed in the BLA \rightarrow NAc pathway.

**Response Fig. 7 (related to Fig. 2a-p)** a Schematic showing the recording of calcium signals in
 GCaMP6s-expressing BLA \rightarrow vHPC PNs. The AAV vectors injected into BLA and vHPC were also
 shown. b Representative image showing GCaMP6s expression in BLA \rightarrow vHPC PNs and the cannula
 implantation above the BLA, scale bar: 200 μ m. c-e Heatmap of fluorescence (c), mean fluorescence (d),
 and comparisons of calcium transients (e) of mice during entry into the center zone in OFT. ($n = 6$ mice
 167 per group). f-h Heatmap of fluorescence (f), mean fluorescence (g), and comparisons of calcium

transients (**h**) of mice during entry into the open arms in EPMT. ($n = 6$ mice per group). **k-p** Same as in
(**c-h**) except that the calcium signals were recorded from BLA→NAc PNs ($n = 6$ mice per group). Data
were shown as mean \pm SEM. *** $p < 0.001$. See Supplementary Data 1 for full statistical information.
Source data are provided as a Source Data file.

*Together, these results demonstrate that FS triggers a delayed enhancement of BLA PN activity,*
*specifically within the BLA→vHPC circuit. This effect is not attributable to viral expression changes or*
*experimental handling, confirming that the observed increase in activity represents a genuine stress-*
*induced neural response.*

b. It is recommended to use a non-shocked group control to compare the average zscore at the Baseline
recording, 1 day and 7 days. (ie: mean z-score of no shock vs shock baseline, mean z-score no shock vs
shock 1 day, etc) Then use the total spikes averaged across animals in the time bin. Additionally, using a
MAD of 2 is low although not unacceptable, but limits interpretation when this is the only statistic given.
The AUC, min/max, and total count of peaks should be included as well for all fiber photometry analyses.
*Response: We thank the reviewer for these valuable suggestions. As described in our response above,*
*we have conducted additional experiments to assess the Ca²⁺ transients of BLA PNs during EPMT and*
*OFT, including appropriate non-shocked control groups for comparison at baseline, 1 day, and 7 days*
*FS. For the revised analysis, we compared both the mean z-scores and area under the curve (AUC)*
*between control and FS groups across these time points. This approach allows a more comprehensive*
*and statistically robust characterization of neural activity patterns.*

c. It is not clear where the recording is taking place. Is it a novel cage? If so is the animal exposed to this
cage before the baseline test day to eliminate artifacts from a novel cage? Is the animal re exposed to the
footshock compartment for recording? Is it homecage and the cagemates are removed? This reviewer
recommends stating clearly where the recordings are taking place because many of these options have
caveats to the interpretation of the data. How often is the animal handled before and during the
experiment, and where the animal is being recorded from is also a concern that is not addressed in the
methods.

*Response: We thank the reviewer for pointing out this important clarification. As mentioned above, we*
*have performed new experiments to monitor the calcium signals during the EPMT and OFT, rather than*
*in a novel or home cage. We have clearly specified the recording conditions in the **Methods section of***
*the revised manuscript (page 16, lines 462-474).*

**d. I recommend recording during OFT and EPM behavior and looking at peri event activity.** It
would serve the interpretation of the experiment to look at the underlying neural activity while the animal
is making the decision to enter open arms or center and look to see if the activity is different in these
zones and compare shock vs no shock.

*Response: We fully agree with the reviewer's recommendation. Accordingly, in the revised manuscript,*
*we performed new fiber photometry recordings of Ca²⁺ transients in BLA PNs during both the EPMT*
*and OFT in control and FS-exposed mice. We analyzed peri-event activity aligned to entries into the*
*open arms or center zones.*

3. For SGK D/N the authors have not clearly established the function of this virus. Although the authors

show increased mRNA of the D/N there is not an experiment to show that the Sgk1-DN is functionally
 valid. I recommend showing an experiment in which the function of Sgk1 is inert, or citations showing
 that this method has been used previously. The concern is also major because the authors did not cite
 where this virus was made or how it was made if it was made in house. More details are necessary. The
 works cited in this paragraph show using AAV-dnSGK1 but not DIO so I recommend a more detailed
 validation of this tool. A minor concern is that In Figure 6i, the authors show in 1 animal an increase in
 sgk1 expression but there is no validity that the protein expression is different across multiple mice. I
 recommend doing a fluorescence analysis in multiple animals to show protein quantification using IHC
 or a western blot analysis similar to the work that they have cited. Similar major concerns remain for
 figure 7 using the AAV-DIO-Sgk1 for over expression. Need to show that this effect is true across many
 mice, especially as the mRNA is variable and there are no statistics in the figure showing that this increase
 is statistically significant.

*Response: We thank the reviewer for their insightful comments. We agree that confirming the efficacy of*
 *both Sgk1-DN and Sgk1-OE is critical for reinforcing the reliability of our findings. In response, we have*
 *taken the following steps: First, detailed information on the viral constructs has been added to the*
 *Materials and Methods section in the revised manuscript (page 15, lines 434-435 and 459-460). Second,*
 *to directly assess the efficacy of Sgk1-DN, we reanalyzed the immunofluorescence images from BLA→*
 *vHPC PNs. Quantification of fluorescence intensity revealed a statistically significant increase in relative*
 *Sgk1 expression in the Sgk1-DN group compared to the mCherry control group (Response Fig. 8a, b,*
 *now Fig. 6i, j in the revised manuscript, with citation on page 8, lines 227-228).*

**Response Fig. 8 (related to Fig. 6i, j)** a Representative images showing expression of Sgk1(S422A)-
 mCherry and immunostaining against Sgk1 in BLA neurons. Scale bar: 200 μ m. b Summary plots
 showing Sgk1 expression in BLA→vHPC PNs (mCherry: $n = 5$ mice; Sgk1-DN: $n = 6$ mice). Data were
 shown as mean \pm SEM. ** $p < 0.01$. See Supplementary Data 1 for full statistical information. Source
 data are provided as a Source Data file.

 *Since Sgk1-DN expression does not inherently demonstrate functional inhibition, we directly assessed*
 *Sgk1 pathway activity by measuring phosphorylation of cAMP response element-binding protein (CREB)*
 *at Ser133 (pCREB)¹, a downstream target. A significant reduction in pCREB was observed in Sgk1-DN-*
 *expressing mice (Response Fig. 9, now Supplementary Fig. 15 in the revised manuscript, with citation*
 *on page 8, lines 229-231), confirming successful suppression of Sgk1 signaling.*

**Response Fig. 9 (related to Supplementary Fig. 15)** a Representative images showing expression of
 Sgk1(S422A)-mCherry and immunostaining against pCREB in BLA neurons. Scale bar: 25 μ m. **b**
 Summary plots showing pCREB expression in BLA→vHPC PNs (mCherry: $n = 5$ mice; Sgk1-DN: $n =$
 5 mice). See Supplementary Data 1 for full statistical information. Source data are provided as a Source
 Data file.

*In parallel, we also reanalyzed the immunofluorescence images from the experiments involving Sgk1*
 *overexpression in BLA→vHPC PNs. Consistent with successful viral transduction, quantification of*
 *fluorescence intensity showed a significant increase in Sgk1 expression in the Sgk1-OE group compared*
 *to controls (Response Fig. 10a, b, now Fig. 7 c, d in the revised manuscript, with citation on page 9, lines*
 *242).*

*Together, these results confirm the efficacy of both loss- and gain-of-function viral manipulations,*
 *thereby strengthening the reliability of our experimental conclusions.*

**Response Fig. 10 (related to Fig. 7 d, e)** a Representative images showing expression of Sgk1-OE-
 mCherry and immunostaining against Sgk1 in BLA neurons. Images are merged in the left panel. Scale
 261 bar: 200 μ m. **b** Summary plots showing Sgk1 expression in BLA→vHPC PNs (mCherry: $n = 5$ mice;
 Sgk1-OE: $n = 5$ mice). Data were shown as mean \pm SEM. *** $p < 0.001$. See Supplementary Data 1 for
 full statistical information. Source data are provided as a Source Data file.

**Minor Concerns**

1. The fiber photometry data are a major conclusion and should be moved to the primary figures from
 supplement fig. 6 and 8.

*Response: Thanks for the reviewer's suggestion. We have moved the fiber photometry data from*
 *Supplementary Figures 6 and 8 to the primary Figures 6p-u and 7k-p in the revised manuscript (with*
 *citation on page 9, lines 234-236 and 254-257).*

2. In figure 2 l-q, adding a schema of the timeline of when the animals are being shocked and what the
 control is would help readers to understand this experimental design. It is unclear when the animals are
 being shocked and when the tissue is harvested. It is recommended that the virus be injected at the same

time and delay the footshock so that the 1 day collected group and 7 day collected group are occurring
 on the same day (like in supplement). This delayed pattern is sometimes used throughout the paper and
 a schema would help decrease these concerns.

*Response: We sincerely appreciate this helpful suggestion. In our study, virus or beads injections were*
 *performed simultaneously in both control and stressed groups. The footshock sessions were scheduled*
 *such that the 1-day and 7-day post-shock collections occurred at the same experimental time points.*
 *Additionally, brain tissues for c-Fos staining were collected 90 minutes after exposure to the open arm*
 *of elevated plus maze. To enhance clarity, we have revised the experimental timelines in Figure 1q in the*
 *revised manuscript (with citation on page 4, lines 110-112) accordingly.*

3. In the DREADDs experiment it is recommended to have a CNO control with mCherry virus control
 *Response: Following the reviewer's and other reviewers' suggestion, we conducted additional control*
 *experiments to assess the effects of virus expression and CNO administration on mouse behavior.*
 *Specifically, we expressed mCherry-only in BLA→vHPC PNs, followed by FS exposure. Mice were then*
 *administered either vehicle or CNO intraperitoneally 30 minutes before behavioral testing (Response*
 *Fig. 11a, now Supplementary Fig. 7a in the revised manuscript). The results showed that both vehicle-*
 *and CNO-treated mice exhibited reduced time spent in the center area during OFT and decreased time*
 *and entries into open arms during EPMT following FS (Response Fig. 11b-e, now Supplementary Fig.*
 *7b-e in the revised manuscript, with citation on page 6, lines 157-159). These data confirm that the*
 *observed behavioral effects were not attributable to CNO or virus expression alone.*

**Response Fig. 11 (related to Supplementary Fig. 7) CNO produced no behavioral changes in mice**
 **expressing the control mCherry virus in BLA→vHPC PNs.** **a** Schematic of the experimental
 procedures. **b, c** Summary plots of time in center area (**b**) and total distance traveled (**c**) in OFT (Vehicle:
 control, $n = 11$ mice, 7d P-FS, $n = 10$ mice; CNO: control, $n = 12$ mice, 7d P-FS, $n = 9$ mice). **d, e**
 Summary plots of time in open arms (**d**) and open arm entries in EPMT (Vehicle+control, $n = 11$ mice,
 Vehicle+7d P-FS, $n = 10$ mice; CNO+control, $n = 12$ mice, CNO+7d P-FS, $n = 9$ mice). Data were shown
 as mean \pm SEM. * $p < 0.05$, ** $p < 0.01$, *** $p < 0.001$. See Supplementary Data 1 for full statistical
 information. Source data are provided as a Source Data file.

4. The sub threshold FS in figure 7 is 0.5mA 120 shocks/day for 2 days with no duration or interval given.
 Why was this paradigm chosen over the 0.3mA shown in supplement. Need clearer rationale.

*Response: We apologize for not specifying the foot shock parameters in the original manuscript. The 0.5*
 *mA intensity was selected based on the preliminary data showing that two-day exposure induced*

*persistent avoidance (7-14 days), a core PTSD-like phenotype (Supplementary Figure 1 in the revised*
*manuscript, with citation on page 3, lines 77-80), whereas one-day exposure caused only short-term*
*effects (Supplementary Figure 2 in the revised manuscript, with citation on page 3, lines 80-81), and 0.3*
*mA had minimal impact (Supplementary Figure 3 in the revised manuscript, with citation on page 4,*
*lines 82-83).*

*To establish the subthreshold stress model in Figure 7e (with citation on page 9, lines 248-249), we*
*followed the conventional approach of reducing the total stress exposure (i.e., shortening the duration)*
*while maintaining the stimulus intensity. Considering that a two-day paradigm is already relatively short,*
*we instead reduced the number of shocks per day to lower the overall stress load. Specifically, we kept*
*the intensity at 0.5 mA and the two-day schedule unchanged but reduced the daily shocks from 360 to*
*120. This adjustment yielded subthreshold effects without inducing avoidance behavior, while*
*minimizing procedural variability by altering only the shock numbers.*

*As clarified in the revised manuscript (page 9, lines 248-253), our stepwise validation confirmed that*
*this protocol —0.5 mA intensity with reduced shock number —represents an appropriate and*
*reproducible subthreshold stress paradigm.*

5. In the graphic figure 8, the sgk1 expression in the BLA-NAc is a highlight of the figure but this is
supplement data. I recommend either making this a main text figure if it is the key conclusion, or
removing it from the graphic.

*Response: We appreciate this thoughtful suggestion. Accordingly, we have removed the depiction of*
*BLA→NAc PNs from the graphical summary in the revised manuscript (page 36, lines 954-958) to*
*maintain consistency with the main text data presentation.*

6. In the abstract authors use abbreviation PN without defining

*Response: We thank the reviewer for noting this oversight. We have corrected it accordingly (page 2,*
*line 27).*

Figure 5 beginning of 6 is a highlight of the paper and shows an interesting mechanism underlying the
avoidance behaviors that the authors describe. This reviewer also acknowledges that there is a strength
to the breadth of techniques shown in the manuscript.

*Response: We thank the reviewer for their positive feedback and for highlighting the significance of*
*these findings.*

**Reviewer #2 (Remarks to the Author):**

Remarks to the Authors

This manuscript combines multiple experimental techniques (behavioral assays, fiber photometry,
chemogenetics, and ex vivo electrophysiology) to investigate how BLA circuit alterations underlie
elevated anxiety in a mouse model of PTSD. The authors report that following footshock stress, BLA-
vHPC neurons exhibit increased spontaneous activity in vivo and heightened excitability ex vivo. They
further show that glucocorticoid receptor activation during stress is necessary for both the behavioral
phenotype and BLA-vHPC activity changes. Moreover, they identify Sgk1 as being upregulated in BLA-
vHPC neurons and bidirectional manipulations of Sgk1 suggest that Sgk1 levels modulate BLA-vHPC
neuronal excitability and anxiety in stress-exposed mice. These findings address a significant question
in the field—how stress during trauma leads to persistent circuit-level alterations that drive long-term
fear and anxiety. The manuscript is generally well-written and organized, and the figures clearly present
the results. However, some of the experiments and analyses do not directly address the relevant questions
leading to overstated conclusions. In particular, the fiber photometry experiments would benefit from
more rigorous design and standard analytical approaches, which currently undermine the overall
interpretation.

**Major Comments**

1) The authors do not describe the procedures for their fiber photometry recordings and cFos experiments,
only stating that they measured “spontaneous activity” presumably at rest. While differences in resting
activity can be informative, they do not directly relate to the state-dependent anxiety behavior in the OFT
and EPM. Consequently, it is not justified to conclude that increased BLA-vHPC activity underlies the
increased anxiety. To address this, the authors should replicate their photometry and cFos findings during
the OFT and EPM. For the photometry experiments, they should correlate neural activity dynamics with
real-time behavior (e.g. exploration of anxiogenic vs. non-anxiogenic zones).

*Response: We sincerely thank the reviewer for these valuable comments. In our study, the brain tissues*
*for c-Fos staining were collected 90 minutes after exposure to the open arm of elevated plus maze, and*
*we have clarified this procedure in the revised Materials and Methods section (page 17, lines 478-481).*
*Additionally, the experimental timelines have been updated in Figure 1q in the revised manuscript to*
*explicitly indicate the timing of tissue collection.*

*In line with the reviewer’s recommendation, we performed new fiber photometry experiments to monitor*
*Ca²⁺ transients of BLA PNs during the potentially threatening situations (open arm of the EPM and*
*center area of open field, replacing the previously reported recordings of spontaneous activity. As shown*
*in Response Fig. 12 (now Fig. 1h-p in the revised manuscript, with citation on page 4, lines 104-110),*
*control mice exhibited stable calcium activity in BLA PNs when entering potentially threatening*
*situations (open arms or center) both at 1 and 7 days post-FS, showing no significant change relative to*
*their pre-stress baseline. In contrast, FS-exposed mice displayed a pronounced increase in Ca²⁺*
*transients specifically at 7 days post-FS, with no notable change at 1 day post-FS.*

**Response Fig. 12 (related to Fig. 1h-p)** a Schematic of injection of AAV-CaMKII α -GCaMP6s into the
 BLA and implantation of optical fiber to record calcium signals. b Representative image showing
 GCaMP6s expression and the cannula implantation onto the BLA, scale bar: 200 μ m. c Time scheme of
 experimental procedures for *in vivo* fiber photometry. d-f Heatmap of fluorescence (d), mean
 fluorescence (e), and comparisons of calcium transients (f) of mice during entry into the center zone in
 OFT. ($n = 6$ mice per group). g-i Heatmap of fluorescence (g), mean fluorescence (h), and comparisons
 of calcium transients (i) of mice during entry into the open arms in EPMT. ($n = 6$ mice per group). Data
 were shown as mean \pm SEM. ** $p < 0.01$. See Supplementary Data 1 for full statistical information.
 Source data are provided as a Source Data file.

*We further examined the activity of projection-specific neuronal populations. As shown in Response Fig.*
 *13 (now Fig. 2a-p in the revised manuscript, with citation on page 5, lines 115-121), in control mice, the*
 *evoked calcium signals showed no notable differences across the pre-FS, 1-day, and 7-day time points*
 *in either BLA \rightarrow vHPC or BLA \rightarrow Nac PNs. In contrast, FS selectively induced a delayed enhancement of*
 *calcium transients specifically in BLA \rightarrow vHPC PNs, whereas BLA \rightarrow Nac PNs remained unaltered.*

Figure 2

**Response Fig. 13 (related to Fig. 2a-p)** a Schematic showing the recording of calcium signals in
 GCaMP6s-expressing BLA→vHPC PN. The AAV vectors injected into BLA and vHPC were also
 shown. b Representative image showing GCaMP6s expression in BLA→vHPC PN and the cannula
 implantation above the BLA, scale bar: 200 μ m. c-e Heatmap of fluorescence (c), mean fluorescence (d),
 and comparisons of calcium transients (e) of mice during entry into the center zone in OFT. ($n = 6$ mice
 401 per group). f-h Heatmap of fluorescence (f), mean fluorescence (g), and comparisons of calcium
 transients (h) of mice during entry into the open arms in EPMT. ($n = 6$ mice per group). k-p Same as in
 (c-h) except that the calcium signals were recorded from BLA→NAc PN ($n = 6$ mice per group). Data
 were shown as mean \pm SEM. *** $p < 0.001$. See Supplementary Data 1 for full statistical information.
 Source data are provided as a Source Data file.

Furthermore, as shown in Response Fig. 14 (now Fig. 6p-u in the revised manuscript, with citation on
 page 9, lines 234-236) and Response Fig. 15 (now Fig. 7k-p in the revised manuscript, with citation on
 page 9, lines 254-257), overexpression of dominant-negative *Sgk1* (*Sgk1*-DN) in BLA→vHPC PN
 effectively prevented the FS-induced increase in Ca^{2+} events during the OFT and EPMT. Conversely,
 overexpression of wild-type *Sgk1* (*Sgk1*-OE) in this subpopulation significantly enhanced Ca^{2+} transients

*under the subthreshold FS condition.*

**Response Fig. 14 (related to Fig. 6p-u)** a-c Heatmap of fluorescence (a), mean fluorescence (b), and
 comparisons of calcium transients (c) of mice during entry into the center zone in OFT ($n = 6$ mice per
 group). d-f Heatmap of fluorescence (d), mean fluorescence (e), and comparisons of calcium transients
 (f) of mice during entry into the open arms in EPMT ($n = 6$ mice per group). Data were shown as mean
 \pm SEM. $**p < 0.01$. See Supplementary Data 1 for full statistical information. Source data are provided
 as a Source Data file.

**Response Fig. 15 (related to Fig. 7k-p)** a-c Heatmap of fluorescence (a), mean fluorescence (b), and
 comparisons of calcium transients (c) of mice during entry into the center zone in OFT. ($n = 6$ mice per
 group). d-f Heatmap of fluorescence (d), mean fluorescence (e), and comparisons of calcium transients
 (f) of mice during entry into the open arms in EPMT. ($n = 6$ mice per group). Data were shown as mean
 \pm SEM. $**p < 0.01$. See Supplementary Data 1 for full statistical information. Source data are provided
 as a Source Data file.

2) The photometry experiments employ a within-subjects design in which the same mice are recorded at
 Pre-FS, 1 day Post-FS, and 7 days Post-FS. This approach does not adequately control for time-dependent
 changes in neural activity that may occur regardless of PTSD induction. To address this, the authors
 should adopt a between-subjects design that includes an experience-matched control group, ideally mice
 exposed to a the subthreshold-FS protocol (as in Fig. 7). This group would help determine whether any
 observed changes in neural activity truly stem from PTSD induction rather than changes in activity over
 time or due to footshock exposure.

**Response:** *We fully agree that including an experience-matched control group is important to distinguish*

PTSD-specific changes from time-dependent fluctuations in neural activity. To address this concern, we
 conducted new experiments examining the activity of BLA PNs in control mice at three separate time
 points. These results are presented in *Response Fig. 16* and *Response Fig. 17*.
 As shown in *Fig. 16* (now *Fig. 1h-p* in the revised manuscript, with citation on page 4, lines 104-110),
 fiber photometry recordings in control mice revealed no significant changes in Ca^{2+} transients of BLA
 PNs at Pre-FS, 1 day post-FS, or 7 days post-FS when the mice entered the open arms of the EPM or the
 center of the OF. In contrast, FS-stressed mice exhibited a selective and delayed increase in Ca^{2+}
 transients at 7 days post-FS, with no notable change at 1 day post-FS.

 **Response Fig. 16 (related to Fig. 1h-p)** a Schematic of injection of AAV-CaMKII α -GCaMP6s into the
 BLA and implantation of optical fiber to record calcium signals. b Representative image showing
 GCaMP6s expression and the cannula implantation onto the BLA, scale bar: 200 μ m. c Time scheme of
 experimental procedures for *in vivo* fiber photometry. d-f Heatmap of fluorescence (d), mean
 fluorescence (e), and comparisons of calcium transients (f) of mice during entry into the center zone in
 OFT. ($n = 6$ mice per group). g-i Heatmap of fluorescence (g), mean fluorescence (h), and comparisons
 of calcium transients (i) of mice during entry into the open arms in EPMT. ($n = 6$ mice per group). Data
 were shown as mean \pm SEM. $**p < 0.01$. See Supplementary Data 1 for full statistical information.
 Source data are provided as a Source Data file.

 Similarly, in *Response Fig. 17* (now *Fig. 2a-p* in the revised manuscript, with citation on page 5, lines
 115-121), activity of projection-specific populations showed that control mice had no significant changes
 in either BLA \rightarrow vHPC or BLA \rightarrow NAc PNs across the three time points during entries into the open arms
 of the EPM or the center of the OF. FS, however, specifically induced a delayed increase in Ca^{2+} events
 in BLA \rightarrow vHPC PNs, while BLA \rightarrow NAc PNs remained unaffected.

**Response Fig. 17 (related to Fig. 2a-p)** a Schematic showing the recording of calcium signals in
 GCaMP6s-expressing BLA→vHPC PN. The AAV vectors injected into BLA and vHPC were also
 shown. b Representative image showing GCaMP6s expression in BLA→vHPC PN and the cannula
 implantation above the BLA, scale bar: 200 μm. c-e Heatmap of fluorescence (c), mean fluorescence (d),
 and comparisons of calcium transients (e) of mice during entry into the center zone in OFT. (*n* = 6 mice
 465 per group). f-h Heatmap of fluorescence (f), mean fluorescence (g), and comparisons of calcium
 transients (h) of mice during entry into the open arms in EPMT. (*n* = 6 mice per group). k-p Same as in
 (c-h) except that the calcium signals were recorded from BLA→NAc PN (*n* = 6 mice per group). Data
 were shown as mean ± SEM. ****p* < 0.001. See Supplementary Data 1 for full statistical information.
 Source data are provided as a Source Data file.

 3) The photometry data require more rigorous analysis. It appears that the authors used the raw
 fluorescence rather than computing the %dF/F₀ which is standard practice to account for baseline
 differences across animals or sessions. More critically, they normalized the Post-FS conditions by the
 Pre-FS condition effectively eliminating variance in the Pre-FS condition and obscuring the true
 variability in the Post-FS data. Furthermore, the statistical tests applied to this data, one-way ANOVA
 and t-tests, are inappropriate since the Pre-FS condition was used for baseline normalization which
 violates model assumptions. The authors should replace these plots with the unnormalized data (Fig 1.

479 L,M; Fig. 2 E-K, Suppl Fig. 6., Supp Fig. 8). Furthermore, the authors should supplement the calcium
 event analysis by presenting the % dF/F0 (e.g. mean %dF/F0 in each condition, time-locked to behavior
 in anxiety tests).

*Response: Following the reviewer's suggestion, we have performed new experiments to investigate the*
 *Ca²⁺ transients of BLA PNs during EPMT and OFT by fiber photometry. In the revised manuscript, all*
 *photometry data are presented as z-score, with the area under the curve (AUC) of the z-scored traces*
 *quantified to provide a robust measure of neural activity. These analyses are shown in Figures 1i-p (with*
 *citation on page 4, lines 104-110), 2a-p (with citation on page 5, lines 115-121), 6p-u (with citation on*
 *page 9, lines 234-236), and 7k-p (with citation on page 9, lines 254-257) in the revised manuscript.*

 4) The authors assert that Sgk1-DN prevents, and Sgk1-overexpression increases susceptibility to FS-
 induced hyperactivity and anxiety. However, the statistical comparisons and some of the results of the
 Sgk1 experiments do not strongly support these claims. The authors mainly compared mice receiving
 FS-exposure+Sgk1 manipulation versus no-FS+Sgk1 manipulation, which does not directly test how
 Sgk1 manipulations affect FS-exposed mice. The critical comparison should be FS-exposure+mCherry
 versus FS-exposure+Sgk1 manipulation. In several plots, this comparison is not significant (Fig. 6m-n
 EPM behavior) and entirely not significant for the Sgk1-overexpression experiment in Fig.7, weakening
 the claim that Sgk1 levels are important for the behavioral and electrophysiological effects of footshock.
 Furthermore, in Fig.6c the authors report elevated Sgk1 mRNA. However, the statistical testing relied on
 multiple t-tests without correcting for multiple comparisons which calls into question the robustness of
 that result.

*Response: We sincerely thank the reviewer for these valuable and constructive suggestions. We fully*
 *agree that addressing these concerns is critical to strengthen the conclusions regarding the role of Sgk1*
 *in FS-induced behavioral and electrophysiological effects.*

*To directly assess how Sgk1 manipulations affect FS-exposed mice, and according to the reviewer's and*
 *reviewer4's suggestions, we repeated the experiments by examining whether Sgk1-DN or Sgk1*
 *overexpression can respectively prevent or enhance the susceptibility of BLA→vHPC PNs to stress*
 *exposure.*

*For Sgk1-DN, our new results showed that FS induced a significant reduction in time spent in the center*
 *of the open field and in the open arms of the elevated plus maze, as well as fewer entries into the open*
 *arms, in mCherry-expressing mice (Response Fig. 18, now Fig. 6l-o in the revised manuscript, with*
 *citation on pages 8-9, lines 231-232). Importantly, these FS-induced behavioral changes were effectively*
 *blocked by overexpression of Sgk1-DN, with Sgk1-DN-expressing FS mice spending significantly more*
 *time in the center and open arms and making more open arm entries compared with FS-exposed mCherry*
 *controls (Response Fig. 18, now Fig. 6l-o in the revised manuscript, with citation on pages 8-9, lines*
 *231-232). These results indicate that Sgk1 activity is critical for mediating the delayed increase in*
 *avoidance behavior following FS.*

Response Fig. 18 (related to Fig. 6l-o) a, b Summary plots of time in center area (a) and total distance

traveled (b) in OFT (mCherry+control: $n = 12$ mice; mCherry +7d P-FS: $n = 13$ mice; Sgk1-DN+control:
 $n = 12$ mice; Sgk1-DN +7d P-FS: $n = 12$ mice). c, d Summary plots of time in open arms (c) and open-
 arm entries (d) in EPMT, sample size as in panel (a, b). Data were shown as mean \pm SEM. $*p < 0.05$,
 $**p < 0.01$, $***p < 0.001$. See Supplementary Data 1 for full statistical information. Source data are
 provided as a Source Data file.

For Sgk1 overexpression, subthreshold footshock (sub-FS) exposure in Sgk1-overexpressing mice
 significantly promoted avoidance-like behavior, as evidenced by reduced center time in the open field
 test and decreased open-arm time and entries in the elevated plus maze (Response Fig. 19a-d, now Fig.
 7g-j in the revised manuscript, with citation on page 9, lines 249-252). Notably, this effect was specific
 to the stress challenge, as Sgk1 overexpression alone under baseline conditions did not alter behavior.
 Crucially, under the sub-FS condition, Sgk1-overexpressing mice showed a marked increase in
 avoidance responses compared to mCherry-control mice (Response Fig. 19a-d, now Fig. 7g-j in the
 revised manuscript, with citation on page 9, lines 249-252). These results indicate that Sgk1
 overexpression selectively enhances the susceptibility of BLA \rightarrow vHPC PNs to subthreshold stress,
 thereby predisposing individuals to develop pathological avoidance behavior.

We appreciate the reviewer's attention to this detail. The presentation of the Sgk1 mRNA statistical
 analysis has been revised for greater clarity and accuracy in the revised manuscript Fig. 6c and
 Supplementary Fig. 14c.

**Response Fig. 19 (related to Fig. 7g-j)** a, b Summary plots of time in center area (a) and total distance
 traveled (b) in OFT (mCherry+control: $n = 11$ mice; mCherry +7d P-sub-FS: $n = 13$ mice; Sgk1-
 OE+control: $n = 13$ mice; Sgk1-OE +7d P-sub-FS: $n = 12$ mice). c, d Summary plots of time in open
 arms (c) and open arm entries (d) in EPMT, sample size as in panel (a, b). Data were shown as mean \pm
 SEM. $*p < 0.05$, $**p < 0.01$, $***p < 0.001$. See Supplementary Data 1 for full statistical information.
 Source data are provided as a Source Data file.

Minor Comments

5) The introduction only describes the clinical literature and is missing the relevant findings in rodents
 that justify the study's experiments.

**Response:** We thank the reviewer for this suggestion. Accordingly, we have added relevant findings from
 rodent studies to the Introduction section in the revised manuscript (page 3, lines 55-60).

6) It remains unclear whether the inescapable footshock primarily models general stress-induced anxiety
 rather than a distinct PTSD-like condition in which fear memory impairments are central. The authors

should conduct fear conditioning and extinction at the 7-day time point and demonstrate an extinction
retention deficit in this model.

*Response:* We thank the reviewer for this insightful suggestion. To directly address whether our
footshock (FS) paradigm models specific PTSD-like memory impairments, we conducted a series of
complementary fear conditioning and extinction experiments. First, we assessed contextual fear memory
by re-exposing mice to the original FS context. FS mice exhibited significantly enhanced freezing levels
at both 1 and 7 days post-stress compared to controls (*Response Fig. 20, now Supplementary Fig. 5 in*
*the revised manuscript, with citation on page 4, lines 96-101*), indicating a potentiation of contextual
fear.

**Response Fig. 20 (related to Supplementary Fig. 5) Footshock stress causes elevated fear memory**
**in male mice. a** Time scheme for contextual fear conditioning (CFC) test. **b** Summary plots of freezing
level of mice in CFC test (Control: $n = 12$ mice; 1d P-FS: $n = 12$ mice; 7d P-FS: $n = 13$ mice). Data were
shown as mean \pm SEM. *** $p < 0.001$. See Supplementary Data 1 for full statistical information. Source
data are provided as a Source Data file.

*Second, to examine the effects of FS on the fear extinction in mice, we subjected all the mice to a fear-*
*conditioning and extinction procedure (Response Fig. 21a). Consistent with the contextual fear results,*
*FS mice exhibited higher freezing levels in response to the cued tone (CS) during the habituation phase.*
*During the fear acquisition session, the control and FS mice displayed a robust increase in the*
*conditioned fear response, which was more pronounced in FS mice (Response Fig. 21b). However, we*
*found that FS significantly hindered extinction (Response Fig. 21c), evidenced by lower the Extinction-*
*Retention Index (ERI) and Extinction-Learning Index (ELI)^{2,3} in FS mice (Response Fig. 21d, e),*
*although the two groups mice showed comparable freezing levels in response to the first two CS within*
*first extinction session. Collectively, these results demonstrate that FS significantly enhances fear levels*
*and impairs fear extinction in mice.*

**Response Fig. 21 Footshock stress impaired fear extinction and extinction memory retention in**
 **male mice. a** Time scheme for fear conditioning and extinction. **b, c** Summary plots of freezing level of
 mice in response to CS during habituation, fear conditioning (**b**), fear extinction, and retrieval test (**c**)
 (Control: $n = 15$ mice; 7d P-FS: $n = 13$ mice). **d, e** Summary plots of ELI (**d**) and ERI (**e**) of mice during
 fear extinction and retrieval test. Same sample size as in **b, c**. Data were shown as mean \pm SEM. $*p <$
 0.05 , $**p < 0.01$, $***p < 0.001$.

7) In the cFos experiments (Fig. 2M-Q), the fewer double-labeled cells in the BLA-NAc condition could
 be partially explained by fewer cFos+ cells overall. The authors should compare the density of cFos+
 cells between tracing conditions to rule this out.

**Response:** We appreciate the reviewer's careful consideration. To rule out the potential confounder of
 overall c-Fos activity, we have conducted an additional quantitative analysis of the total c-Fos+ cell
 density in the BLA across the experimental groups. This new analysis, now included in the revised
 manuscript, demonstrates that the global level of neuronal activation was comparable between the
 BLA \rightarrow vHPC and BLA \rightarrow NAc conditions at day 7 post-FS (**Response Fig. 22, now Fig. 2s, w in the revised**
 **manuscript**). Consequently, the fewer double-labeled cells observed in the BLA \rightarrow NAc pathway can be
 confidently attributed to a lower specific recruitment of these projection neurons by the aversive
 experience, rather than to a baseline difference in c-Fos expression.

**Response Fig. 22 (related to Fig. 2s, w) a** Summary plots of c-Fos+ cells expression in BLA of
 BLA \rightarrow vHPC PN labelled mice ($n = 6$ mice per group). **b** Summary plots of c-Fos+ cells expression in
 BLA of BLA \rightarrow NAc PN labelled mice (Control: $n = 6$ mice; 1d P-FS: $n = 6$ mice; 7d P-FS: $n = 5$ mice).
 Data were shown as mean \pm SEM. $*p < 0.05$, $***p < 0.001$. See Supplementary Data 1 for full statistical
 information. Source data are provided as a Source Data file.

8) In Fig. 4, the authors administered either vehicle or CNO prior to OFT and EPM testing and observed
reduced avoidance behavior in the CNO group. Previous studies indicate that inhibiting the BLA-vHPC
circuit can reduce anxiety-like behavior in healthy mice, so it is unclear if the observed reduction reflects
a general anxiolytic effect or a restoration of normal anxiety levels. Also, because CNO (through its
metabolite clozapine) is known to have off-target effects, it is essential that the authors administer CNO
to the control groups. A more rigorous design would include 2 mCherry-expressing control groups that
both receive CNO: one subjected to the footshock protocol and one naive. Additionally, to substantiate
the claim that BLA-vHPC hyperactivity underlies the increased anxiety, they should demonstrate
specificity by confirming that inhibiting BLA-vHPC neurons but not BLA-NAc neurons in FS mice
normalizes anxiety.

*Response: We thank the reviewer for these insightful comments and constructive suggestions. We have
conducted additional experiments to address the points raised, as detailed below.*

**a. Specificity of the behavioral effect:**

*To rule out potential off-target effects of CNO and to establish a more robust baseline, we performed
new control experiments. We expressed mCherry (control virus) in BLA→vHPC PNs and subjected these
mice to footshock (FS) (Response Fig. 23a, now Supplementary Fig. 7a in the revised manuscript). When
treated with CNO, these mice still exhibited significant avoidance behavior in both OFT and EPMT,
comparable to the vehicle-treated FS group (Response Fig. 23b-e, now Supplementary Fig. 7b-e in the
revised manuscript, with citation on page 6, lines 157-159). This result confirms that CNO itself does not
produce a general anxiolytic effect in our paradigm. In contrast, chemogenetic inhibition of BLA→vHPC
PNs in FS mice specifically reversed the pathological avoidance behavior, restoring performance to
levels indistinguishable from unstressed controls. This pattern of results indicates that the intervention
does not merely reduce anxiety but normalizes it by reversing the FS-induced pathological state.*

**b. Circuit-specificity of the effect:**

*Regarding the necessity of inhibiting the BLA→NAc pathway, our data do not support its involvement in
the observed phenotype. We found no significant changes in the activity of BLA→NAc PNs following FS,
as assessed by fiber photometry, c-Fos expression, intrinsic excitability, and synaptic transmission.
Given the absence of FS-induced hyperactivation in this pathway, we concluded that there is no
compelling rationale for targeting BLA→NAc neurons to reverse a behavioral deficit that they do not
underlie.*

**Response Fig. 23 (related to Supplementary Fig. 7) CNO produced no behavioral changes in mice**
**expressing the control mCherry virus in BLA→vHPC PNs. a** Schematic of the experimental

procedures. **b, c** Summary plots of time in center area (**b**) and total distance traveled (**c**) in OFT (Vehicle:
control, $n = 11$ mice, 7d P-FS, $n = 10$ mice; CNO: control, $n = 12$ mice, 7d P-FS, $n = 9$ mice). **d, e**
Summary plots of time in open arms (**d**) and open arm entries (**e**) in EPMT (Vehicle+control, $n = 11$ mice,
Vehicle+7d P-FS, $n = 10$ mice; CNO+control, $n = 12$ mice, CNO+7d P-FS, $n = 9$ mice). Data were shown
as mean \pm SEM. $*p < 0.05$, $**p < 0.01$, $***p < 0.001$. See Supplementary Data 1 for full statistical
information. Source data are provided as a Source Data file.

9) In Fig. 5, mifepristone but not spironolactone administration during FS was found to block subsequent
behavioral and electrophysiological changes. This result motivates the subsequent exploration of
glucocorticoid-associated mechanisms within BLA-vHPC neurons triggered by FS. However, because
these drugs were provided systemically, it is impossible to determine whether their effects are mediated
by actions at the amygdala or elsewhere in the brain and periphery. To strengthen the mechanistic link
between glucocorticoid receptor activation and the upregulation of Sgk1 in BLA-vHPC neurons, the
authors should perform local microinfusions of mifepristone into the BLA during FS and assess its impact
on the emergence of behavioral and electrophysiological phenotypes.

*Response: We thank the reviewer for this insightful suggestion. Accordingly, we performed local*
*microinfusions of mifepristone into the BLA prior to FS to directly determine whether glucocorticoid*
*receptor (GR) signaling within the BLA mediates the stress effects (Response Fig. 24a, b, now Fig. 5c, d*
*in the revised manuscript). Our behavioral results demonstrate that intra-BLA infusion of mifepristone*
*significantly attenuated the avoidance-like behavior induced by FS (Response Fig. 24c-f, now Fig. 5e-h*
*in the revised manuscript, with citation on page 7, lines 189-193). Electrophysiologically, it blocked the*
*FS-induced hyperexcitability and increased synaptic excitation in BLA→vHPC PNs (Response Fig. 24g-*
*m, now Fig. 5i-o in the revised manuscript, with citation on page 7, lines 195-197). These results confirm*
*that GR activation within the BLA is necessary for the stress-induced phenotypic changes, thereby*
*strengthening the proposed mechanistic link.*

**Response Fig. 24 (related to Fig. 5) Blocking of GR signaling within BLA mitigates FS-induced**
 **increase in avoidance behavior and BLA→vHPC PN overactivation in male mice.** **a** Schematic of
 experiment procedures. **b** Representative images showing mCherry expression in BLA→vHPC PN and
 the cannula implantation in BLA. **c, d** Summary plots of time in center area (**c**) and total distance traveled
 (**d**) in OFT (Vehicle+control: $n = 12$ mice; Vehicle+7d P-FS: $n = 14$ mice, Mifepristone+control: $n = 12$
 mice; Mifepristone +7d P-FS: $n = 12$ mice). **e, f** Summary plots of time in open arms (**e**) and open arm
 entries (**f**) in EPMT, sample size as in panel (**c, d**). **g** Schematic of experiment procedures. **h**
 Representative images showing mCherry expression in BLA→vHPC PN and the cannula implantation
 in BLA. **i** Representative traces showing the firing in response to 250 pA injected current. Scale bar: 200
 675 ms, 30 mV. **j** Summary plots of action potentials (APs) number against the injected current with different
 strength (Vehicle+control: $n = 16$ neurons/4 mice; Vehicle+7d P-FS: $n = 14$ neurons/4 mice,
 Mifepristone+control: $n = 13$ neurons/4 mice; Mifepristone +7d P-FS: $n = 14$ neurons/4 mice). **k**
 Representative traces showing mEPSCs in BLA → vHPC PN. Scale bar: 500 ms, 10 pA. **l, m** Summary
 plots of averaged mEPSC frequency (**l**) and amplitude (**m**) of BLA → vHPC PN (control+ Vehicle: n
 = 14 neurons/3 mice; 7d P-FS+ Vehicle: $n = 14$ neurons/4 mice; control+ Mifepristone: $n = 14$ neurons/4
 mice; 7d P-FS+ Mifepristone: $n = 13$ neurons/4 mice). Data were shown as mean \pm SEM. $**p < 0.01$,
 $***p < 0.001$, $\#p < 0.05$, $##p < 0.01$, $###p < 0.001$. See Supplementary Data 1 for full statistical
 information. Source data are provided as a Source Data file.

10) In line 105-106, the statement that BLA-vHPC neurons are preferentially engaged in aversive

behaviors is not entirely accurate based on previous studies (Pi et al. 2020, Nat Comm; Beyeler et al.
2016 Neuron).

*Response: We appreciate the reviewer for pointing out this necessary clarification. Accordingly, we have*
*rewritten the relevant sentence and have added the cited studies (Pi et al., 2020; Beyeler et al., 2016) to*
*provide a more comprehensive and accurate description, as guided by the reviewer (revised manuscript,*
*page 5, lines 115-118).*

11) What is the difference in experimental conditions between the control data presented in Fig. 7g-j and
Suppl Fig. 7?

*Response: We apologize for confusion regarding the control data. The experimental conditions for the*
*control groups in Fig. 7g-j (with citation on page 9, lines 249-252) and Supplementary Fig. 7 (now*
*Supplementary Fig. 16 in the revised manuscript, with citation on page 9, lines 242-245) were indeed*
*identical.*

*Our investigation proceeded in two steps. First, we examined whether Sgk1 overexpression alone, under*
*physiological conditions, affects behavior. As shown in Supplementary Fig. 16, we found that Sgk1*
*overexpression in BLA→vHPC PNs did not alter avoidance-like behavior in naive mice.*

*Based on this null result, we then tested the specific hypothesis that Sgk1 overexpression enhances*
*susceptibility to stress. In this subsequent experiment, we overexpressed Sgk1 in BLA→vHPC PNs and*
*exposed the mice to subthreshold footshock (Sub-FS). The critical comparison here is between mCherry*
*and Sgk1-overexpressing mice under Sub-FS conditions. We observed that Sub-FS did not induce*
*behavioral changes in mCherry-control mice, but it significantly increased avoidance behavior in mice*
*overexpressing Sgk1. This key finding demonstrates that Sgk1 overexpression specifically potentiates the*
*behavioral impact of subthreshold stress.*

12) Fig.6P and 7L should be analyzed with 3-way ANOVA (stress, protein, current factors).

*Response: We are grateful to the reviewer for raising this important point regarding statistical analysis.*
*Accordingly, we have re-analyzed the data in Fig. 6P and 7L using a three-way ANOVA. The updated*
*analysis further supports our original conclusions, and we have revised the figures section accordingly*
*(Fig. 6w and Fig.7r in the revised manuscript).*

13) The authors have not provided critical details of their fiber photometry experiments. They should
describe in the methods their optical setup, sampling rate, light intensity, how raw signals were
filtered/transformed, how z-scores were taken and any further normalization.

what the experimental procedures were for the spontaneous recordings ie. arena, handling, recording
protocol, and duration. If the recording duration was 1 minute as presented in the figures, then this is an
insufficient amount of sampling time to draw conclusions about resting state activity.

*Response: We sincerely thank the reviewer for the valuable comments. According to the constructive*
*suggestions from this reviewer and the others, we have conducted new experiments to monitor the Ca²⁺*
*transients of BLA neurons during EPMT and OFT, thereby replacing the original spontaneous calcium*
*event recordings in the revised manuscript. Additionally, we have now included detailed information of*
*fiber photometry methodology in the Materials and Methods section in the revised manuscript (page 16,*
*lines 462-474).*

14) In line 334, they state 4-10 week old mice were used. Since mice reach adulthood at 8 weeks, did the

authors perform behavioral tests on adolescents?

*Response: We sincerely apologize for this oversight. The span of 4-10 week old reflected the entire*
*experiment timeline, encompassing viral injection, behavioral testing, and fiber photometry recordings.*
*We acknowledge that our initial description was insufficiently precise. In response, we have now*
*explicitly specified the exact age of the animals at the onset of each specific experimental procedure in*
*the revised Materials and Methods section (page 14, lines 386-389).*

15) For the anxiety tests, they should describe the lighting conditions since brightness correlates with
avoidance. For the EPM, they should provide the height of the maze.

*Response: We sincerely apologize for this oversight. In our experiments, all behavioral tests were indeed*
*conducted under red light conditions to control for brightness. This information, along with the exact*
*elevation of the elevated plus maze (EPM) above the floor, has now been explicitly stated in the revised*
*Materials and Methods section (page 14, lines 404, 410 and 411).*

16) In line 467, “concentration” should be “dose”.

*Response: We thank the reviewer for the careful observation. The term "concentration" has been*
*corrected to "dose" in the revised manuscript (page 19, line 544).*

17) In the methods for RNAscope, they should describe how cells were selected, how many cells per
mouse, and how they ensured consistent selection between groups.

*Response: We have added the detailed information of RNAscope to the Materials and Methods section*
*in the revised manuscript (page 21, lines 607-612).*

18) The authors have not provided a description of the procedures for the cFos experiments in the
methods.

*Response: We appreciate the reviewer’s careful reading. The detailed protocol for c-Fos staining,*
*including the specific tissue collection timepoint (90 min post open arm exposure), has now been added*
*to the Materials and Methods section (page 17, lines 478-481), and Figure 1q in the revised manuscript*
*has been updated to reflect this timeline.*

**Reviewer #3 (Remarks to the Author):**

The authors used a mouse model of PTSD induced by inescapable footshock (FS) stress to identify a
delayed yet persistent escalation in avoidance behavior, which was specifically associated with the
activation of basolateral amygdala (BLA) neurons projecting to the ventral hippocampus (BLA→vHPC
PNs), but not those targeting the nucleus accumbens. This projection-specific activation arises from
enhanced excitability and increased excitatory transmission in BLA→vHPC PNs, mediated by
glucocorticoid receptor (GR) signaling. They also discovered that upregulation of serum- and
glucocorticoid-regulated kinase 1 (Sgk1) acts as a crucial regulator of FS-induced hyperactivation in
BLA→vHPC PNs and the consequent development of PTSD-like avoidance behaviors. These results
suggest a potential role of projection-specific Sgk1 upregulation in BLA neurons in the pathophysiology
of PTSD-like avoidance behavior. While the results are intriguing, I have several concerns:

1. Regarding the animal models of PTSD, avoidance behavior is one of the behavioral characteristics
that occur after stress stimuli such as footshock. However, can it fully represent the core symptoms of
PTSD? The freezing levels, which are a critical measure of fear response, are not presented anywhere in
the manuscript. Do freezing levels increase or decrease after one or two weeks? Does fear generalization
develop after one or two weeks? Given that PTSD is fundamentally a memory disorder, the authors did
not provide sufficient evidence to demonstrate that their animal model successfully recapitulates PTSD.
How does fear memory change with inhibition of BLA→vHPC projections, or disruption of GR
signaling in BLA→vHPC pathway, or Sgk1 knockout? It is hard to evaluate the PTSD syndrome only
with avoidance behaviors.

*Response: We sincerely thank the reviewer for the thorough assessment and valuable comments*
*regarding our PTSD animal model. In response to the points raised, we have conducted additional*
*experiments to comprehensively evaluate how well our model recapitulates core PTSD symptoms,*
*particularly regarding fear memory and extinction deficits.*

*To assess fear memory, we re-exposed mice to the original footshock context and measured contextual*
*freezing response. Our results show that FS significantly enhanced contextual fear conditioning at both*
*1 and 7 days post-FS (Response Fig. 25, now Supplementary Fig. 5 in the revised manuscript, with*
*citation on page 4, lines 96-100).*

**Response Fig. 25 (related to Supplementary Fig. 5) Footshock stress causes elevated fear memory**
**in male mice. a** Time scheme for contextual fear conditioning (CFC) test. **b** Summary plots of freezing
level of mice in CFC test (Control: $n = 12$ mice; 1d P-FS: $n = 12$ mice; 7d P-FS: $n = 13$ mice). Data were
shown as mean \pm SEM. $*p < 0.05$, $**p < 0.01$, $***p < 0.001$. See Supplementary Data 1 for full statistical

information. Source data are provided as a Source Data file.

We further examined fear extinction using a standardized fear conditioning-extinction paradigm
(Response Fig. 26a). During the habituation phase, FS mice already exhibited elevated freezing to the
conditioned stimulus (CS), and showed a more pronounced increase in conditioned fear during
acquisition (Response Fig. 26b). Most importantly, FS significantly impaired extinction, as reflected by
reduced Extinction Recall Index (ERI) and Extinction Learning Index (ELI)^{2,3}, despite comparable initial
freezing levels during early extinction (Response Fig. 26c-e). Collectively, these results demonstrate that
FS significantly enhances fear levels and impairs fear extinction in mice.

**Response Fig. 26 Footshock stress impaired fear extinction in male mice.** **a** Time scheme for fear
conditioning and extinction. **b, c** Summary plots of freezing level of mice in response to CS during
habituation, fear conditioning (**b**), fear extinction, and retrieval test (**c**) (Control: $n = 15$ mice; 7d P-FS:
$n = 13$ mice). **d, e** Summary plots of ELI (**d**) and ERI (**e**) of mice during fear extinction and retrieval test.
Same sample size as in **b, c**. Data were shown as mean \pm SEM. * $p < 0.05$.

To investigate the molecular mechanisms, we tested whether glucocorticoid receptor (GR) activation in
the BLA contributes to FS-induced fear enhancement. Local microinfusion of mifepristone into the BLA
completely blocked the FS-induced enhancement of contextual fear (Response Fig. 27, now
Supplementary Fig. 10 in the revised manuscript, with citation on page 7, line 194).

Together with the avoidance behavior reported in the original manuscript, these new findings
demonstrate that our FS model effectively captures multiple core PTSD-like symptoms: persistent fear
memory, impaired fear extinction, and pronounced avoidance. This comprehensive behavioral profile
supports the validity of our model for studying PTSD pathophysiology and underlying circuit
mechanisms.

**Response Fig. 27 (related to Supplementary Fig. 10) Blocking of GR signaling within BLA mitigates**

**FS-induced increase in fear memory in male mice. a** Schematic of experiment procedures. **b** Summary
plots of freezing level of mice in CFC test (Vehicle+control: $n = 12$ mice; Vehicle+7d P-FS: $n = 14$ mice,
Mifepristone+control: $n = 12$ mice; Mifepristone +7d P-FS: $n = 12$ mice).

Some labs have already reported that animals exhibit avoidance behaviors after acute stress, such as
footshock. While the authors conducted numerous experiments and concluded that "Footshock stress
exposure causes delayed and enduring PTSD-like avoidance behavior in mice," this finding is not novel
and has been previously documented. The results are presented in Figure 1 and Supplementary Figures
1-3, but they do not significantly advance the field.

*Response: We thank the reviewer for this comment. As rightly noted, PTSD is a heterogeneous disorder*
*encompassing multiple core symptoms, including persistent fear and delayed avoidance. While prior*
*studies have established that acute stress can induce avoidance behaviors in animals, and considerable*
*research has illuminated the amygdala-related circuitry of fear responses—as well as the involvement*
*of other regions such as the prefrontal cortex and hippocampus in avoidance regulation⁴⁻⁶—the specific*
*role of the basolateral amygdala (BLA) in the delayed expression of PTSD-like avoidance remains*
*underexplored. Furthermore, the underlying circuit and molecular mechanisms are still not well defined.*
*To address these gaps, we designed a series of experiments to establish a reliable model of PTSD-like*
*delayed avoidance. Our findings ultimately reveal that upregulation of Sgk1 in hippocampus-projecting*
*BLA neurons mediates the delayed emergence of avoidance behavior, providing new mechanistic insight*
*into this specific symptom dimension.*

Additionally, how does the activity of BLA neurons or BLA-vHPC/NAc projection neurons (PNs)
change during behavioral tasks such as the Open Field and Elevated Plus Maze? This information is
crucial for understanding the broader functional implications of these neuronal populations in PTSD-like
behaviors.

*Response: As recommended, we have performed new fiber photometry experiments to monitor calcium*
*transients in BLA PNs during the Elevated Plus Maze (EPM) and Open Field (OFT) tests. As shown in*
*Response Fig. 28 (now Fig. 1h-p in the revised manuscript, with citation on page 4, lines 104-110),*
*control mice exhibited no significant changes in calcium activity when entering the open arms of the*
*EPM or the center of the OF, either before or after foot shock (FS). In contrast, FS-stressed mice*
*displayed a specific and significant increase in calcium transients at 7 days post-FS during exploration*
*of potentially threatening situations, with no notable change at 1 day post-FS. These results indicate that*
*the emergence of delayed avoidance behavior coincides with a hyperactive state of BLA PNs in response*
*to potentially threatening situations.*

**Response Fig. 28 (related to Fig. 1h-p)** a Schematic of injection of AAV-CaMKII α -GCaMP6s into the
 BLA and implantation of optical fiber to record calcium signals. b Representative image showing
 GCaMP6s expression and the cannula implantation onto the BLA, scale bar: 200 μ m. c Time scheme of
 experimental procedures for *in vivo* fiber photometry. d-f Heatmap of fluorescence (d), mean
 fluorescence (e), and comparisons of calcium transients (f) of mice during entry into the center zone in
 OFT. ($n = 6$ mice per group). g-i Heatmap of fluorescence (g), mean fluorescence (h), and comparisons
 of calcium transients (i) of mice during entry into the open arms in EPMT. ($n = 6$ mice per group). Data
 were shown as mean \pm SEM. ** $p < 0.01$. See Supplementary Data 1 for full statistical information.
 Source data are provided as a Source Data file.

*In Response Fig. 29 (now Fig. 2a-p in the revised manuscript, with citation on page 5, lines 119-121),*
 *we also found no notable changes in the activity of either BLA \rightarrow vHPC PNs or BLA \rightarrow NAc PNs in control*
 *mice upon entering the open arms of the EPM or the center of the OF at any of the three time points.*
 *However, FS specifically induced a delayed increase in Ca²⁺ events in BLA \rightarrow vHPC PNs, but not in*
 *BLA \rightarrow NAc PNs.*

**Response Fig. 29 (related to Fig. 2a-p)** a Schematic showing the recording of calcium signals in
 GCaMP6s-expressing BLA→vHPC PN. The AAV vectors injected into BLA and vHPC were also
 shown. b Representative image showing GCaMP6s expression in BLA→vHPC PN and the cannula
 implantation above the BLA, scale bar: 200 μ m. c-e Heatmap of fluorescence (c), mean fluorescence (d),
 and comparisons of calcium transients (e) of mice during entry into the center zone in OFT. (n = 6 mice
 877 per group). f-h Heatmap of fluorescence (f), mean fluorescence (g), and comparisons of calcium
 transients (h) of mice during entry into the open arms in EPMT. (n = 6 mice per group). k-p Same as in
 (c-h) except that the calcium signals were recorded from BLA→NAc PN (n = 6 mice per group). Data
 were shown as mean \pm SEM. ***p < 0.001. See Supplementary Data 1 for full statistical information.
 Source data are provided as a Source Data file.

2. Are BLA→vHPC PN and BLA→NAc PN are two different neuronal populations in the BLA? What
 is the overlap rate of BLA→vHPC PN and BLA→NAc PN? The double retrograde tracing experiment
 is suggested.

**Response:** We thank the reviewer for this valuable comment. The BLA→vHPC and BLA→NAc
 projection neurons represent two largely distinct subpopulations within the BLA, with minimal overlap
 between them (Response Fig. 30), consistent with previous anatomical studies^{7,8}. We have added a

clarifying statement and relevant references in the revised manuscript to explicitly note the distinction
between these two projection pathways (page 5, lines 115-119).

**Response Fig. 30 BLA→vHPC PNs are largely non-overlapping with BLA→NAc PNs.** a Schematic
showing the injection of retrograde AAV-CaMKII α -mCherry and AAV-CaMKII α -EGFP into NAc and
vHPC, respectively. b Representative images show the injection site in NAc and vHPC. Scale bar = 200
μ m. c Representative images showing fluorescently -labeled BLA→NAc PNs and BLA→vHPC PNs.
Left panel, scale bar = 100 μ m, right panel, scale bar = 25 μ m. d Proportions of the fluorescently -
labeled PNs in BLA.

3. CORT signaling plays a critical role in PTSD is well-known and the results of CORT synthesis
inhibitor or GR antagonist on FS-induced avoidance behavior are not that surprise or attractive.

**Response:** We appreciate the reviewer's insightful comment and fully agree that the central role of
CORT/GR signaling in PTSD is well established. Accordingly, our findings using CORT synthesis
inhibitors and GR antagonists are consistent with this canonical pathway. However, the main objective
of our study was to move beyond this well-known hormonal mechanism and to identify the molecular
determinants underlying cell-type-specific vulnerability within the amygdala.

Our results demonstrate that, although GR is broadly expressed across BLA neurons, the critical
alteration occurs downstream at the level of Sgk1. Specifically, we found that Sgk1 was selectively
upregulated in BLA→vHPC projection neurons after stress, whereas GR expression remained
unchanged. These findings suggest that Sgk1 functions as a key molecular node that translates
generalized GR activation into circuit-specific maladaptive plasticity. Therefore, while GR signaling
serves as a common initiating event, the stress-induced dysregulation of Sgk1 provides a novel layer of
mechanistic specificity that advances our understanding of how broad hormonal signals are transformed
into selective pathological outcomes.

**Minor concerns:**

1. In Figure 1i-o, did the author do the photometry recording in the context of FS or in homecage? So,

the question is whether the enhanced activities of BLA neurons is induced by FS-related context or
automatically appears. The information is lack in the whole manuscript.

*Response: We thank the reviewer for raising this important comment. In response to this and similar*
*suggestions from other reviewers, we have conducted additional fiber photometry experiments to monitor*
*calcium transients in BLA PNs during behaviorally relevant, potentially threatening situations*
*(specifically when mice entered the open arms of the elevated plus maze or the center area of the open*
*field). These new data have replaced the previous data obtained under spontaneous calcium activity in*
*the homepage and are now presented in the revised manuscript as **Response Fig. 28 (now Fig. 1h-p in***
*the revised manuscript, with citation on page 4, lines 104-110). This revision enables us to directly assess*
*neuronal dynamics within contexts associated with threat or anxiety, thereby providing more meaningful*
*and ecologically valid insights into BLA activity following stress exposure.*

2. How does knockout or expression of Sgk1 affects GR signaling in the BLA neurons? At least, this
should be discussed.

*Response: We sincerely thank the reviewer for this insightful suggestion. We have now expanded the*
*Discussion in the revised manuscript (page 12, lines 344-359) to include a dedicated segment addressing*
*how Sgk1 may influence GR-related signaling pathways in BLA neurons, thereby contributing to the*
*delayed emergence of PTSD-like avoidance behavior.*

3. The overexpression levels of Sgk1-DN or Sgk1 are excessively high and appear to induce toxic effects.
The calculation of mRNA expression is incorrect. They should be compared with the housekeeping gene
(e.g., GAPDH or β -actin), rather than mCherry.

*Response: We thank the reviewer for this thoughtful comment and for pointing out these important*
*methodological issues. We agree that excessive overexpression may lead to cellular toxicity. The*
*relatively high mRNA levels of Sgk1-DN or Sgk1-OE observed in our qPCR analyses are likely due to*
*methodological bias during sample collection. Specifically, during fluorescence microscopy – guided*
*sorting, neurons with stronger fluorescence signals were more likely to be selected, which could have*
*skewed the sample toward cells with higher transgene expression and thus exaggerated the apparent*
*mRNA fold-change. In contrast, immunostaining-based quantification showed only a modest increase in*
*Sgk1 protein levels (approximately 2-3 fold) following viral expression, suggesting that the actual protein*
*overexpression was moderate and well within physiological limits. Furthermore, functional assessments*
*revealed no signs of neuronal toxicity or abnormal excitability. Neither Sgk1-DN nor Sgk1-OE*
*expression altered calcium dynamics, intrinsic membrane excitability, or excitatory synaptic*
*transmission in BLA neurons. At the behavioral level, Sgk1 overexpression under basal conditions did*
*not induce avoidance-like behavior, further indicating that the viral manipulations did not cause*
*nonspecific or deleterious effects. Together, these results may indicate that our viral constructs*
*effectively modulated Sgk1 signaling without inducing detectable toxicity, supporting their suitability for*
*mechanistic investigations.*

*Regarding the reviewer's concern about mRNA normalization, in all qPCR analyses, Sgk1 and Sgk1-DN*
*mRNA levels were normalized to the housekeeping gene GAPDH, not to mCherry. **The revised Materials***
*and Methods section (page 20, lines 583-584) now includes a detailed description of this quantification*
*procedure to ensure methodological clarity and reproducibility.*

**Reviewer #4 (Remarks to the Author):**

This paper by Jia-Xin Zou, Bing-Xing Pan, and colleagues presents an exciting set of experiments
showing that stress-induced increases in Sgk1 within the amygdala – specifically those neurons that
project to the hippocampus - support persistent alterations in anxiety-related behaviors. They replicate
findings that amygdala projections to the ventral hippocampus are important for anxiety-related behavior
(Felix-Ortiz et al, 2013). More importantly, they go on to show that stress-induced glucocorticoid release
and persistent increases in Sgk1 expression within this cell population are important for enhancements
in neuronal excitability and changes anxiety related-behavior. This work provides a more mechanistic
link between stress-induced glucocorticoid release and limbic circuit alterations relevant to anxiety
disorders. There a few conceptual and experimental issues with the work but I think all of these can be
easily remedied.

**Major concerns:**

1. The authors erroneously claim that over-expression of Sgk1 is sufficient to promote anxiety-related
behavior in the open field and elevated plus maze tests. The authors suggest that because animals in
which they overexpress Sgk1 in the amygdala show a significant increase in anxiety-related behavior in
response to a ‘subthreshold’ stressor, whereas control animals that do not overexpress Sgk1 do not, that
Sgk1 promotes this behavior. However, there is an apparent trend in the control animals. Moreover, there
is not a statistical interaction between viral group and stress condition – only a main effect of stress. As
such, the statistical reasoning they make is false. The authors need to either omit the claim that
overexpression promotes anxiety-related behavior or increase sample size until they have sufficient
power to detect the interaction. That said, I find the paper just as impactful if Sgk1 is necessary for the
observed behavioral change but not sufficient.

*Response: We thank the reviewer for the insightful statistical critique. In response, we have repeated the
experiments with an increased sample size to ensure adequate statistical power.*

*As shown in Response Figure 31 (now Figs. 6l-o in the revised manuscript, with citation on pages 8-9,
lines 231-232), Sgk1-DN expression in BLA→vHPC PNs had no effect under baseline conditions but
significantly attenuated avoidance behavior following footshock stress, confirming that Sgk1 is necessary
for the full expression of stress-induced avoidance.*

**Response Fig. 31 (related to Fig. 6l-o) a, b** Summary plots of time in center area (a) and total distance
traveled (b) in OFT (mCherry+control: $n = 12$ mice; mCherry +7d P-FS: $n = 13$ mice; Sgk1-DN+control:
$n = 12$ mice; Sgk1-DN +7d P-FS: $n = 12$ mice). c, d Summary plots of time in open arms (c) and open-
arm entries (d) in EPMT, sample size as in panel (a, b). Data were shown as mean \pm SEM. * $p < 0.05$,
** $p < 0.01$, *** $p < 0.001$. See Supplementary Data 1 for full statistical information. Source data are
provided as a Source Data file.

Critically, the new results in *Response Figure 32* (now *Figs. 7g-j* in the revised manuscript, with citation on page 9, lines 249-253) demonstrate a statistically significant interaction between viral manipulation and stress condition. Specifically, *Sgk1*-overexpressing mice, but not *mCherry* controls, exhibited a marked increase in avoidance behavior following subthreshold footshock. This indicates that *Sgk1* overexpression is sufficient to confer behavioral susceptibility to a otherwise subeffective stressor. We have revised our conclusions accordingly and thank the reviewer for prompting this important clarification.

Response Fig. 32 (related to Fig. 7g-j) a, b Summary plots of time in center area (a) and total distance traveled (b) in OFT (mCherry+control: $n = 11$ mice; mCherry +7d P-sub-FS: $n = 13$ mice; Sgk1-OE+control: $n = 13$ mice; Sgk1-OE +7d P-sub-FS: $n = 12$ mice). c, d Summary plots of time in open arms (c) and open arm entries (d) in EPMT, sample size as in panel (a, b). Data were shown as mean \pm SEM. * $p < 0.05$, ** $p < 0.01$, *** $p < 0.001$. See Supplementary Data 1 for full statistical information. Source data are provided as a Source Data file.

2. The authors repeatedly use the term ‘avoidance’ and I think that this is problematic for several reasons. First, the behavioral tests they utilize are most often referred to as tests for anxiety-related behavior, including work they base their findings on (Felix-Ortiz, 2013). Moreover, typically people use avoidance in the context of active avoidance procedures (e.g., pressing a lever or moving to platform to avoid an aversive outcome). Consequently, calling this avoidance results in confusion. Second, and perhaps more importantly, avoidance has historically been defined in the learning literature as an instrumental action taken to remove the onset of an aversive outcome (Domjan, Principles of Learning and Behavior textbook). In such cases, there is a clear contingency/relationship between taking an action and the presentation of an aversive outcome. The open-field and elevated plus maze tests are problematic because it is unclear what these contingencies are.

Response: We thank the reviewer for raising this important point regarding the use of the term “avoidance”. We fully appreciate that, in the context of instrumental learning, “avoidance” classically refers to an operant behavior performed to prevent an aversive outcome based on a clear action–outcome contingency. In our study, we use the term to describe the innate behavioral tendency of mice to stay away from potentially threatening areas—such as open, bright, or elevated spaces—in tests like the open field and elevated plus maze. The behavioral measures we used, including reduced time spent in the center or open arms, are widely interpreted in the literature as reflecting “approach–avoidance conflict” or “risk avoidance”, capturing a naturalistic form of avoidance in response to potential threat^{9–11}. This interpretation aligns with a body of preclinical studies that employ these tests to quantify PTSD-related avoidance behaviors in rodents^{11,12}. We will clarify this terminology in the revised manuscript (page 3,

*line 76) to prevent any potential misunderstanding.*

3. The authors demonstrate that administration of CNO (vs vehicle) is able to diminish anxiety-related
behaviors in animals that express HM4Di in BLA->vHC projections. However, they utilize a high dose
of CNO that can have off-target effects. A supplementary experiment showing that the dose of CNO used
has no effect in control animals would be ideal.

*Response: We thank the reviewer for raising this important point regarding CNO dosage. Following the*
*suggestion, we have conducted new control experiments in which mCherry (without hM4Di) was*
*expressed in BLA->vHPC PNs.*

*As summarized in Response Fig. 33a (now Supplementary Fig. 7a in the revised manuscript), mice*
*underwent FS and were administered either vehicle or CNO (at the same dose used in the main*
*experiments) 30 minutes before behavioral tests. We observed that FS significantly reduced time spent*
*in the center of the OF and in the open arms of the EPM to a similar extent in both vehicle- and CNO-*
*treated mCherry control groups (Response Fig. 33d-e, now Supplementary Fig. 7d-e in the revised*
*manuscript, with citation on page 6, lines 157-159).*

*These results confirm that CNO at the applied dose may not produce off-target effects on avoidance*
*behaviors under our experimental conditions, thereby validating the specificity of the chemogenetic*
*inhibition effects observed in hM4Di-expressing mice.*

**Response Fig. 33 (related to Supplementary Fig. 7) CNO produced no behavioral changes in mice**
**expressing the control mCherry virus in BLA->vHPC PNs.** **a** Schematic of the experimental
procedures. **b, c** Summary plots of time in center area (**b**) and total distance traveled (**c**) in OFT (Vehicle:
control, $n = 11$ mice, 7d P-FS, $n = 10$ mice; CNO: control, $n = 12$ mice, 7d P-FS, $n = 9$ mice). **d, e**
Summary plots of time in open arms (**d**) and open arm entries (**e**) in EPMT (Vehicle+control, $n = 11$ mice,
Vehicle+7d P-FS, $n = 10$ mice; CNO+control, $n = 12$ mice, CNO+7d P-FS, $n = 9$ mice). Data were shown
as mean \pm SEM. * $p < 0.05$, ** $p < 0.01$, *** $p < 0.001$. See Supplementary Data 1 for full statistical
information. Source data are provided as a Source Data file.

4. The authors look at spontaneous activity of BLA neurons using fiber photometry and cFos. First, it is
unclear where this was done. In the home cage? These methods should be outlined. Secondly, it
somewhat confusing why it was done outside of the open-field or elevated plus maze. Often times there
is little to no expression of cFos in the homecage. Could the authors explain. Thirdly, given that
heightened activity of BLA to vHC projection neurons are observed outside of the elevated plus maze
and open field, what does this say about the expression of the behavioral changes observed. How can

activity of a projection be important for a behavior that is not occurring?

**Response:** We thank the reviewer for these insightful questions regarding the experimental context and
 interpretation of our neural activity data. In response, we have replaced the original homepage
 recordings with new fiber photometry experiments conducted during the EPMT and OFT, as outlined in
 the revised Methods. This approach allows us to assess neural activity directly in the specific behavioral
 contexts.

As shown in **Response Figure 34** (now **Fig. 1h-p** in the revised manuscript, with citation on page 4, lines
 104-110), control mice exhibited stable calcium activity in BLA PNs when entering potentially
 threatening situations (open arms or center) both at 1 and 7 days post-FS, showing no significant change
 relative to their pre-stress baseline. In contrast, FS-stressed mice displayed a significant increase in
 calcium activity specifically at 7 days post-stress in these same contexts. This context-dependent
 hyperactivation, emerging only in the stressed group, aligns temporally with the development of
 avoidance behavior.

**Response Fig. 34** (related to **Fig. 1h-p**) a Schematic of injection of AAV-CaMKII α -GCaMP6s into the
 BLA and implantation of optical fiber to record calcium signals. b Representative image showing
 GCaMP6s expression and the cannula implantation onto the BLA, scale bar: 200 μ m. c Time scheme of
 experimental procedures for *in vivo* fiber photometry. d-f Heatmap of fluorescence (d), mean
 fluorescence (e), and comparisons of calcium transients (f) of mice during entry into the center zone in
 OFT. ($n = 6$ mice per group). g-i Heatmap of fluorescence (g), mean fluorescence (h), and comparisons
 of calcium transients (i) of mice during entry into the open arms in EPMT. ($n = 6$ mice per group). Data
 were shown as mean \pm SEM. ****** $p < 0.01$. See Supplementary Data 1 for full statistical information.
 Source data are provided as a Source Data file.

 We next examined changes in calcium signaling at the circuit level. As shown in **Response Figure 35**
 (now **Fig. 2a-p** in the revised manuscript, with citation on page 5, lines 115-121), control mice showed
 no significant change in the activity of either BLA \rightarrow vHPC or BLA \rightarrow NAc PNs at 1 or 7 days post-FS
 relative to their own pre-FS baseline. In the FS group, however, a delayed (at 7 days post-FS)

enhancement of calcium activity was specifically induced in the BLA→vHPC pathway, whereas no concurrent change was observed in the BLA→NAc pathway.

**Response Fig. 35 (related to Fig. 2a-p) FS preferentially enhances the activity of BLA→vHPC PN.**

**a** Schematic showing the recording of calcium signals in GCaMP6s-expressing BLA→vHPC PN. The
AAV vectors injected into BLA and vHPC were also shown. **b** Representative image showing GCaMP6s
expression in BLA→vHPC PN and the cannula implantation above the BLA, scale bar: 200 μ m. **c-e**
Heatmap of fluorescence (**c**), mean fluorescence (**d**), and comparisons of calcium transients (**e**) of mice
during entry into the center zone in OFT. ($n = 6$ mice per group). **f-h** Heatmap of fluorescence (**f**), mean
fluorescence (**g**), and comparisons of calcium transients (**h**) of mice during entry into the open arms in
EPMT. ($n = 6$ mice per group). **k-p** Same as in (**c-h**) except that the calcium signals were recorded from
BLA→NAc PN ($n = 6$ mice per group). Data were shown as mean \pm SEM. *** $p < 0.001$. See
Supplementary Data 1 for full statistical information. Source data are provided as a Source Data file.

*We also determined whether Sgk1 activity in BLA→vHPC PN regulates the observed calcium dynamics.*

*As shown in Response Figure 36 (now Fig. 6p-u in the revised manuscript, with citation on page 9, lines*
*234-236) and Response Fig. 37 (now Fig. 7k-p in the revised manuscript, with citation on page 9, lines*

254-257), overexpressing *Sgk1*-DN in *BLA*→*vHPC* PNs effectively prevented the increase of calcium
 signal induced by FS during the OFT and EPMT. Conversely, *Sgk1* overexpression in this subpopulation
 significantly elevated the calcium signal under the sub-FS condition.

 **Response Fig. 36 (related to Fig. 6p-u)** a-c Heatmap of fluorescence (a), mean fluorescence (b), and
 comparisons of calcium transients (c) of mice during entry into the center zone in OFT ($n = 6$ mice per
 group). d-f Heatmap of fluorescence (d), mean fluorescence (e), and comparisons of calcium transients
 (f) of mice during entry into the open arms in EPMT ($n = 6$ mice per group). Data were shown as mean
 \pm SEM. $**p < 0.01$. See Supplementary Data 1 for full statistical information. Source data are provided
 as a Source Data file.

 **Response Fig. 37 (related to Fig. 7k-p)** a-c Heatmap of fluorescence (a), mean fluorescence (b), and
 comparisons of calcium transients (c) of mice during entry into the center zone in OFT. ($n = 6$ mice per
 group). d-f Heatmap of fluorescence (d), mean fluorescence (e), and comparisons of calcium transients
 (f) of mice during entry into the open arms in EPMT. ($n = 6$ mice per group). Data were shown as mean
 \pm SEM. $**p < 0.01$. See Supplementary Data 1 for full statistical information. Source data are provided
 as a Source Data file.

**MINOR**

1. Please include titers for viruses

**Response:** We thank the reviewer for pointing out this omission. The titer information for all viruses used

*in this study has now been added to the **Materials and Methods** section of the revised manuscript (page*
*16, lines 459-460).*

2. It is stated that calcium events were defined as increases of 2 std deviations above baseline, but the
authors do not say what that baseline is (the minimum value over the entire session?)

***Response:** We thank the reviewer for raising this important methodological point. In response, we have*
*replaced the original spontaneous activity recordings with new fiber photometry data acquired during*
*behavioral tests (EPMT and OFT). All calcium transient traces are now presented as z-scores, with the*
*area under the curve (AUC) quantified for statistical comparison. A detailed description of the fiber*
*photometry analysis, including the baseline definition and z-score calculation method, has been added*
*to the **Materials and Methods** section (page 16, lines 462-474).*

3. The authors show that the change in anxiety-related behavior observed after an initial stressor increases
with time from that stressor. Notably, fear generalization is often thought to increase over time (Wiltgen
and Silva, 2007). To what extent are the authors findings dependent upon associative fear of the initial
stressor context? Does silencing the amygdala to ventral hippocampus projection also reduce fear of the
initial stressor context? I don't feel addressing this question is critical. Mostly my intrigue.

***Response:** We appreciate this insightful suggestion. To address the question, we conducted a series of*
*additional experiments investigating the effects of foot shock (FS) on both fear conditioning and*
*extinction in mice.*

*First, to evaluate the impact of FS on contextual fear conditioning, we re-exposed FS-treated mice to the*
*same context in which foot shocks were administered and measured their contextual fear responses. Our*
*results indicate that FS significantly enhanced contextual fear at 1 and 7 days post-stress (**Response Fig.***
*38, now **Supplementary Fig. 5** in the revised manuscript, with citation on page 4, lines 96-100).*

*Second, to examine the effects of FS on the fear extinction in mice, all mice were subjected to a*
*standardized fear-conditioning and extinction procedure (**Response Fig. 39a**). Consistent with the*
*contextual fear results, FS-treated mice exhibited increased freezing levels in response to the CS during*
*the habituation phase (**Response Fig. 39b**). During the fear acquisition session, both control and FS mice*
*displayed a robust increase in conditioned fear response, which was more pronounced in FS mice*
*(**Response Fig. 39b**). However, FS significantly impaired extinction learning, as evidenced by lower*
*Extinction Recall Index (ERI) and Extinction Learning Index (ELI)^{2,3} in FS mice, despite comparable*
*freezing levels between groups in response to the first two CS presentations within the initial extinction*
*session (**Response Fig. 39c-e**). Collectively, these findings indicate that FS not only elevates fear levels*
*but also disrupts fear extinction in mice.*

*Furthermore, to determine whether the activation of GR in the BLA underlies FS-induced contextual fear,*
*we locally microinfused mifepristone into the BLA and found that this pretreatment effectively prevented*
*the FS-induced enhancement of contextual fear (**Response Fig. 40, related to **Supplementary Fig. 10** in***
*the revised manuscript, with citation on page 7, line 194).*

Response Fig. 38 (related to Supplementary Fig. 5) Footshock stress causes elevated fear memory in male mice. a Time scheme for contextual fear conditioning (CFC) test. **b** Summary plots of freezing level of mice in CFC test (Control: $n = 12$ mice; 1d P-FS: $n = 12$ mice; 7d P-FS: $n = 13$ mice). Data were shown as mean \pm SEM. *** $p < 0.001$. See Supplementary Data 1 for full statistical information. Source data are provided as a Source Data file.

Response Fig. 39 Footshock stress impaired fear extinction and extinction memory retention in male mice. a Time scheme for fear conditioning and extinction. **b, c** Summary plots of freezing level of mice in response to CS during habituation, fear conditioning (**b**), fear extinction, and retrieval test (**c**) (Control: $n = 15$ mice; 7d P-FS: $n = 13$ mice). **d, e** Summary plots of ELI (**d**) and ERI (**e**) of mice during fear extinction and retrieval test. Same sample size as in **b, c**. Data were shown as mean \pm SEM. * $p < 0.05$. See Supplementary Data 1 for full statistical information. Source data are provided as a Source Data file.

Response Fig. 40 (related to Supplementary Fig. 10) Blocking of GR signaling within BLA mitigates

**FS-induced increase in fear memory in male mice.** **a** Schematic of experiment procedures. **b** Summary
plots of freezing level of mice in CFC test (Vehicle+control: $n = 12$ mice; Vehicle+7d P-FS: $n = 14$ mice,
Mifepristone+control: $n = 12$ mice; Mifepristone +7d P-FS: $n = 12$ mice). Data were shown as mean \pm
SEM. *** $p < 0.001$. See Supplementary Data 1 for full statistical information. Source data are provided
as a Source Data file.

**References**

- 1. Licznarski, P. *et al.* Decreased SGK1 Expression and Function Contributes to Behavioral Deficits
Induced by Traumatic Stress. *PLoS Biol* **13**, e1002282 (2015).
- 2. Pan, H.-Q. *et al.* Prefrontal GABAA(δ)R Promotes Fear Extinction through Enabling the Plastic
Regulation of Neuronal Intrinsic Excitability. *J Neurosci* **42**, 5755–5770 (2022).
- 3. Milad, M. R. *et al.* Presence and acquired origin of reduced recall for fear extinction in PTSD: results
of a twin study. *J Psychiatr Res* **42**, 515–520 (2008).
- 4. C, S.-B., R, A., K, M., Rws, W. & Af, M. Two opposing hippocampus to prefrontal cortex pathways
for the control of approach and avoidance behaviour. *Nature communications* **13**, (2022).
- 5. Padilla-Coreano, N. *et al.* Hippocampal-Prefrontal Theta Transmission Regulates Avoidance
Behavior. *Neuron* **104**, 601-610.e4 (2019).
- 6. Godino, A. *et al.* Dopamine D1–D2 signalling in hippocampus arbitrates approach and avoidance.
*Nature* 1–10 (2025) doi:10.1038/s41586-025-08957-5.
- 7. Zhang, W. H. *et al.* Chronic Stress Causes Projection-Specific Adaptation of Amygdala Neurons via
Small-Conductance Calcium-Activated Potassium Channel Downregulation. *Biol Psychiatry* **85**,
812–828 (2019).
- 8. Zhang, J. Y. *et al.* Chronic Stress Remodels Synapses in an Amygdala Circuit-Specific Manner. *Biol*
*Psychiatry* **85**, 189–201 (2019).
- 9. Jacinto, L. R., Cerqueira, J. J. & Sousa, N. Patterns of Theta Activity in Limbic Anxiety Circuit
Preceding Exploratory Behavior in Approach-Avoidance Conflict. *Front Behav Neurosci* **10**, 171
(2016).
- 10. Loewke, A. C., Minerva, A. R., Nelson, A. B., Kreitzer, A. C. & Gunaydin, L. A. Frontostriatal
Projections Regulate Innate Avoidance Behavior. *J Neurosci* **41**, 5487–5501 (2021).
- 11. Verbitsky, A., Dopfel, D. & Zhang, N. Rodent models of post-traumatic stress disorder: behavioral
assessment. *Transl Psychiatry* **10**, 132 (2020).
- 12. Zhang, H.-H. *et al.* Traumatic Stress Produces Delayed Alterations of Synaptic Plasticity in
Basolateral Amygdala. *Front Psychol* **10**, 2394 (2019).

Point-to-point response to the reviewers' comments

REVIEWER COMMENTS

Reviewer #1 (Remarks to the Author):

Zou et al present a manuscript that outlines a specific amygdalar mechanism that underlies the delayed onset of PTSD-like avoidance behavior. They show that 7 days following an inescapable footshock protocol, there is hyperactivation of the BLA-vHPC pathway that is driven by upregulation of Sgk1, which functions downstream of the glucocorticoid receptor.

The authors did an impressive, thorough job addressing the reviewers' comments. The significance and impact of the primary findings are made much clearer in the revision. These findings show a clear mechanistic underpinning of the mechanism of persistent PTSD-like avoidance behavior, and set next steps for intervention to prevent delayed avoidance behavior after inescapable stress.

Two very minor suggestions for the text:

1) In figure 2s and w, the authors should consider making it clearer in the figure legend that these figures are for cfos only in the BLA (if this is the correct interpretation). A simple text edit for clarity would help readers. This wording is particularly confusing for q-t because the figure legend says "same as in q-t except that the data were from BLA-NAc PNs," except the cfos data in w is from BLA, not from the PNs, I believe.

Response: We appreciate the reviewer's suggestion to clarify the c-Fos data in Figure 2. We have revised the figure legends to explicitly distinguish between the total BLA c-Fos expression (Figs. 2s and 2w) and the projection-specific c-Fos expression in BLA→vHPC and BLA→NAc PNs (Figs. 2r, t, v, x) (see page 32, lines 901-908).

2) The use of just a generic term "control group" became confusing after a few figures because the control can change, particularly when thinking about the control for the DREADD experiment is vehicle not an unstressed control. In figure 5, it was slightly unclear what the control was, given a different control in figure 4. The figure legend only says that "data was normalized to control group." It would help improve readers interpretation to reiterate unstressed control throughout the text. Also in line 91, where this is first defined, it can help the reader to add clarifying detail that this is how "control" is defined throughout. Overall, the manuscript is well done. Congratulations to the authors on the work.

Response: We sincerely appreciate the reviewer for this important suggestion to improve the manuscript's readability. Throughout the text, we have replaced the generic term "control" in text with specific descriptors (e.g., "unstressed control") to explicitly indicate the comparison group in each experiment. (see page4, lines 91, 98, and figure legends: pages 32-38)

We thank the reviewer for helping us improve the manuscript's readability and precision.

Reviewer #2 (Remarks to the Author):

I commend the authors for the substantial amount of work they have done for the revision. The photometry experiments add strong support for the main claims, and my questions have been well addressed. I have a few minor comments that primarily concern clarity of the newly added photometry results.

1) Figures 1, 2, 6, and 7: For the AUC analysis, the authors report a 2-way RM ANOVA (treatment x day) and follow up with within-group comparisons. Since the photometry signal can change across days for reasons unrelated to the PTSD manipulation (ie. expression, recording stability, habituation) the key question is whether the FS group differs from the control group at each time point. Please report the group effects at pre-, 1d, and 7d (corrected for multiple comparisons) and describe these findings in the Results.

Response: We sincerely thank the reviewer for the valuable comments and fully agree that addressing these concerns is critical to strengthen the conclusions regarding role of increased activity of BLA→vHPC PNs in the delayed onset of PTSD-like avoidance behavior. Following the reviewer's suggestion, we have re-analyzed these data in figures 1, 2, 6, and 7.

In Response Figure 1 (now Fig. 1i-p in the revised manuscript, with citation on page 4, lines 108-118), we observed no significant differences in the Ca²⁺ transients of BLA PNs between the control and FS-stressed mice at Pre-FS, 1day post-FS. In contrast, FS-stressed mice exhibited an increase in Ca²⁺ transients at 7 days post-FS relative to control mice.

Similarly, in Response Figure 2 (now Fig. 2a-p in the revised manuscript, with citation on page 5, lines 129-135), the activity of BLA→vHPC PNs did not differ between control and FS-stressed mice at Pre-FS, 1 day post-FS, but was significantly elevated at 7 days post-FS in the stressed group. However, the Ca²⁺ transients of BLA→NAc PNs remained unchanged across all time points in both groups.

In Response Figure 3 (now Fig. 6p-u in the revised manuscript, with citation on page 9, lines 248-252), we found that FS markedly increased the activity of BLA→vHPC PNs at 7 days post-FS. Whereas this increase was effectively blocked by overexpression of Sgk1-DN, as evidenced by a lower AUC of Ca²⁺ transients compared to FS-exposed mCherry controls.

Finally, in Response Figure 4 (now Fig. 7k-p in the revised manuscript, with citation on page 10, lines 273-276), while no significant difference in the AUC of Ca²⁺ transients was observed between mCherry control and Sgk1-OE control mice, sub-FS exposure significantly elevated calcium signals in Sgk1-OE mice relative to mCherry controls.

Accordingly, we have added the description of group effects at each time point in the relevant result sections in the revised manuscript (see page4, lines 108-118; page 5, lines 129-135; page 9, lines 248-252, and page 10, lines 273-276).

Response Fig. 1 (related to Fig. 1i-p). **a** Representative image showing GCaMP6s expression and the cannula implantation onto the BLA, scale bar: 200 μ m. **b** Time scheme of experimental procedures for in vivo fiber photometry. Unstressed control mice received the same virus injections and fiber implantations but did not undergo the FS procedure. **c** Heatmap of Z-score of calcium signals (aligned to OFT center zone entry). Each row displays trial-averaged responses per individual mouse; the color scale denotes fluorescence intensity. **d** Peri-event average Z-score of calcium signals (aligned to OFT center zone entry) for visualization. Solid line: mean; shaded area: SEM. **e** Summary plots of the AUC of calcium signals of mice during entry into the center zone in OFT calculated from the data in panel (d) (n = 6 mice per group). **f** Heatmap of Z-score of calcium signals (aligned to EPMT open arm entry). **g** Peri-event average trace of mean fluorescence of calcium signals (aligned to EPMT open arm entry) for visualization. Solid line: mean; shaded area: SEM. **h** Summary plots of AUC of calcium signals during entry into the open arms in EPMT, calculated from the data in panel (g) (n = 6 mice per group). Data were shown as mean \pm SEM. ** p < 0.01, *** p < 0.001. See Supplementary Data 1 for full statistical information. Source data are provided as a Source Data file.

Response Fig. 2 (related to Fig. 2a-p). **a** Schematic showing the recording of calcium signals in GCaMP6s-expressing BLA→vHPC PNs. The AAV vectors injected into BLA and vHPC were also shown. **b** Representative image showing GCaMP6s expression in BLA→vHPC PNs and the cannula implantation above the BLA, scale bar: 200 μ m. **c** Heatmap of Z-score of calcium signals (aligned to OFT center zone entry). **d** Peri-event average Z-score of calcium signals (aligned to OFT center zone entry) for visualization. Solid line: mean; shaded area: SEM. **e** Summary plots of AUC of calcium signals, calculated from the data in panel (d) ($n = 6$ mice per group). **f** Heatmap of Z-score of calcium signals (aligned to EPMT open arm entry). **g** Peri-event average trace of mean fluorescence of calcium signals (aligned to EPMT open arm entry) for visualization. Solid line: mean; shaded area: SEM. **h** Summary plots of AUC of calcium signals, calculated from the data in panel (g) ($n = 6$ mice per group). **i-p** Same as in (a-h) except that the calcium signals were recorded from BLA→NAc PNs ($n = 6$ mice per group). Data were shown as mean \pm SEM. ** $p < 0.01$, *** $p < 0.001$. See Supplementary Data 1 for full statistical information. Source data are provided as a Source Data file.

Response Fig. 3 (related to Fig. 6p-u). **a** Heatmap of Z-score of calcium signals (aligned to OFT center zone entry) (mCherry control mice expressed the mCherry and GCaMP6s in BLA→vHPC neurons. Sgk1-DN mice expressed the Sgk1(S422A)-mCherry fusion protein and GCaMP6s in BLA→vHPC neurons. Both groups underwent the same FS procedure.). **b** Peri-event average Z-score of calcium signals (aligned to OFT center zone entry) for visualization. Solid line: mean; shaded area: SEM. **c** Summary plots of AUC of calcium signals, calculated from the data in panel (b) ($n = 6$ mice per group). **d** Heatmap of Z-score of calcium signals (aligned to EPMT open arm entry). **e** Peri-event average Z-score of calcium signals (aligned to EPMT open arm entry) for visualization. Solid line: mean; shaded area: SEM. **f** Summary plots of AUC of calcium signals, calculated from the data in panel (e) ($n = 6$ mice per group). Data were shown as mean \pm SEM. * $p < 0.05$, ** $p < 0.01$, *** $p < 0.001$. See Supplementary Data 1 for full statistical information. Source data are provided as a Source Data file.

Response Fig. 4 (related to Fig. 7k-p). **a** Heatmap of Z-score of calcium signals (aligned to OFT center zone entry) (mCherry control mice expressed the mCherry and GCaMP6s in BLA→vHPC neurons. Sgk1-OE mice expressed the Sgk1-mCherry fusion protein and GCaMP6s in BLA→vHPC neurons. Both groups underwent the same sub-FS procedure.). **b** Peri-event average Z-score of calcium signals (aligned to OFT center zone entry) for visualization. Solid line: mean; shaded area: SEM. **c** Summary plots of AUC of calcium signals, calculated from the data in panel (b) ($n = 6$ mice per group). **d** Heatmap of Z-score of calcium signals (aligned to EPMT open arm entry) **e** Peri-event average Z-score of calcium signals (aligned to EPMT open arm entry) for visualization. Solid line: mean; shaded area: SEM. **f** Summary plots of AUC of calcium signals, calculated from the data in panel (e) ($n = 6$ mice per group). Data were shown as mean \pm SEM. * $p < 0.05$, ** $p < 0.01$. See Supplementary Data 1 for full statistical information.

information. Source data are provided as a Source Data file.

2) Figures 1, 2, 6, and 7: Peri-event activity is shown but there is no initial description of what the plots display. Please add a brief description in the Fig 1 legend where these plots first appear. If the peri-event traces are intended for visualization only, please state this explicitly and confirm that AUC analyses were derived from the same data.

Response: *We apologize for the oversight and appreciate the reviewer's request for clarification. In the revised manuscript, we have added a detailed description of peri-event activity to the legend of Figure 1. We also confirm that the Area Under the Curve (AUC) analyses presented in the adjacent bar graphs were indeed derived from the same peri-event data windows shown in the traces. To ensure transparency, we have explicitly stated this in the revised Figure legends (page 31, lines 875-878). These traces are intended to visualize the real-time neural dynamics that underpin the subsequent statistical comparisons.*

3) Because the photometry mice underwent a different behavioral sequence from the main PTSD paradigm (three repeated OFT/EPM tests at pre-, 1d, and 7d), please show their behavioral performance. This will allow readers to relate the day-dependent neural changes and behavioral changes in the same animals. Please also state whether the photometry mice exhibit the same anxiety-like phenotype at the 7-day time point.

Response: *We thank the reviewer for these insightful comments and constructive suggestions. We have re-analyzed the behavioral data from all mice that underwent fiber photometry recording (Response Fig. 5-8, now Supplementary Fig.7, with citation on page 4, lines 110-112; Supplementary Fig.8, with citation on page 5, lines 129-132; Supplementary Fig.18, with citation on page 9, lines 250-252; and Supplementary Fig.20, with citation on page 10, lines 273-275 in the revised manuscript). In control mice, neither the time spent in nor the entries into the center area of the open field were altered across the Pre-FS, 1-day post-FS, and 7-day post-FS time points. In contrast, FS-stressed mice exhibited significant avoidance behavior specifically at 7 days post-FS, but not at 1 day post-FS. This avoidance was also significantly greater than that in control mice. Consistent with the open field results, FS-stressed mice also spent less time in and made fewer entries into the open arms of the EPM at 7 days post-FS.*

Response Fig. 5 (related to Supplementary Fig. 7) FS induces delayed PTSD-like avoidance behavior in fiber photometry recording male mice. **a** Summary plots of time in the center area ($n = 6$ mice per group). **b** Summary plots of total distance traveled in OFT ($n = 6$ mice per group). **c** Summary plots of time in the open arms ($n = 6$ mice per group). **d** Summary plots of open arm entries in EPMT ($n = 6$ mice per group). Data were shown as mean \pm SEM. * $p < 0.05$. See Supplementary Data 1 for full statistical information. Source data are provided as a Source Data file.

Response Fig. 6 (related to Supplementary Fig. 8) FS-exposed mice show delayed PTSD-like avoidance behavior at 7 days post-stress during fiber photometry recording. **a** Summary plots of time in the center area in OFT from mice with specifically labeled BLA→vHPC PN. **b** Summary plots of total distance traveled in OFT ($n = 6$ mice per group). **c** Summary plots of time in the open arms in EPMT from mice with specifically labeled BLA→vHPC PN ($n = 6$ mice per group). **d** Summary plots of open arm entries in EPMT ($n = 6$ mice per group). **e** Summary plots of time in the center area in OFT from mice with specifically labeled BLA→NAc PN ($n = 6$ mice per group). **f** Summary plots of total distance traveled in OFT ($n = 6$ mice per group). **g** Summary plots of time in the open arms in EPMT from mice with specifically labeled BLA→NAc PN ($n = 6$ mice per group). **h** Summary plots of time open arm entries in EPMT ($n = 6$ mice per group). Data were shown as mean \pm SEM. * $p < 0.05$, ** $p < 0.01$, *** $p < 0.001$. See Supplementary Data 1 for full statistical information. Source data are provided as a Source Data file.

Response Fig. 7 (related to Supplementary Fig. 18) Sgk1-DN expression prevents the delayed PTSD-like avoidance behavior by FS in fiber photometry recording male mice. **a** Summary plots of time in the center area in OFT ($n = 6$ mice per group). **b** Summary plots of total distance traveled in OFT ($n = 6$ mice per group). **c** Summary plots of time in the open arms in EPMT ($n = 6$ mice per group). **d** Summary plots of open arm entries in EPMT ($n = 6$ mice per group). Data were shown as mean \pm SEM. * $p < 0.05$, ** $p < 0.01$. See Supplementary Data 1 for full statistical information. Source data are provided as a Source Data file.

Response Fig. 8 (related to Supplementary Fig. 20) Sub-FS induces delayed PTSD-like avoidance behavior in Sgk1- overexpressing mice in fiber photometry recording. **a** Summary plots of time in the center area ($n = 6$ mice per group). **b** Summary plots of total distance traveled in OFT ($n = 6$ mice per group). **c** Summary plots of time in the open arms in EPMT ($n = 6$ mice per group). **d** Summary plots of open arm entries in EPMT ($n = 6$ mice per group). Data were shown as mean \pm SEM. $*p < 0.05$, $***p < 0.001$. See Supplementary Data 1 for full statistical information. Source data are provided as a Source Data file.

4) Methods: The photometry section is missing important details on data handling and behavioral alignment. Was mouse position tracked with software? How was activity aligned to behavior? How were entries/exits/bouts defined (manual vs. automated, head or centre tracking, bout cutoffs, exclusion criteria)? How was AUC computed (e.g. per bout over a fixed window and then averaged across bouts, or some other approach?). Please also indicate whether the same criteria were applied across animals and recording days.

Response: We thank the reviewer for pointing out the need for additional methodological details regarding fiber photometry. The requested methodological details regarding fiber photometry, which were applied consistently across all animals and recording days, have been added to the revised manuscript (page 17, lines 493-510). Below, we provide a point-by-point response based on the methods described in the manuscript.

1. Mouse position tracking and behavioral alignment:

Mouse position and behavior were tracked via an overhead camera (Softmaze, XR-Video) mounted above the behavioral apparatus. To synchronize neural activity with behavior, the video feed was recorded using screen-capture software (EV Capture) alongside timestamps from the fiber photometry system. This allowed precise alignment of zone entry events with the corresponding calcium signal traces.

2. Definition of entries/exits/bouts:

All behavioral events were defined through offline video analysis. A valid entry was operationally defined as the point when the mouse placed all four paws into the target zone (center zone in the OFT, or open arms in the EPM). An exit was correspondingly defined as the point when all four paws left the target zone. A valid bout was defined as a continuous stay within the zone lasting at least 0.5 seconds.

3. Calculation of the Area Under the Curve (AUC):

The area under the curve (AUC) of the z-scored fluorescence signal was calculated as the sum of z-scores within a fixed time window after entry (0–3 s for OFT center entries; 0–5 s for EPM open-arm entries). For each mouse and test session, AUC values from all events of the same type were averaged to obtain a single session-level mean AUC.

5) Figures 1, 2, 6, and 7: Please specify in the results and/or legend the exact conditions used for the photometry control group.

Response: We sincerely thank the reviewer for the valuable comments. We have now explicitly defined the photometry control groups in the revised manuscript and updated the figure legends accordingly.

In Figures 1 and 2, the control group refers to unstressed control mice (see page 5, lines 113-114 and lines 134-135). These mice expressing GCaMP6s in the respective neuronal populations that were placed in the test room but did not receive foot shock.

In Figures 6 and 7, the control group is a viral control (see page 9, lines 249-250 and page 10, lines 275-276). Mice co-expressing GCaMP6s and mCherry (control virus) in the respective neuronal populations, which underwent the identical FS or sub-FS procedure as the experimental group. The corresponding figure legends have been updated (see page 31, lines 872-873; page 36, lines 974-977; page 38, lines 1004-1006).

6) Fig. 1: The legend should state what the photometry heatmaps represent (e.g. data from a single animal vs. group-average, are rows individual bouts/trials?).

Response: We thank the reviewer for highlighting the need for a more precise description of the photometry heatmaps. To clarify, the heatmaps visualize the trial-averaged neural activity (Z-score), aligned to the behavioral event (entry into the center zone in the OFT or open arms in the EPMT). Specifically, each row in the heatmap represents the average response across all trials for a single animal, and the color scale reflects the fluorescence intensity. We have updated the figure legends for Fig. 1 (page 31, lines 873-875) and related figures in the revised manuscript (now Fig. 1h-p, Fig. 2a-p, Fig. 6p-u, Fig. 7k-p in the revised manuscript).

7) Fig. 4: Please include the mCherry 7d-P-FS results (vehicle and CNO) in the graphs alongside the hM4D results. Statistical testing should be performed between mCherry and hM4D groups (e.g. mixed ANOVA (group, drug) followed by post-hoc tests within each drug condition).

Response: We sincerely thank the reviewer for the thorough and valuable comments. Accordingly, we performed the additional experiments (Response Fig. 9, now Fig. 4 in the revised manuscript).

As recommended, we performed a two-way ANOVA test (factors: Group and Drug) followed by post-hoc tests within each drug condition. Post-hoc comparisons confirmed that CNO-treated, FS-exposed hM4Di mice spent significantly more time in the open field center and in the EPM open arms, along with more open-arm entries, compared to the CNO-treated, FS-exposed mCherry mice or vehicle-treated, FS-exposed hM4Di mice. (Response Fig. 9 d-g, now Fig. 4d-g, with citation on page 6, lines 169-173 in the revised manuscript).

These results provide stronger statistical support for our conclusion that inhibiting BLA→vHPC PNs alleviates FS-induced PTSD-like avoidance behavior in mice.

Response Fig. 9 (related to Fig. 4). Chemogenetic inhibition of BLA→vHPC PNs alleviates FS-induced PTSD-like avoidance behavior in mice. **a** Schematic of the experimental procedures. **b** Representative image showing hM4D (Gi) expression in BLA→vHPC PNs. Scale bar: 500 μm. **c** Representative traces showing CNO-induced inhibition of the firing of BLA→vHPC PNs upon depolarizing current injection. Scale bar: 60 s, 25 mV. **d, e** Summary plots of time in center area (**d**) and total distance traveled (**e**) in OFT (mCherry+Vehicle: $n = 12$ mice; mCherry+CNO: $n = 10$ mice; hM4Di+Vehicle: $n = 12$ mice; hM4Di+CNO: $n = 10$ mice). **f, g** Summary plots of time in open arms (**f**) and open arm entries (**g**) in EPMT (mCherry+Vehicle: $n = 12$ mice; mCherry+CNO: $n = 10$ mice; hM4Di+Vehicle: $n = 12$ mice; hM4Di+CNO: $n = 10$ mice). Data were shown as mean ± SEM. * $p < 0.05$, ** $p < 0.01$. See Supplementary Data 1 for full statistical information. Source data are provided as a Source Data file.

8) Methods: The contextual fear conditioning section has an unfinished sentence. Please also report if any software was used for freezing detection and the criteria used.

Response: *We sincerely thank the reviewer for carefully reading the Methods section and pointing out these omissions. We have corrected the unfinished sentence and added the requested details regarding freezing detection in the revised manuscript (page 15, lines 436-442).*

Reviewer #3 (Remarks to the Author):

I commend the authors on the revised version of the manuscript. The authors have addressed all of my concerns. The experiments are extensive and the results are exciting.

Response: *We are sincerely grateful to the reviewer for their positive assessment of our revised manuscript and for their encouraging comments. We would also like to express our appreciation for the constructive feedback throughout the review process, which has significantly strengthened the quality of our study.*

Reviewer #4 (Remarks to the Author):

The authors have addressed all of my initial concerns. This is an impressive set of experiments!

Response: *We sincerely thank the reviewer for the supportive feedback and for the positive evaluation of*

our work. We greatly appreciate the time and insight invested from reviewer during review process, which has been instrumental in improving our manuscript.